# Identification of *Chlamydia pneumoniae* and NLRP3 inflammasome activation in Alzheimer's disease retina

Bhakta Prasad Gaire [1,16], Yosef Koronyo[1,16], Jean-Philippe Vit[1], Alexandre Hutton [2], Lalita Subedi[3,4], Dieu-Trang Fuchs[1], Natalie Swerdlow[1], Altan Rentsendorj[1], Saba Shahin [1], Daisy Martinon[3,4], Edward Robinson[1], Alexander V. Ljubimov [1,5,6,7], Julie A. Schneider [8], Lon S. Schneider[9,10], Debra Hawes[11], Stuart L. Graham [12], Vivek K. Gupta[12], Mehdi Mirzaei[12,13], Keith L. Black [1], Jesse G. Meyer [2,14], Moshe Arditi [3,4], Timothy R. Crother [3,4] ✉ & Maya Koronyo-Hamaoui [1,5,15] ✉

*Chlamydia pneumoniae* is an intracellular bacterium implicated in Alzheimer's disease (AD), but its role in retinal pathology and disease progression is unclear. Here we identify *Chlamydia pneumoniae* inclusions in the retina, showing higher burden in AD retina and brain, increasing with APOEε4, disease stage, and cognitive deficit. Retinal and cortical proteomics reveal bacterial-infection and related NLRP3-inflammasome pathways. Retinal NLRP3 is elevated in mild cognitive impairment and activated in AD dementia, evidenced by increased caspase-1, cleaved interleukin-1β, and cleaved N-terminal gasdermin-D. *Chlamydia pneumoniae* associates with amyloid-β$_{42}$, inflammation, apoptosis, pyroptosis, and AD status. In neuronal cultures and APP$_{SWE}$/PS1$_{ΔE9}$ model mice, infection induces amyloid-β, inflammasome activation, neuroinflammation, and neurotoxicity, and chronic infection worsens cognition. Fewer pathogen-colocalized microglia are found in AD retinas, implying impaired clearance. Machine learning detects retinal *Chlamydia pneumoniae* or NLRP3, combined with amyloid-β$_{42}$, as predictors of AD diagnosis and stage. These findings support a disease-amplifying role for *Chlamydia pneumoniae* and propose NLRP3-attenuation or antibiotic-based early interventions.

Alzheimer's disease (AD) is a debilitating neurodegenerative condition and the leading cause of dementia in the elderly, currently ranked as the seventh most common cause of death worldwide. Affecting over 55 million individuals globally, with projections indicating a nearly threefold increase in cases by 2050, AD represents a major health crisis with profound social and economic implications[1]. The potential role of infectious agents in AD pathogenesis has gained increasing attention[2–5], with *Chlamydia pneumoniae*, an obligate gram-negative bacterium primarily responsible for community-acquired pneumonia, emerging as a significant pathogen[6–8]. Genome-wide association studies (GWAS) in AD have identified genes associated with immune responses to pathogens, including the *Chlamydia* interactome, which overlaps with the AD hippocampal transcriptome and proteins involved in amyloid β-protein (Aβ) plaques and neurofibrillary tangles (NFTs)[9]. These GWAS findings show significant enrichment of AD-associated genes in pathways relevant to pathogen diversity, implicating their involvement in immune defense and pathogen resistance mechanisms.

*Chlamydia pneumoniae* inclusions have been detected in the postmortem brains of AD patients[10–14]. Immunohistochemical analyses

have revealed *Chlamydia pneumoniae* proteins within vascular endothelial cells, microglia, astrocytes, and neurons, particularly in the frontal and temporal cortices, where *Chlamydia pneumoniae* was localized near Aβ plaques and NFTs[10,12,13]. Additional studies have identified *Chlamydia pneumoniae* DNA in the cerebrospinal fluid (CSF), correlating with an increased risk of developing AD[15], and *Chlamydia pneumoniae* infection has been linked to the progression of dementia[16]. Notably, a recent nationwide cohort study in Taiwan demonstrated that antibiotic treatment targeting *Chlamydia pneumoniae* significantly reduced the risk of AD onset[17]. These findings suggest that *Chlamydia pneumoniae* infection may exacerbate AD pathology and that therapeutic strategies targeting *Chlamydia pneumoniae* could potentially slow or mitigate AD progression.

Furthermore, *Chlamydia pneumoniae* infection has been shown to induce neuroinflammation and promote amyloid aggregation in murine models of AD, processes linked to AD progression[18-20]. Importantly, the nucleotide-binding oligomerization domain, leucine-rich repeat, and pyrin domain-containing 3 (NLRP3) inflammasome, a key mediator of innate immunity activated in response to *Chlamydia pneumoniae* infection[21,22], has been implicated in AD pathogenesis[23-25]. NLRP3 inflammasome has been associated with Aβ-induced tauopathy in murine models[26,27] and was upregulated in the brains of AD patients[28]. The NLRP3 inflammasome and its components, including NLRP3, pro-caspase-1, and the adaptor protein ASC, trigger the release of pro-inflammatory cytokines such as interleukin (IL)1β, and lead to gasdermin D-mediated pyroptotic cell death[29], likely contributing to the chronic inflammation and subsequent neurodegeneration that characterize AD.

Histological, biochemical, and in vivo imaging studies demonstrate that the pathological processes of AD extend beyond the brain to the neurosensory retina, a direct extension of the central nervous system that offers a unique opportunity for live observation[30]. In particular, the pathological hallmarks of AD, abnormal Aβ and tau aggregates, were identified in the retina of patients with mild cognitive impairment (MCI) and AD, along with vascular changes, inflammation, and neurodegeneration[31-44]. Hence, the unique accessibility of the retina to noninvasive imaging with high resolution and specificity may allow for early detection and monitoring of AD[30,45]. Despite emerging evidence indicating the presence of *Chlamydia pneumoniae* and NLRP3 inflammasome activation in the AD brain[10-14,23-25], the role of *Chlamydia pneumoniae* infection and NLRP3-mediated inflammation in the retina remains unexplored. This knowledge gap underscores the need to investigate whether *Chlamydia pneumoniae* infection and NLRP3 inflammasome activation occur in the AD retina, to explore potential interactions during both early and advanced stages of AD, and to examine their correlations with brain pathology and cognitive decline. These insights could unveil *Chlamydia pneumoniae* and NLRP3 inflammasome activation as potential therapeutic targets and biomarkers of AD.

In this study, we provide immunohistochemical evidence of *Chlamydia pneumoniae* inclusions in postmortem retina and matched brain from individuals with normal cognition, MCI (due to AD), and AD dementia, with retinal findings validated by fluorescence in situ hybridization (FISH), Giemsa staining, and genomic DNA quantitative polymerase chain reaction (qPCR). We show that retinal and cerebral *Chlamydia pneumoniae* burden is elevated in AD dementia, but not MCI, and increases with APOEε4 and clinical–neuropathological severity. Consistently, targeted retinal and cortical proteomics reveal dysregulated bacterial-infection pathways, including a *Chlamydia* interactome coupled to NLRP3-inflammasome signaling. In human neuronal cells and APP$_{SWE}$/PS1$_{ΔE9}$ murine model of AD, infection induces Aβ accumulation, NLRP3 activation, neuroinflammation and cytotoxicity, and chronic infection exacerbates neuropathology and cognitive decline. In human retinas, *Chlamydia pneumoniae* associates with Aβ$_{42}$, gliosis, apoptosis, gasdermin-dependent pyroptosis, and

tissue atrophy, alongside a relative deficit of pathogen-engaged microglia, consistent with impaired clearance. Finally, machine-learning models indicate that retinal *Chlamydia pneumoniae* and NLRP3, alone and combined with Aβ$_{42}$, may predict AD diagnosis, stage, and cognitive status, supporting a role for *Chlamydia pneumoniae* in disease progression and motivating NLRP3-targeted and/or antibiotic-based early treatment strategies.

## Results

To investigate *Chlamydia pneumoniae* infection and its association with neuroinflammation and neurodegeneration in the AD retina, we analyzed retinal and brain tissues from a cohort of 104 patients: 51 with AD dementia (mean age ± SD: 85.90 ± 10.06 years, 29 females/22 males), 16 with MCI due to AD (89.43 ± 6.90 years, 7 females/9 males), and 37 normal cognition controls (83.96 ± 10.96 years, 21 females/16 males). There were no significant differences in age, sex, or postmortem interval across groups. Demographic, clinical, and neuropathological characteristics of donor cohorts for retinal and brain histological and biochemical analyses are summarized in Fig. 1A, Table 1, and Supplementary Tables 1–5. To investigate protein expression profiles related to *Chlamydia pneumoniae* infection, we reanalyzed our recently reported mass spectrometry (MS)-based proteomic datasets[33] from human retinal and cortical tissues. Complementary Western blot and genomic DNA qPCR assays were performed on retinal samples. Immunohistochemical (IHC) analyses focused on the superior-temporal (ST) retina and dorsolateral prefrontal cortex (A9), given their strong association with AD pathology[32,33,46]. In parallel, we examined whether *Chlamydia pneumoniae* infection modulates AD-related pathological progression in human neuronal cells and AD+ mice (n = 45).

### *Chlamydia pneumoniae* burden increases in AD dementia retina and brain

We applied a stepwise strategy to establish the existence and distribution of *Chlamydia pneumoniae* in the human retina. Initially, we broadly screened for *Chlamydia* species by performing IHC with an anti-*Chlamydia* polyclonal antibody (pAb) verified against *Chlamydia pneumoniae*, *Chlamydia trachomatis*, and *Chlamydia psittaci*, and thus capable of cross-reacting with these species. Using this antibody, *Chlamydia*-positive inclusion bodies were readily visualized by both fluorescence-based (Fig. 1B; n = 6) and peroxidase-based (Supplementary Fig. 1A, B; n = 6) immunolabeling. We then confirmed the specific presence of *Chlamydia pneumoniae* inclusions in retinal cross-sections using an anti-*Chlamydia pneumoniae* monoclonal antibody (mAb) that does not cross-react with other *Chlamydia* species, with peroxidase-based (n = 18) and fluorescence (n = 69) detection (Fig. 1C, D and Supplementary Fig. 1C, D). All quantitative analyses of *Chlamydia pneumoniae* burden reported in this study, including positive cell counts, immunoreactive (IR) area measurements, and correlation analyses, are based exclusively on this *Chlamydia pneumoniae*–specific mAb. The retinas of MCI and AD patients, compared with those of normal cognition controls, exhibited predominant *Chlamydia pneumoniae*-positive signals in the retinal ganglion cell layer (GCL) and inner nuclear layer (INL). Most *Chlamydia pneumoniae* inclusions were observed as cytosolic puncta aggregates, whereas other inclusions were detected in the peri-nucleus or nucleus and colocalized with DAPI, resembling the patterns observed using the anti-*Chlamydia* pAb. *Chlamydia pneumoniae* inclusions were also detected with the mAb in the corresponding A9 cortices of AD patients (Supplementary Fig. 1C, E), identifying typical intracellular inclusions similar to previously reported inclusion patterns in the AD brain[12,13].

To quantify *Chlamydia pneumoniae* burden in the retina, we first performed a cell count–based analysis, which revealed 2.1- and 1.9-fold increases in retinal *Chlamydia pneumoniae*–positive cells in the AD

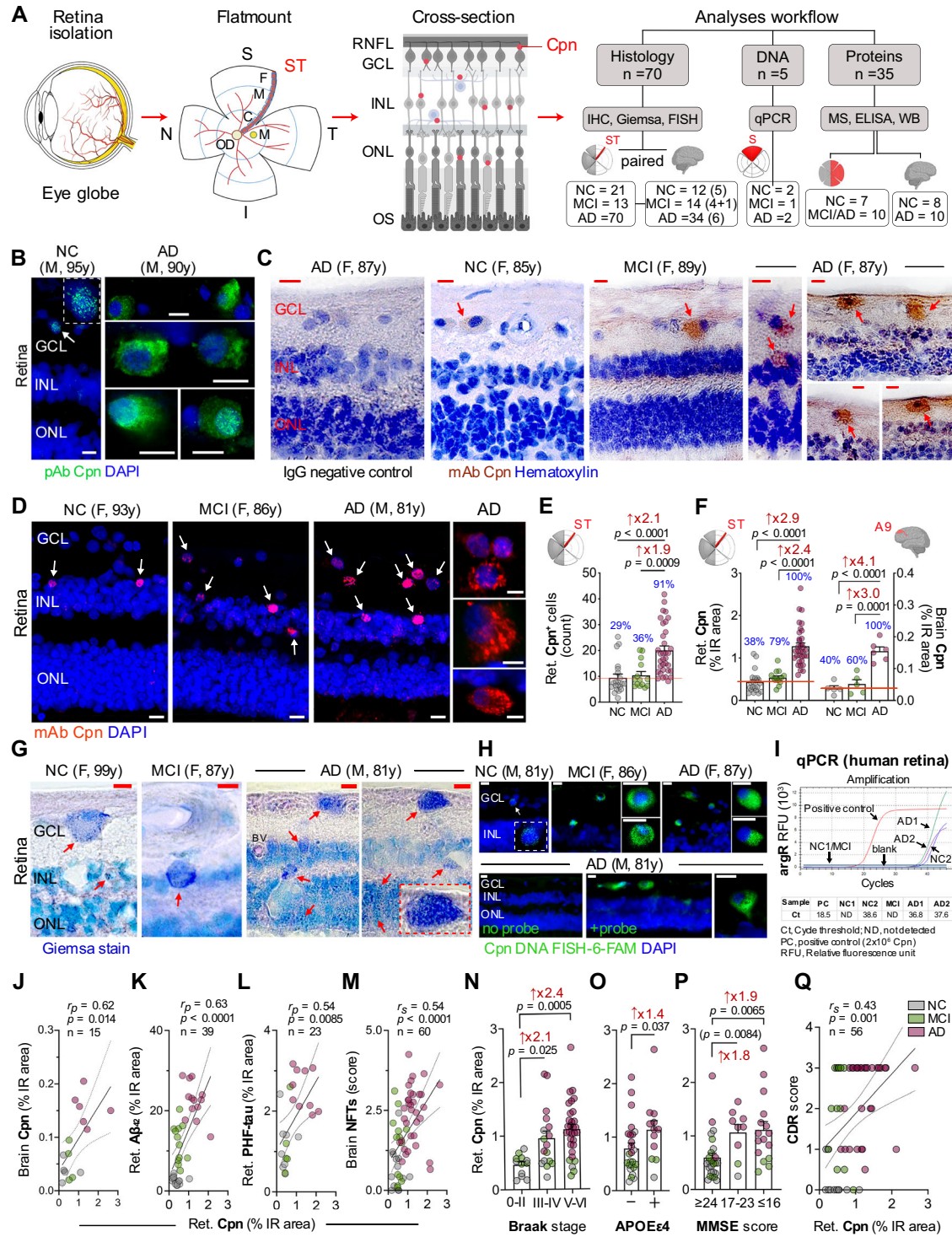

group compared with normal cognition and MCI groups, respectively (Fig. 1E; $p < 0.001$–0.0001). In addition, we analyzed the IR area of retinal and paired brain *Chlamydia pneumoniae* to determine a more integrated measure of bacterial burden, revealing significant 2.9- and 4.1-fold increases in *Chlamydia pneumoniae* inclusions, respectively, in AD patients compared with normal cognition controls (Fig. 1F; $p < 0.0001$; retina: $n = 69$, brain: $n = 16$). Consistent across both analyses (Fig. 1E, F), no significant difference in *Chlamydia pneumoniae* load was observed between normal cognition and MCI groups, indicating that expansion of *Chlamydia pneumoniae* infection likely spreads later in disease progression, during the clinical dementia stages of AD. Indeed, Gaussian distribution curves for both retinal and

brain *Chlamydia pneumoniae* levels showed strong overlap between the normal cognition and MCI groups compared with the AD group (Supplementary Fig. 1F, G). Yet, the proportion of individuals with retinal or brain *Chlamydia pneumoniae* levels exceeding the normal-cognition mean (red line) was 60–79% in MCI and 100% in AD dementia, compared with only 38–40% in the normal-cognition group (Fig. 1F). Examination of retinal *Chlamydia pneumoniae* spatial distribution indicated a uniform burden across central (C), mid-peripheral (M), and far-peripheral (F) ST subregions (Supplementary Fig. 2A, B). Retinal *Chlamydia pneumoniae* levels did not differ between males and females within any diagnostic group (Supplementary Fig. 2C), indicating no sex-specific dimorphism in retinal burden.

**Fig. 1 | Identification of *Chlamydia pneumoniae* inclusions in retinas and brains from MCI and AD patients and correlations with disease status. A** Schematic of Retinal isolation and cross-section preparation. Red puncta (red) indicate putative *Chlamydia pneumoniae* (Cpn) inclusions. Analyses workflow summarizes cohort sizes. Numbers in parentheses indicate brain subsets for Cpn histology. **B** Immunofluorescence using anti-Cpn polyclonal antibody (pAb, green, white arrow; 3 repetitions) in 3 AD versus 3 normal cognition (NC) retinas. **C** Peroxidase-based immunohistochemistry using anti-Cpn monoclonal antibody (mAb, brown, red arrows) in 6 MCI and 9 AD versus 3 NC retinas; hematoxylin (purple) and IgG negative control (3 repetitions). **D** Retinal Cpn mAb immunofluorescence (red; white arrows, 3-repetition); cytosolic inclusions at higher magnification. **E** Quantification of retinal Cpn+ cell count (21 NC, 14 MCI, 34 AD). **F** Quantification of Cpn immunoreactivity (%IR area) in same retinal cohort (21 NC, 14 MCI, 34 AD) and paired brain tissues (5 NC, 5 MCI, 6 AD). Subjects with %IR area above the NC mean (red line) were classified as Cpn-positive. **G** Giemsa staining visualizes retinal Cpn-like inclusions (dark-blue, red arrows; BV, blood vessel; $n = 8$ donors, 4 repetitions). **H** FISH confirms retinal Cpn-specific genomic material (green, white arrow; $n = 11$ donors, 3 repetitions). **I** qPCR curves (Ct values) for the Cpn *argR* gene in 2 NC, 1 MCI, and 2 AD retinas. Pearson's correlation ($r_P$) between retinal Cpn and **(J)** brain Cpn, **(K)** retinal Aβ$_{42}$, or **(L)** PHF-tau. **M** Spearman's correlation ($r_s$) between retinal Cpn and brain NFT severity. Retinal Cpn stratified by **(N)** Braak stage ($n = 12$ Braak 0–II, 17 Braak III–IV, 31 Braak V–VI), **(O)** APOEε4 genotype ($n = 12$ carriers, 25 non-carriers), and **(P)** MMSE score ($n = 26$ MMSE ≥ 24, 9 MMSE 17–23, 16 MMSE ≤ 16). **Q** Correlation ($r_s$) between retinal Cpn burden and CDR score. M-male, F-female, and age (y-years) are shown. All scale bars, 10 μm. Data are shown as individual values with means ± SEMs. Fold changes are shown in red. *p* values were determined by one-way ANOVA with Tukey's post-hoc test (**E, F, N, P**), two-sided unpaired *t*-test (**P**), or Mann–Whitney *U*-test (**O**). Illustration **A** was created in Biorender.com. Fuchs, D. (2026) https://BioRender.com/msadzfx. Source data are provided as a Source Data file.

**Table 1 | Demographic and neuropathological data on human donors for histological analysis**

| | | Normal cognition | MCI | AD | F value | p value |
|---|---|---|---|---|---|---|
| Human donors ($n = 70$) | | 21 | 15 | 34 | | |
| Female (%), Male | | 11 (52%), 10 | 7 (47%), 8 | 17 (50%), 17 | | |
| Age at death (years) | | 84.57 ± 10.94 | 88.60 ± 6.25 | 86.18 ± 9.10 | 0.84 | 0.437 |
| Race (No.) | | White (17); Black/His-panic (1/3) | White (13); Black/His-panic (1) | White (26); Hispanic/Asian (5/3) | | |
| Postmortem interval (hours) | | 9.03 ± 5.88 | 12.23 ± 11.47 | 10.47 ± 11.85 | 0.39 | 0.676 |
| MMSE Score ($n = 51$) | | 29.06 ± 1.78 | 22.00 ± 6.93 | 14.50 ± 7.83 | 26.29 | <0.0001 |
| CDR Score ($n = 57$) | | 0.17 ± 0.39 | 1.97 ± 1.18 | 2.32 ± 0.91 | 24.43 | <0.0001 |
| Brain Neuropathology [Severity Score ($n = 61$)] | Sample size | 12 | 15 | 34 | | |
| | Braak Stage (%) | 0 (16.67) I-II (41.67) III-IV (33.33) V-VI (8.33) | 0 (6.67) I-II (33.33) III-IV (33.33) V-VI (26.66) | 0 (0) I-II (0) III-IV (23.53) V-VI (76.47) | 30.01 | <0.0001 |
| | ABC average (Amyloid, Braak, CERAD) | 1.43 ± 0.79 | 1.98 ± 0.60 | 2.75 ± 0.38 | 30.72 | <0.0001 |
| | Aβ Plaque | 0.93 ± 1.05 | 1.64 ± 0.90 | 2.53 ± 1.00 | 12.89 | <0.0001 |
| | NFT | 0.54 ± 0.46 | 1.44 ± 0.97 | 2.36 ± 0.90 | 21.67 | <0.0001 |
| | NT | 0.77 ± 0.96 | 0.90 ± 0.78 | 2.01 ± 1.14 | 9.83 | 0.0002 |
| | Atrophy | 0.55 ± 0.63 | 1.14 ± 0.91 | 1.99 ± 1.08 | 11.13 | <0.0001 |

Paired brains with full neuropathological assessments were available for 61 human donors ($n = 12$ normal cognition, $n = 15$ MCI, $n = 34$ AD). Values are presented as mean ± standard deviation. F and *p* values were determined by one-way ANOVA with Tukey's multiple comparisons test. Mean ABC scores were determined as: A, Aβ plaque score modified from Thal; B, NFT severity score modified from Braak; C, neuritic plaque score modified from CERAD. *Aβ* amyloid beta-protein, *AD* Alzheimer's disease, *CDR* Clinical dementia rating, *CERAD* consortium to establish a registry for Alzheimer's disease, *IHC* immunohistochemistry, *MCI* mild cognitive impairment, *MMSE* mini-mental state examination, *NFT* neurofibrillary tangle, *NT* neuropil thread.

The existence of *Chlamydia pneumoniae* in the human retina was further validated using three complementary histological and molecular approaches: Giemsa staining, FISH, and genomic DNA detection by real-time qPCR (Fig. 1G–I; extended data in Supplementary Fig. 3–4). Although Giemsa staining is not specific to a particular bacterial species, inclusion bodies in retinal cross-sections appeared as dark blue structures, consistent with those observed in *Chlamydia pneumoniae*–infected mouse lung tissues (Fig. 1G and Supplementary Fig. 3 A, B; $n = 8$ donors) and with *Chlamydia pneumoniae* inclusions identified by immunostaining with the mAb in the AD retina and brain (Fig. 1C, D). A FISH analysis using a fluorescently labeled *Chlamydia pneumoniae*-specific DNA probe further verified the presence of this bacterial genomic material within retinal tissues (Fig. 1H and Supplementary Fig. 4; $n = 11$ donors), which was absent in the no probe retinal AD and normal cognition tissues. Both Giemsa and FISH analyses demonstrated a higher burden of inclusions in AD retinas compared with those from individuals with normal cognition. Notably, qPCR analysis detected the *Chlamydia pneumoniae*–specific *arginine repressor (argR)* gene in retinal tissues from 2 of 2 AD cases, 0 of 1 MCI case, and 1 of 2 normal cognition controls, confirming the presence of

*Chlamydia pneumoniae* in the human retina (Fig. 1I; $n = 5$ donors). Moreover, Pearson's correlation ($r$) between retinal and corresponding brain *Chlamydia pneumoniae* burdens revealed a strong concordance of bacterial load in both central nervous system (CNS) tissues (Fig. 1J; $r_p = 0.62$, $p = 0.0143$); extended correlations between *Chlamydia pneumoniae* load across retinal subregions and the brain showed the strongest association for the mid-periphery (Supplementary Fig. 5A, B; $r_p = 0.71$, $p = 0.0046$).

Given the strong connection between retinal and brain *Chlamydia pneumoniae* burden, we next examined how this bacterial load relates to AD relevant pathology and disease severity across both CNS tissues. We found a strong correlation between retinal *Chlamydia pneumoniae* burden and retinal Aβ species (Fig. 1K and Supplementary Table 8; Aβ$_{42}$: $r_p = 0.63$, $p < 0.0001$, Aβ$_{40}$: $r_p = 0.65$, $p = 0.0014$), with no correlation with intracellular Aβ oligomers (Supplementary Table 8), indicating a specific association with the extracellular plaque-dominant Aβ$_{42}$ and vascular-dominant Aβ$_{40}$ alloforms. Significant correlations between retinal *Chlamydia pneumoniae* burden and markers of retinal tauopathy were also detected, including paired-helical filament (PHF)-tau ($r_p = 0.54$, $p = 0.0085$, Fig. 1L), hyperphosphorylated (p)S396-tau

($r_p = 0.38$, $p = 0.0116$), T22$^+$ oligomeric tau ($r_p = 0.43$, $p = 0.0040$), and citrullinated tau (CitR$_{209}$: $r_p = 0.48$, $p = 0.0028$; Supplementary Table 8). Retinal *Chlamydia pneumoniae* burden showed no significant association with retinal AT8-positive p-tau or MC-1-positive mature tau tangles (Supplementary Table 8), nor with p-tau/total tau ratios at phosphorylation sites S404, S396, S199, S231, or S214 quantified by NanoString GeoMx digital spatial profiling in a subset of this cohort[40] (Supplementary Fig. 5C–G; $n = 22$ donors). Overall, these data indicate that *Chlamydia pneumoniae* inclusions occur in the MCI and AD retina, predominantly in the GCL and INL, closely interact with amyloidogenic Aβ, and modestly associate with certain retinal tau isoforms but not with others.

Next, we determined whether retinal *Chlamydia pneumoniae* burden associates with AD-related brain pathology, apolipoprotein E (APOE) ε4 allele, disease staging, or the extent of cognitive deficit (Fig. 1M–Q; extended data in Supplementary Fig. 5H–J and Supplementary Table 8). We found that retinal *Chlamydia pneumoniae* significantly correlated with the severity of brain NFTs (Fig. 1M; $r_s = 0.54$, $p < 0.0001$) and was 2.1-2.4-fold higher in patients with advanced Braak stages (Fig. 1N; Stage III-IV or V-VI versus 0-II: $p < 0.05$–0.001, $n = 60$ donors), suggesting *Chlamydia pneumoniae*'s involvement in brain tauopathy progression. Additionally, retinal *Chlamydia pneumoniae* burden significantly correlated with the following brain pathologies: Aβ plaques ($r_s = 0.40$, $p = 0.0014$), ABC severity score ($r_s = 0.54$ $p < 0.0001$), neuropil threads (NT; $r_s = 0.37$, $p = 0.0033$), cerebral amyloid angiopathy (CAA; $r_s = 0.35$, $p = 0.0057$), gliosis ($r_s = 0.40$, $p = 0.0016$), and brain atrophy ($r_s = 0.48$, $p = 0.0001$; Supplementary Table 8). Notably, both retinal and brain *Chlamydia pneumoniae* burdens were higher in APOE ε4 allele carriers compared with non-carriers, regardless of AD diagnosis (Fig. 1O; $p = 0.037$, $n = 37$; and Supplementary Fig. 5H; a trend, $p = 0.06$, $n = 13$).

Common bacterial infections, such as *Helicobacter pylori*, *Chlamydia pneumoniae*, *Borrelia burgdorferi*, and spirochetal *Treponema* have previously been linked to cognitive decline and increased dementia risk in elderly[47]. In this cohort, retinal *Chlamydia pneumoniae* burden inversely correlated with Mini-Mental State Examination (MMSE) scores (Fig. 1P, $n = 47$ donors; and Supplementary Fig. 5I; $r_s = -0.53$, $p < 0.0001$), Clinical Dementia Rating (CDR) scores (Fig. 1Q; $r_s = -0.43$, $p = 0.0010$), and Montreal Cognitive Assessment (MOCA) scores (Supplementary Fig. 5J; $r_s = -0.56$, $p = 0.0334$; Supplementary Table 8), reinforcing the association between retinal *Chlamydia pneumoniae* load and global cognitive impairment. Despite the modest cohort size ($n = 15$), brain *Chlamydia pneumoniae* burden strongly correlated with increased AD brain pathology, including ABC score, Braak stage, NFTs, NTs, gliosis, and atrophy (Supplementary Table 9; $r_s = 0.60$–0.77, $p < 0.05$–0.001), showed a moderate association with cerebral amyloid angiopathy (CAA) scores but not Aβ-plaque burden, and was also strongly associated with poorer MMSE performance ($r_s = -0.73$, $p = 0.0043$). Collectively, these data tightly link retinal and brain *Chlamydia pneumoniae* burden to APOE ε4 status, widespread AD neuropathology, and global cognitive deterioration.

### *Chlamydia* interactome links to NLRP3 activation and cell death in AD

Detection of gram-negative *Chlamydia pneumoniae* inclusions in the retinas and brains of AD patients prompted us to investigate infection-driven protein dysregulation in these tissues by performing a secondary MS-based proteomic reanalysis in independent human retinal and cortical cohorts[33] (Supplementary Tables 2–5; retina: $n = 12$, brain: $n = 18$). Metascape gene ontology (GO) analysis identified multiple dysregulated human proteins implicated in response to bacterial infection, including gram-negative intracellular bacteria, in the brains and retinas of AD patients (Fig. 2A, B), suggesting a significant involvement of bacterial infection in AD pathogenesis. To gain a closer look

at *Chlamydia* infection, we searched for differentially expressed proteins (DEPs) in AD versus normal cognition brains and retinas that were included in the *Chlamydia* interactome (Fig. 2C, D; extended data on up- and down-regulated DEPs in Supplementary Tables 10 and 11). Of the 787 proteins in the *Chlamydia* interactome[48–52], 607 were identified in human brain, of which 84 were differentially expressed (52 down-regulated, 32 upregulated; 13.8%) in AD patients compared with normal-cognition controls (Fig. 2C). Importantly, despite being derived from separate cohorts, similar bacterial infection-associated pathways (Fig. 2A, B) and dysregulated *Chlamydia* interactome DEPs were identified in the AD retina (Fig. 2D and Supplementary Fig. 6A), with 52 downregulated DEPs and 40 upregulated DEPs (13.0% DEPs) among the 710 identified (Fig. 2D). GO network analysis further revealed enrichment of proteins involved in immune responses to microorganisms and cell death in AD brains and retinas (Fig. 2E, F, Supplementary Fig. 6B–D, and Supplementary Fig. 7A–C). Inflammation-related proteins were primarily associated with cytokine signaling, toll-like receptor (TLR) pathways, interferon responses, nuclear factor kappa B (NF-κB) activation, NLRP3 inflammasome activation, and pyroptosis, pathways typically triggered by gram-negative bacteria in peripheral tissues[53,54]. These data suggest shared infection- and immune-associated mechanisms in AD brains and retinas.

*Chlamydia* has been shown to trigger the host's innate immune response, requiring TLR2/MYD88 signaling and NLRP3/ASC/caspase-1 inflammasome activation[21,22]. Indeed, both MYD88 innate immune signal transduction adaptor (MYD88) and PYD and CARD domain containing protein (PYCARD or ASC) were upregulated in the AD retina (Fig. 2F). Additionally, the DNA pathogen sensor and *Chlamydia* interactor, leucine-rich repeat-binding FLII interacting protein 1 (LRRFIP1), which positively regulates TLR4 by competing with FLII actin remodeling protein (FLII) for interaction with MYD88[55], was upregulated in both the AD brain and retina (Fig. 2F and Supplementary Fig. 6A). Importantly, retinal AD proteome was enriched in proteins linked to pyroptosis (Fig. 2E and Supplementary Fig. 6B, D), a form of inflammatory regulated necrosis triggered by intracellular pathogens, including *Chlamydia*[56]. Notably, three members of the gasdermin (GSDM) family, GSDMD, GSDME (or DFNA5), and GSDMA[33], which are involved in pyroptosis/necrosis, were upregulated in the AD retina (Fig. 2F). Proteins involved in apoptosis, pyroptosis, and inflammation were generally associated with levels of cerebral and retinal tau isoforms quantified by MS and with retinal Aβ$_{1-42}$ measured by enzyme-linked immunosorbent assay (ELISA) (Supplementary Figs. 7D–H and 8A–J). Notably, retinal Aβ$_{1-42}$ levels strong-to-very strongly correlated with proteins associated with cell degeneration, including Casp3 ($r_p = 0.77$, $p = 0.0099$), Bcl-2-associated athanogene 3 (BAG3, $r_p = 0.76$, $p = 0.012$), and GSDMD ($r_p = 0.89$, $p = 0.0006$), as well as inflammatory regulators such as dermcidin (DCD, $r_p = 0.78$, $p = 0.0084$) and LRRFIP1 ($r_p = 0.81$, $p = 0.0046$) (Supplementary Fig. 8A–E). In contrast, cytoprotective and anti-inflammatory proteins, including thiol methyltransferase 1 A (TMT1A, $r_p = -0.77$, $p = 0.015$) and adaptor protein complex 2, alpha 2 subunit (AP2A2, $r_p = -0.88$, $p = 0.0008$), were inversely correlated with retinal Aβ$_{1-42}$. Similar to retinal amyloidosis, *Chlamydia* interactome proteins also exhibited significant associations with retinal (0N4R) tau isoform and brain (1N3R, 2N4R) tau isoforms (Supplementary Figs. 8F–J and 7G, H). In particular, retinal 0N4R tau strong-to-very strongly correlated with GSDMD ($r_p = 0.65$, $p = 0.022$), BAG3 ($r_p = 0.85$, $p = 0.0005$), RAD23 nucleotide excision repair protein B (RAD23B, $r_p = 0.90$, $p < 0.0001$), LRRFIP1 ($r_p = 0.81$, $p = 0.0015$), calpastatin (CAST, $r_p = 0.91$, $p < 0.0001$), and TMT1A ($r_p = -0.92$, $p < 0.0001$; Supplementary Fig. 8F–J). Overall, these findings indicate enrichment of proteins implicated in intracellular gram-negative bacterial infection, including *Chlamydia*-associated proteins, together with signatures of inflammasome activation and degeneration in AD brain and retina.

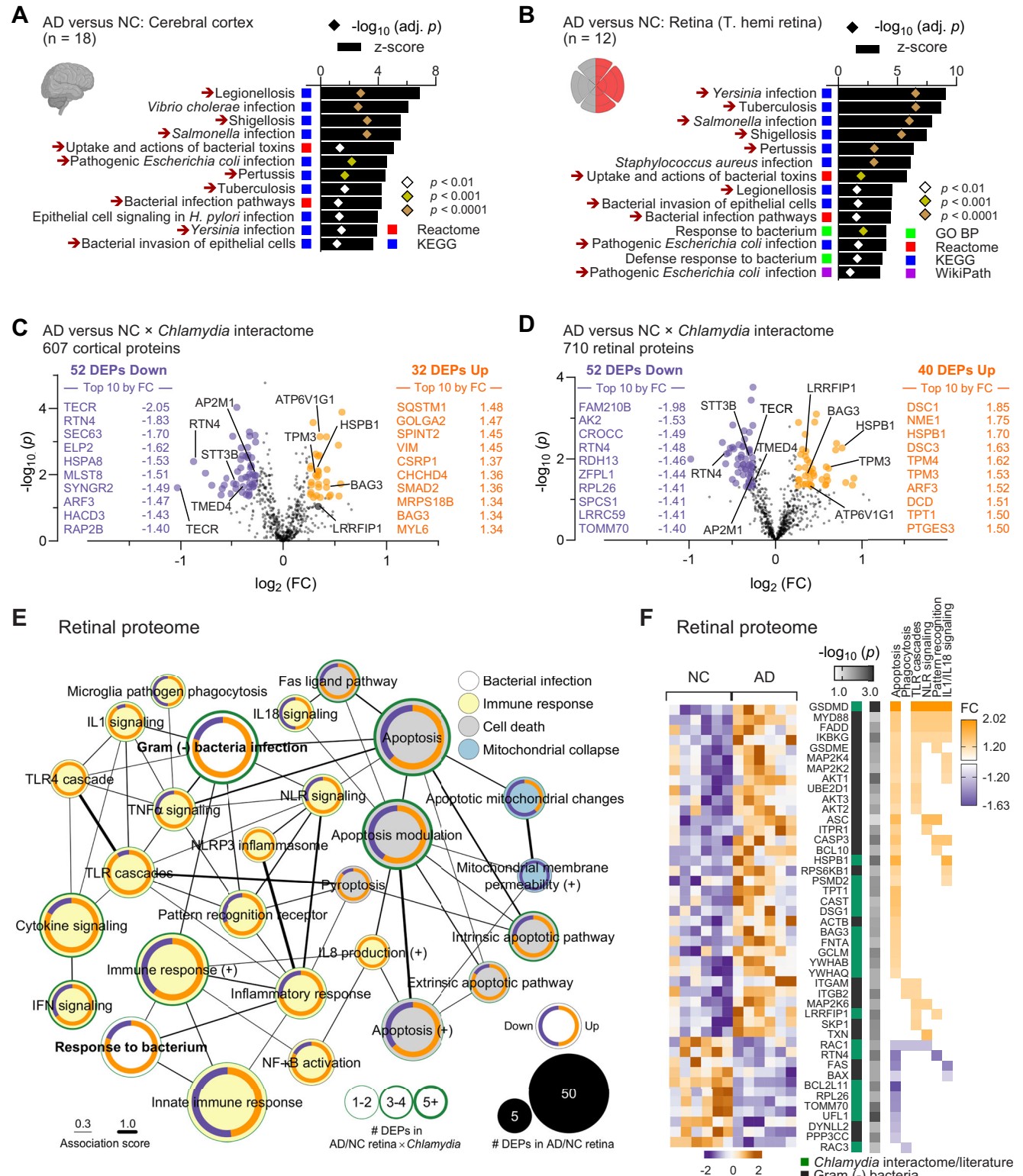

### *Chlamydia pneumoniae* drives AD pathology and worsens cognition

To determine whether *Chlamydia pneumoniae* acts as a driver rather than a bystander in AD, we next tested whether infection of neuronal cells and AD transgenic mice is sufficient to trigger inflammasome activation and exacerbate AD-related pathology. Infection of SH-SY5Y human neuroblastoma cells with *Chlamydia pneumoniae* (multiplicity of infection [MOI] 5) for 68 hours markedly induced $A\beta_{42}$, NLRP3, and IL1β levels and triggered cell membrane damage, as assessed by lactate

dehydrogenase (LDH) release (Fig. 3A–F). Immunocytochemistry confirmed robust *Chlamydia pneumoniae* infection of SH-SY5Y neurons and revealed that infected cells, compared with uninfected controls, exhibited 2.5-fold higher NLRP3, 3.2-fold higher IL1β, and 3.5-fold higher H31L21+ $A\beta_{42}$ IR areas (all $p < 0.0001$; Fig. 3B, C; extended data in Supplementary Fig. 9A; $n = 6$ wells per condition, $n = 41$–74 cells per group). Moreover, pronounced 3.9-fold increase in LDH leakage was detected in infected neuronal cells versus uninfected controls (Fig. 3D, $p < 0.001$, $n = 6$ wells per group), as assessed by LDH release into the

**Fig. 2 | Bacterial infection-associated proteome pathways in AD retina and cerebral cortex.** Gene ontology (GO) analysis of differentially expressed proteins (DEPs) related to bacterial infection (**A**) in the cerebral temporal cortex and (**B**) in the temporal hemi retina from 2 separate cohorts of human donors with AD ($n = 10$ brains, 6 retinas) versus NC ($n = 8$ brains, 6 retinas). The analysis was carried out in Metascape and included the Reactome, Kyoto Encyclopedia of Genes and Genomes (KEGG), and WikiPathways (WikiPath) databases. Red arrows indicate the shared pathways between brain and retina. Bar and symbol graphs represent z-scores, and Benjamini-Hochberg adjusted *p*-values from Metascape analysis, respectively. Range of *p*-values is presented as color-coded symbols. Volcano plots display the fold changes [$\log_2$(FC)] and significance level [$-\log_{10}$(*p*)] by two-sided *t*-test in the (**C**) cortex and (**D**) retina of AD versus NC subjects for *Chlamydia* inclusion interactome. Top 10 DEPs by FC upregulated (orange) and downregulated (purple) interactors are shown. The highlighted proteins, five downregulated and five upregulated, were found in both tissues. **E** GO network (Metascape) of enriched retinal pathways related to bacterial infection, immune response cell death, and mitochondrial collapse. The size of the nodes represents the number of DEPs, with the inner ring showing the proportion of these DEPs that are downregulated (purple) or upregulated (orange) in AD. The thickness of the green border represents the number of DEPs that interact with *Chlamydia* inclusion. The thickness of the connection lines between nodes represents the shared DEPs (association score) between pathways. **F** Heatmaps of upregulated (orange) and downregulated (purple) DEPs [$-\log_{10}$(*p*) by two-sided *t*-test and FC] normalized by unit variance scaling and generated in ClustVis in AD versus NC retina for selected pathways. Only proteins connected to gram-negative bacterial infection (Metascape analysis) and *Chlamydia* infection (*Chlamydia* interactome and literature) are shown for each pathway. Clustering of DEPs was carried out manually based on their involvement in select pathways for visual clarity. Source data are provided as a Source Data file.

culture medium, suggesting that *Chlamydia pneumoniae* infection induces neurotoxicity. We further substantiated these findings by Western blot analysis, which demonstrated increases in NLRP3, cleaved IL1β (1.5-fold, $p < 0.05$), 12F4$^+$ Aβ$_{42}$ (6.7-fold, $p < 0.05$), and N-terminal cleaved gasdermin D (NGSDMD) in infected neurons compared with controls (Fig. 3E, F; $n = 3–6$ wells per group). Together, these findings demonstrate that *Chlamydia pneumoniae* infection is sufficient to drive NLRP3 inflammasome activation, pyroptotic cell death, and Aβ$_{42}$ accumulation in neuronal cells—cellular features of AD pathology. Future studies will be required to delineate the molecular pathways by which *Chlamydia pneumoniae* amplifies Aβ production, sustains inflammasome signaling, and ultimately promotes neurodegeneration.

These in vitro observations prompted us to examine the impact of acute and long-term *Chlamydia pneumoniae* infection on in vivo Alzheimer-like pathology and cognition in APP$_{SWE}$/PS1$_{ΔE9}$ (AD$^+$) mouse models (Fig. 3G–S). In the acute *Chlamydia pneumoniae* infection paradigm, we examined 8 phosphate-buffered saline (PBS)-treated AD$^+$ controls and 14 infected AD$^+$ mice ($n = 22$). In the long-term paradigm, we studied 6 PBS-treated wild-type (WT), 8 PBS-treated AD$^+$, and 9 infected AD$^+$ mice ($n = 23$). Intranasal inoculation with *Chlamydia pneumoniae* ($1 \times 10^6$ inclusion-forming units, IFU) resulted in a marked increase in bacterial inclusions in the AD$^+$ mouse brain (Fig. 3G, H), as confirmed by higher *Chlamydia pneumoniae* IFUs in HEp2 cells treated with brain lysates from infected versus uninfected mice and by increased *Chlamydia pneumoniae* genomic DNA copy numbers (Fig. 3H). Seven-days (acute) *Chlamydia pneumoniae* postinfection caused a 3.2-fold increase in ionized calcium-binding adaptor molecule 1 (IBA1)$^+$ microgliosis and a 2.5-fold increase in glial fibrillary acidic protein (GFAP)$^+$ astrogliosis in the hippocampi and cortices of infected versus uninfected AD$^+$ mice (Fig. 3I, J; $p < 0.05–0.01$; extended data in Supplementary Fig. 9B–D), indicating amplified neuroinflammation due to infection. Furthermore, infected AD$^+$ mice exhibited elevated mRNA expression of *Il6* ($p < 0.01$), *Il1β* ($p < 0.05$), and *Nlrp3* ($p < 0.01$) (Fig. 3K), further supporting activation of NLRP3-inflammasome signaling in response to *Chlamydia pneumoniae* infection. These findings demonstrate that acute intranasal infection is sufficient for *Chlamydia pneumoniae* to reach the brain, establish infection, and subsequently trigger inflammasome activation and neuroinflammation.

The long-term behavioral and pathological consequences of *Chlamydia pneumoniae* infection in AD$^+$ mice were assessed 6 months after a single intranasal inoculation ($1 \times 10^6$ IFU *Chlamydia pneumoniae* or PBS) administered at 8 months of age (Fig. 3L–S; extended data in Supplementary Fig. 9E–J). Multidomain behavioral testing was conducted over 12 days in 14-month-old mice, with PBS-treated WT animals serving as healthy behavioral controls. In the open field and X-maze tests, *Chlamydia pneumoniae* infection did not affect locomotor function of AD$^+$ mice, as indicated by rearing and ambulatory activity or total arm entries (Fig. 3L and Supplementary Fig. 9E, H).

However, alternations in both color- and contrast-stimuli modes of the X-maze, which assess visuo-cognitive function and are decreased in AD$^+$ mice[57], were further reduced in *Chlamydia pneumoniae*-infected AD$^+$ mice (Fig. 3M, N). In the color-mode X-maze, while infection did not affect specific arm entries, it further decreased bidirectional blue (B)↔white (W) transitions in infected AD$^+$ mice, indicating color vision dysfunction (Fig. 3M and Supplementary Fig. 9F, G). In the contrast-mode X-maze, entries into the arm with the white object were increased in the PBS-control AD$^+$ mice and further increased in the *Chlamydia pneumoniae*-infected AD$^+$ mice (Fig. 3N and Supplementary Fig. 9I). The infected AD$^+$ mice also exhibited increased black (B)↔W and decrease of black (B)↔clear (C) bidirectional transitions, indicating a worsening of contrast sensitivity vision due to infection (Supplementary Fig. 9J). In the Barnes maze, PBS-control AD$^+$ mice (vs. WT) made significantly more errors prior to finding the escape box, during the 4-day acquisition phase, the long-term memory retention phase, and the 2-day reversal phase (Fig. 3O). Importantly, *Chlamydia pneumoniae* infection in AD$^+$ mice further increased the number of errors made on reversal day 9 (Fig. 3O, P), which measures spatial learning and cognitive flexibility. Search coverage analysis showed that *Chlamydia pneumoniae*-infected AD$^+$ mice made more errors locally in the area that is both on the side of the old and new escape box locations (Fig. 3P). These results indicate that long-term *Chlamydia pneumoniae* infection exacerbates visuocognitive dysfunction in AD$^+$ mice without affecting locomotor function.

We subsequently examined AD-related pathology in the cortex and hippocampus of long-term infected versus uninfected AD$^+$ mice (Fig. 3Q–S). Our analysis revealed significant increases in 6E10$^+$ Aβ plaques (1.6-fold, $p < 0.001$), IBA1$^+$ microglia (1.3-fold, $p < 0.01$), and GFAP$^+$ astrocytes (1.3-fold, $p < 0.05$) in the cortex of *Chlamydia pneumoniae* infected AD$^+$ mice compared with PBS-administered AD$^+$ mice (Fig. 3R). Similar increases were also observed in the hippocampus (Fig. 3S). These findings demonstrate that long-term *Chlamydia pneumoniae* infection in AD$^+$ mice aggravates neuroglial activation and Aβ pathology, supporting the hypothesis that chronic infection exacerbates AD-like neuropathology.

### Retinal NLRP3 activation links *Chlamydia pneumoniae* to cell death

Convergent evidence from MS-based proteomics, *Chlamydia pneumoniae*-infected cell cultures, and AD$^+$ mouse models implicating this pathogen in cerebral NLRP3 inflammasome activation, together with prior murine infection studies demonstrating *Chlamydia pneumoniae*-driven NLRP3 activation[21,22,58], led us to test whether a similar inflammasome axis and associated cell death mechanisms operate in the human AD retina. To this end, we first applied a quantitative IHC analysis to retinal cross-sections from patients with MCI due to AD and AD dementia, compared with matched non-AD individuals with normal cognition (Fig. 4A–I; extended data in Supplementary Fig. 10; $n = 25–27$

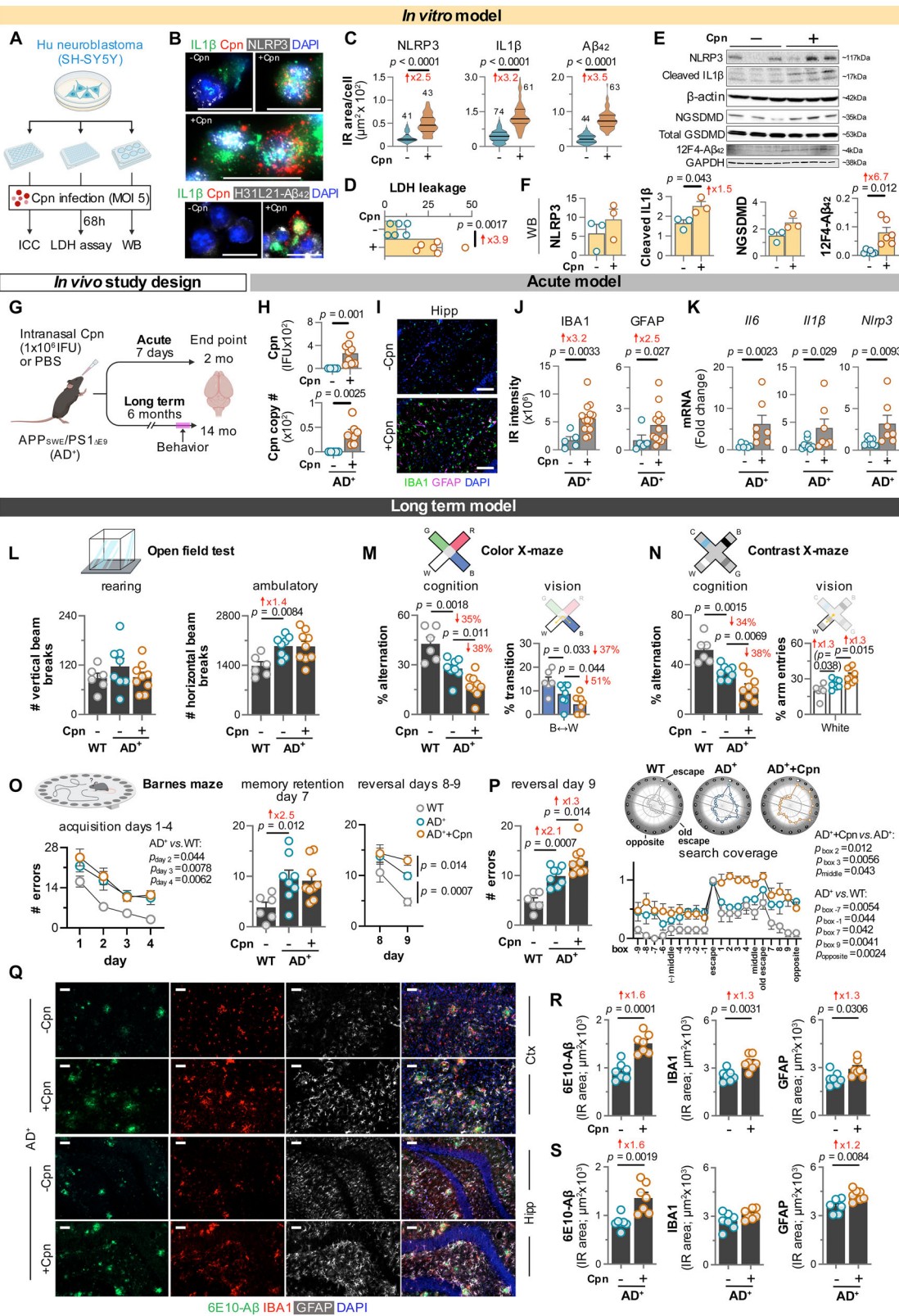

donors). Retinal NLRP3 expression was significantly elevated in MCI and further increased in AD compared with normal-cognition controls (2.1- and 3.6-fold, respectively; $p < 0.001$–$0.0001$), with strong colocalization with caspase-1, which itself was upregulated 2.5-fold in AD but not MCI. This inflammasome-activation signature was accompanied by a 3.1-fold increase in *Chlamydia pneumoniae*–associated ASC speck signals in AD, but not in MCI (Fig. 4A–E; extended data in

Supplementary Fig. 10A, B). The early rise in NLRP3 immunoreactivity and later induction of caspase-1 and ASC markers may suggest that NLRP3 is activated by earlier processes such as misfolded Aβ and tau accumulation in the retina.

Next, we investigated the impact of retinal *Chlamydia pneumoniae*-mediated NLRP3 inflammasome activation on key apoptotic and pyroptotic components. *Chlamydia pneumoniae*–infected cells in AD

**Fig. 3 | Effects of *Chlamydia pneumoniae* infection in SH-SY5Y cells and AD+ mice. A** Schematic of Cpn infection in SH-SY5Y cells. **B** Immunofluorescence of SH-SY5Y cells ± Cpn showing IL1β (green), Cpn (red), NLRP3 or Aβ$_{42}$ (H31L21, white). **C** Immunoreactive area quantification in SH-SY5Y cells for NLRP3 (−Cpn = 41, +Cpn = 43), IL1β (−Cpn = 74, +Cpn = 61), and Aβ$_{42}$ (−Cpn=44, +Cpn=63). **D** Quantification of LDH release in SH-SY5Y cells culture medium ± Cpn (*n* = 6 wells/group). **E, F** Western blot (WB) gels and densitometric analysis of NLRP3 (*n* = 3/group), IL1β (*n* = 3/group), NGSDMD (*n* = 3/group), and 12F4-Aβ$_{42}$ (*n* = 6/group). **G** Schematic of acute and long-term intranasal Cpn infection (1 × 10$^6$ inclusion-forming units, IFU) in APP$_{SWE}$/PS1$_{ΔE9}$ (AD+) mice. **H** Cerebral Cpn load at 7 days post-infection, quantified by live Cpn growth ((IFU; −Cpn=5, +Cpn=9) and by qPCR (Cpn DNA copy number; −Cpn=5, +Cpn=7). **I** Immunofluorescence of mouse hippocampi ± Cpn infection, showing IBA1 (microglia, green) and GFAP (astrocytes, magenta). **J** Quantification of IBA1 and GFAP immunoreactivity in mouse hippocampi (−Cpn=5, +Cpn=14). **K** Cerebral mRNA expression levels of *Il6* (−Cpn=6, +Cpn=7), *Il1β* (−Cpn=8, +Cpn=7, and *Nlrp3* (−Cpn=8, +Cpn=7) in AD+ mice. **L−P** Behavioral tests in AD+ mice 6-month Cpn post-infection or PBS administration (WT = 6, AD+-Cp=8, AD++Cp=9). **L** Locomotor activity in open field test. **M, N** Percent alternations and transitions in color-mode or contrast-mode X-maze. **O, P** Number of errors at 1–4-day acquisition phase, day-7 memory retention, the 8–9-day reversal phase, and day-9 search coverage, in Barnes maze. **Q** Immunofluorescence of cortical and hippocampal Aβ (6E10, green), IBA1 (red), and GFAP (white). **R, S** Quantification of cortical and hippocampal 6E10, IBA1, and GFAP immunoreactive area (*n* = 7/group). Scale bars, 50 μm (**I**) and 20 μm (**Q**). Individual data points and group mean ± SEMs are shown. Violin plots display median, upper, and lower quartiles. Fold or percent changes are in red. *p* values by one-way ANOVA and Tukey's or Fisher's LSD post-hoc test (**L−P**), two-sided unpaired *t*- test (**C, D, F, R, S**), or Mann–Whitney U test (**H, J, K**). Illustrations **A** and **G** created in Biorender.com. Fuchs, D. (2026) https://BioRender.com/lj8g4yb and https://BioRender.com/k77gi93. Source data are provided as a Source Data file.

retinas frequently co-expressed the pyroptotic effector NGSDMD and the early apoptotic marker cleaved caspase-3 (CCasp3) (Fig. 4F–I; extended data in Supplementary Fig. 10C, D). Quantitative IHC analysis revealed significant 2.2- and 3.0-fold increases in retinal NGSDMD and CCasp3 signals, respectively, in AD versus normal-cognition controls (*p* < 0.001–0.0001), whereas MCI retinas exhibited a nonsignificant trend toward elevation (Fig. 4G, I). Most *Chlamydia pneumoniae*-positive cells coexpressed either pyroptotic or apoptotic markers, suggesting that retinal *Chlamydia pneumoniae* infection in AD engages both cell death pathways. Moreover, the marked increase in NGSDMD in AD retinas provides functional evidence of NLRP3 inflammasome-mediated pyroptotic activation.

The inter-relationships between retinal *Chlamydia pneumoniae* burden, NLRP3 inflammasome components, and cell death markers were subsequently evaluated (Fig. 4J; extended data in Supplementary Table 12). Multivariate correlation analysis in our cohort revealed that retinal *Chlamydia pneumoniae* burden was strongly to very strongly associated with retinal NLRP3 inflammasome components, particularly caspase-1 (*r* = 0.87, *p* < 0.0001), NLRP3 (*r* = 0.70, *p* < 0.0001), and ASC (*r* = 0.60, *p* = 0.0012) (Fig. 4J). In addition to their correlations with retinal *Chlamydia pneumoniae* load (Supplementary Table 8), both retinal Aβ$_{42}$ and T22+ oligomeric tau were strongly to very strongly associated with NLRP3 and caspase-1 (*r* = 0.60–0.81, *p* < 0.01–0.0001; Supplementary Table 12), supporting their role as potential activators of the retinal NLRP3 inflammasome. Retinal oligomeric tau was strongly correlated with both apoptotic and pyroptotic markers, including CCasp3 (*r* = 0.80, *p* < 0.0001) and NGSDMD (*r* = 0.77, *p* < 0.0001), and retinal Aβ$_{42}$ likewise showed strong associations with CCasp3 (*r* = 0.77, *p* = 0.0003) and NGSDMD (*r* = 0.64, *p* = 0.0247; Supplementary Table 12). Retinal NLRP3 inflammasome components were significantly inter-correlated (*r* = 0.58–0.83, *p* = 0.0016–*p* < 0.0001; Fig. 4J), with NLRP3 and caspase-1 most tightly linked. All three components were strongly to very strongly associated with CCasp3 (*r* = 0.76–0.81, *p* < 0.0001), whereas NGSDMD pyroptosis was most closely related to NLRP3 (*r* = 0.74, *p* < 0.0001) and only moderately correlated with caspase-1 and ASC (*r* = 0.57–0.58, *p* < 0.01; Fig. 4J and Supplementary Table 12). To assess the canonical downstream effector of NLRP3 inflammasome activation, we quantified the pro-inflammatory cytokine IL1β by Western blot in retinal homogenates from AD patients and normal-cognition controls. Consistent with inflammasome engagement, AD retinas exhibited a marked shift from pro–IL1β to its active cleaved form, with significantly reduced pro–IL1β by 49.8% and markedly elevated mature IL1β by 2.1 folds (Fig. 4K; *p* < 0.01–0.001, *n* = 10 donors). Together, these observations delineate a *Chlamydia pneumoniae*-, Aβ$_{42}$-, and oligomeric tau-linked retinal NLRP3 inflammasome axis that converges on IL1β maturation and apoptotic/pyroptotic cell death in AD (summarized in Fig. 4L).

We next examined how retinal *Chlamydia pneumoniae*-related NLRP3 inflammasome activation components relate to retinal atrophy

and a spectrum of cerebral AD pathological indices and clinical outcomes (Fig. 4M; extended data in Supplementary Tables 12 and 13). Retinal NLRP3 and caspase-1 were very strongly and positively correlated with retinal atrophy severity, quantified as a thinning index from the inner limiting membrane to the outer limiting membrane (*r* = 0.82–0.86, *p* < 0.001–0.0001), and showed moderate positive correlations with global brain atrophy scores (*r* = 0.41–0.47, *p* < 0.05). Retinal ASC was strongly associated with retinal atrophy (*r* = 0.62, *p* = 0.0171), but not with brain atrophy. Retinal *Chlamydia pneumoniae* burden exhibited a strong association with retinal atrophy (*r* = 0.75, *p* < 0.0001) and a moderate correlation with brain atrophy (*r* = 0.48, *p* < 0.001). Retinal CCasp3 showed moderate to very strong correlations with both brain and retinal atrophy (Fig. 4M; *r* = 0.45–0.86, *p* < 0.05–0.001), whereas NGSDMD was strongly correlated with retinal atrophy (*r* = 0.60, *p* = 0.0247) but not with brain atrophy. Similar to retinal *Chlamydia pneumoniae*, all retinal inflammasome components and related cell death markers correlated moderately to strongly with Braak stage (*r* = 0.55–0.72, *p* < 0.01–0.0001) and inversely with MMSE cognitive performance (Fig. 4M; *r* = −0.49 to −0.69, *p* < 0.05–0.001). Together, these relationships position retinal *Chlamydia pneumoniae* burden and NLRP3 inflammasome activation as integrated indices of local neurodegeneration and global AD severity. They support a model in which pathogen-linked retinal inflammasome signaling tracks with, and may contribute to, parallel brain atrophy and cognitive decline.

## Gliosis surrounds retinal *Chlamydia* as phagocytosis declines in AD

*Chlamydia pneumoniae* infects and extensively replicates in astrocytes and neurons, whereas microglia are primarily involved in *Chlamydia pneumoniae* phagocytosis[59]. Although previous studies, including our own, have documented robust retinal gliosis in MCI and AD retinas[33,44], the potential interplay between glial cells and *Chlamydia pneumoniae* in these patients' retinas remains unexplored. Here, we observed prominent spatial interactions between *Chlamydia pneumoniae* inclusions and retinal microglia, astrocytes, and Müller glia in MCI and AD, with glial cells frequently surrounding or internalizing bacteria-positive inclusions (Fig. 5A–J; extended data in Supplementary Fig. 11A, B; *n* = 21–32 donors). Quantitative IHC analyses revealed significantly increased GFAP+ astrocyte and IBA1+ microglial cell counts in the AD retina (1.5-fold, *p* < 0.05–0.001) but not in the MCI retina (Fig. 5B, H). Interestingly, analysis of the IR area of GFAP+ and vimentin+ macroglia, as well as IBA1+ microglia, thereby accounting for cell morphology and process hypertrophy, showed significant expansion of all gliosis markers in MCI retinas (1.5–1.9-folds, *p* < 0.05–0.01), with further marked expansion observed in AD retinas (2.3–2.8-folds, *p* < 0.0001) relative to normal-cognition controls (Fig. 5C, E, I; extended data in Supplementary Fig. 11C–E). These findings suggest that glial activation and process hypertrophy emerge early along the retinal AD continuum, followed by overt glial proliferation in established AD dementia.

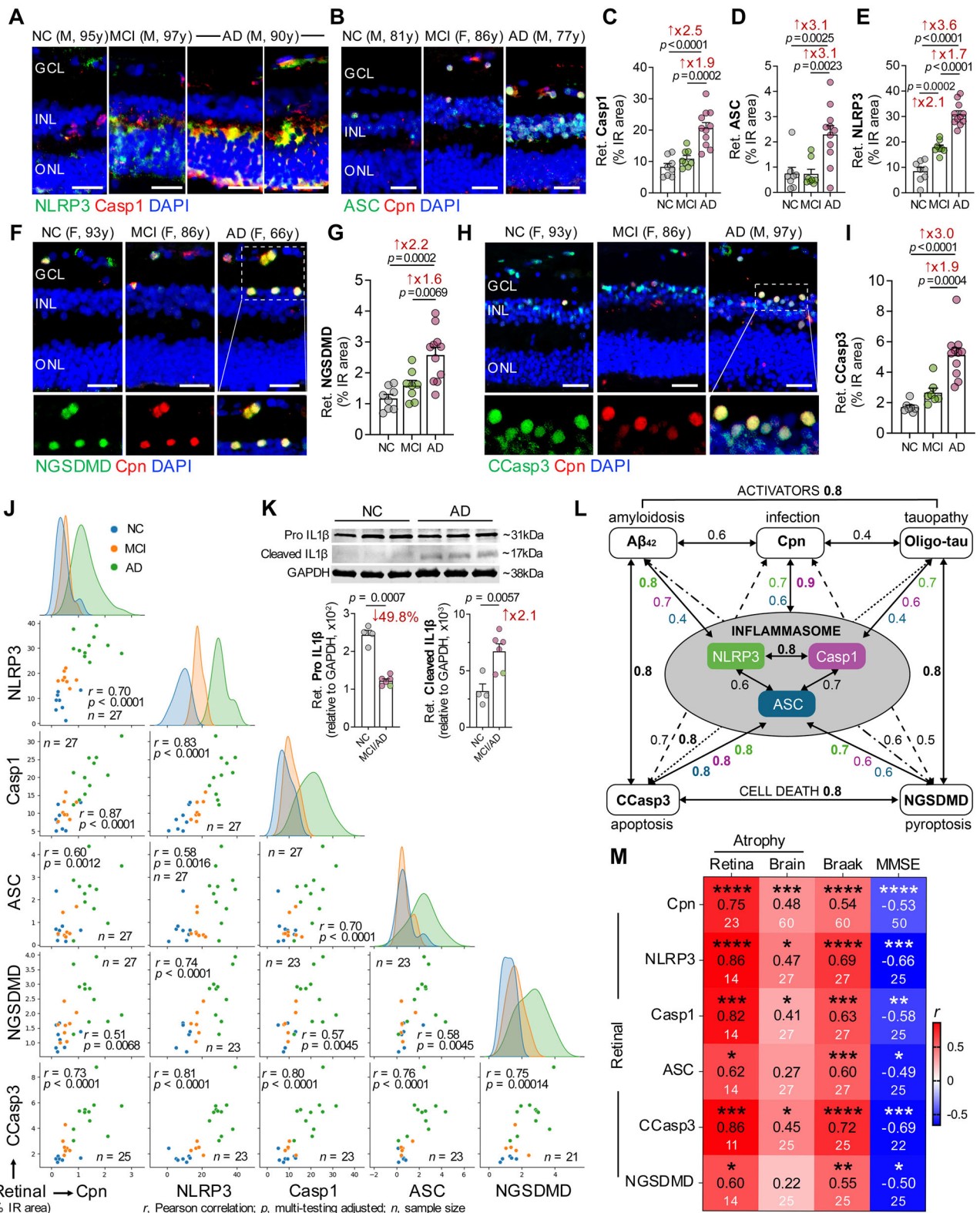

We next examined whether retinal bacterial load scales with gliosis burden. Retinal *Chlamydia pneumoniae* burden showed strong associations with GFAP⁺ astrocytosis and IBA1⁺ microgliosis (Fig. 5D, J; $r = 0.65$–$0.70$, $p < 0.0001$) and a moderate association with vimentin⁺ macroglial reactivity (Fig. 5F; $r = 0.55$, $p = 0.0090$). Consistent with these retinal interactions, strong correlations were detected between brain *Chlamydia pneumoniae* burden and brain gliosis ($r = 0.77$, $p = 0.0008$), as mentioned above. The strong association between

*Chlamydia pneumoniae* burden and gliosis indicates potential chronic inflammatory cascades linked to bacterial infection. Notably, while retinal astrocytes appeared to be infected by *Chlamydia pneumoniae*, retinal microglia appeared to phagocytose bacterial inclusions or bacteria-infected cells (Fig. 5A, K). Specifically, retinal microglia appeared to exhibit different stages of responses to *Chlamydia pneumoniae*-infected cells, with most cells in close proximity and partial contact with *Chlamydia pneumoniae*-positive cells (recognition

**Fig. 4 | Retinal NLRP3 inflammasome, pyroptotic, and apoptotic markers and associations with *Chlamydia pneumoniae* infection in early and advanced AD.** **A**, **B** Immunofluorescence images of retinal cross-sections from AD and MCI donors compared with NC controls, showing NLRP3 inflammasome activation markers: **A** NLRP3 (green), Caspase-1 (Casp1, red), and nuclei (blue); **B** ASC (green), Cpn inclusions (red), and nuclei (blue). Quantification of retinal percentage immunoreactive (IR) area for **C** Casp1, **D** ASC, and **E** NLRP3 in 8 NC, 8 MCI, and 11 AD donors. **F** Representative images of retinal cross-sections stained for the pyroptotic marker N-terminal cleaved gasdermin D (NGSDMD, green), Cpn (red), and nuclei (blue). **G** Quantification of retinal NGSDMD %IR area in 8 NC, 8 MCI, and 11 AD donors. **H** Representative images of retinal cross-sections stained for early-apoptotic marker cleaved caspase-3 (CCasp3⁺, green), Cpn (red), and nuclei (blue). **I** Quantification of CCasp3 %IR area in a subset of donors with 7 NC, 7 MCI, and 11 AD. **J** Gaussian distribution curves and Pearson's correlation ($r_p$) analyses presented as scatter plots with adjusted $p$ values, showing relationships among retinal markers: Cpn, NLRP3, Casp1, ASC, CCasp3-apoptosis, and NGSDMD-pyroptosis. **K** Western blot gel images and quantification of pro- and mature IL1β in retinal homogenates with band intensity normalized to GAPDH. [(4 NC, 1 MCI (green dot) and 5 AD (magenta dots)]. **L** Schematic summary of correlation strength ($r_p$) among three categories: (1) inflammasome activators (Cpn, Aβ₄₂, oligomeric tau); (2) active NLRP3-inflammasome markers (NLRP3, Casp1, ASC); and (3) cell-death markers (CCasp3, NGSDMD). **M** Heatmap showing pairwise Pearson's correlations ($r_p$) between retinal Cpn–related markers and retinal atrophy, and Spearman's correlations ($r_s$) with brain atrophy, Braak stage, and MMSE score. Stars indicate significance based on unadjusted $p$ values; middle-row numbers show $r_p$ or $r_s$, and lower-row values indicate sample sizes. All scale bars, 25 µm. Data are shown as individual values with group means ± SEMs. Fold increase or percent changes are marked in red. *$p < 0.05$, **$p < 0.01$, ***$p < 0.001$, ****$p < 0.0001$, by one-way ANOVA with Tukey's post hoc test (**C**, **D**, **E**, **G**, **I**), two-sided unpaired $t$-test (**K**), or pairwise Pearson's/Spearman's correlation analyses (**M**). Source data are provided as a Source Data file.

phase). Other microglia were directly involved in engulfing or ingesting *Chlamydia pneumoniae*-infected cells (Fig. 5K). The percentage of *Chlamydia pneumoniae*-associated microglia (CAM) cell count was increased by 50% in the AD retina, but not in the MCI retina, compared with normal-cognition controls (Fig. 5L; 1.5-fold, $p < 0.01$, $n = 44$ donors). However, the ratio of retinal CAM cell count to bacterial load was reduced by 61% in the AD retina versus normal-cognition controls (Fig. 5M; $p < 0.0001$, $n = 44$; extended data on microglia recognizing, engulfing or phagocytosing *Chlamydia pneumoniae*-infected cells in Supplementary Fig. 12A–C), implying defective microglial phagocytosis of *Chlamydia pneumoniae* in the AD retina. To further verify that *Chlamydia pneumoniae*–associated IBA1⁺ cells were resident microglia rather than perivascular or infiltrating macrophages, we co-labeled retinal sections with transmembrane protein 119 (TMEM119), a marker enriched in resident microglia in the human brain and retina[60,61] (Fig. 5N, $n = 12$; extended images in Supplementary Fig. 12D). All observed IBA1⁺ cells engaged in recognition, engulfment, or ingestion of *Chlamydia pneumoniae* inclusions co-expressed TMEM119, indicating a microglial identity. We refer to these cells as CAM. The percentage of retinal CAM cells relative to bacterial load was strongly and inversely correlated with retinal *Chlamydia pneumoniae* burden (Fig. 5O; $r = -0.72$, $p < 0.0001$), with an even stronger correlation among individuals with normal cognition (Supplementary Fig. 12E; $r = -0.79$, $p = 0.0004$). Within the normal-cognition group, three female individuals exhibited a disproportionately low percentage of CAM relative to retinal bacterial burden, comparable to the pattern observed in AD (Fig. 5M and Supplementary Fig. 12E). Notably, these individuals also showed the highest retinal *Chlamydia pneumoniae* loads (Fig. 1F), with comparable levels of retinal and brain AD-related pathology, suggesting a selective impairment of microglial engagement with bacteria-infected cells even in some clinically normal individuals. Together, these data indicate that, despite increased recruitment of CAM in the AD retina, their phagocytic engagement relative to bacterial burden is markedly reduced, consistent with impaired microglial clearance of retinal *Chlamydia pneumoniae* in AD.

A multi-interaction heatmap between retinal gliosis and various AD biomarkers in the retina and brain (Fig. 5P; extended data in Supplementary Tables 12 and 13) revealed very strong associations between retinal GFAP⁺ or vimentin⁺ macrogliosis and NLRP3 load ($r = 0.84$–$0.91$, $p < 0.01$–$0.0001$), and between retinal GFAP⁺ astrogliosis and CCasp3⁺ apoptosis ($r = 0.85$, $p < 0.0001$). As it relates to amyloidosis and tauopathy, retinal Aβ₄₂ and oligo-tau burdens most closely correlated with retinal IBA1⁺ microgliosis levels (Fig. 5P; $r = 0.69$–$0.85$, $p < 0.0001$). In addition, retinal GFAP⁺ strongly correlated with Braak scores ($r = 0.78$, $p < 0.0001$). These findings suggest close interactions between retinal glial cells and *Chlamydia pneumoniae*-infected cells, strongly correlating with NLRP3 inflammasome components and apoptosis/pyroptosis cell death markers, and a

potentially impaired ability of microglia to phagocytose and clear *Chlamydia pneumoniae* infection in the AD retina.

## Retinal *Chlamydia pneumoniae* and NLRP3 predict AD diagnosis and stage

We next tested whether retinal *Chlamydia pneumoniae* burden and associated markers could serve as predictors of AD diagnosis, cerebral pathology severity, disease stage, and cognitive dysfunction (Fig. 6; extended data in Supplementary Figs. 13–15). In addition to retinal *Chlamydia pneumoniae*, we included key retinal markers that demonstrated significant correlations with infection and AD pathology, specifically NLRP3, CCasp3-apoptosis, and Aβ₄₂. These markers were analyzed either in isolation or in combination with retinal gliosis (IBA1+GFAP+vimentin), atrophy, and Aβ₄₂; the latter can potentially be imaged in living patients[30,32,37,62,63]. We evaluated 80 biomarker-derived estimators for both regression and classification, using a diagnosis-stratified split of 56 donors (80%) for model training and 14 donors (20%) for testing. Multivariable analysis employing random forest machine learning models indicated that retinal *Chlamydia pneumoniae* alone weakly predicted AD-related brain pathologies, including ABC score and Braak stage (Fig. 6A, B), as well as cognitive function (MMSE, Fig. 6C; MOCA, Supplementary Fig. 13B). Multiple models were compared using 5×2 cross validation to obtain distributions of model performance. The predictive power of retinal *Chlamydia pneumoniae* was generally enhanced when combined with retinal Aβ₄₂ or gliosis (Fig. 6A–C and Supplementary Fig. 13A–E). We found that retinal Aβ₄₂ alone was a good predictor of brain NFTs, ABC score, Braak stage, and the MMSE score (Fig. 6A–C and Supplementary Fig. 13A). Notably, the combined retinal *Chlamydia pneumoniae* and gliosis index was the best predictor of ABC score (Fig. 6A, $r^2 = 0.34$) and brain gliosis (Supplementary Fig. 13E, $r^2 = 0.26$). The best predictor of Braak stage was retinal Aβ₄₂ combined with CCasp3 ($r^2 = 0.41$), followed by combinations with NLRP3 ($r^2 = 0.38$), and *Chlamydia pneumoniae* ($r^2 = 0.28$; Fig. 6B). In addition, the combined retinal NLRP3 and Aβ₄₂ index was the best predictor of MMSE score (Fig. 6C, $r^2 = 0.25$), and retinal Aβ₄₂ combined with either *Chlamydia pneumoniae* or CCasp3 also predicted MMSE (Fig. 6C, $r^2 = 0.22$–$0.23$), while no individual marker predicted brain atrophy (Supplementary Fig. 13D).

We further evaluated the performance of these variables using the area under the receiver operating characteristic (ROC) curve (AUC) for disease diagnosis (Fig. 6D–F and Supplementary Fig. 14A–D; separate analyses for each diagnostic group in Supplementary Fig. 15A–D). When combined with retinal amyloidopathy (Aβ₄₂), the AUC for retinal *Chlamydia pneumoniae* increased from 0.80 to 0.94 (Fig. 6D, E), with particular gains in distinguishing MCI from normal-cognition diagnoses (Fig. 6D). These findings indicate that retinal *Chlamydia pneumoniae* in combination with Aβ₄₂ may constitute a highly informative marker pair for discriminating disease status (Fig. 6F). Retinal Aβ₄₂

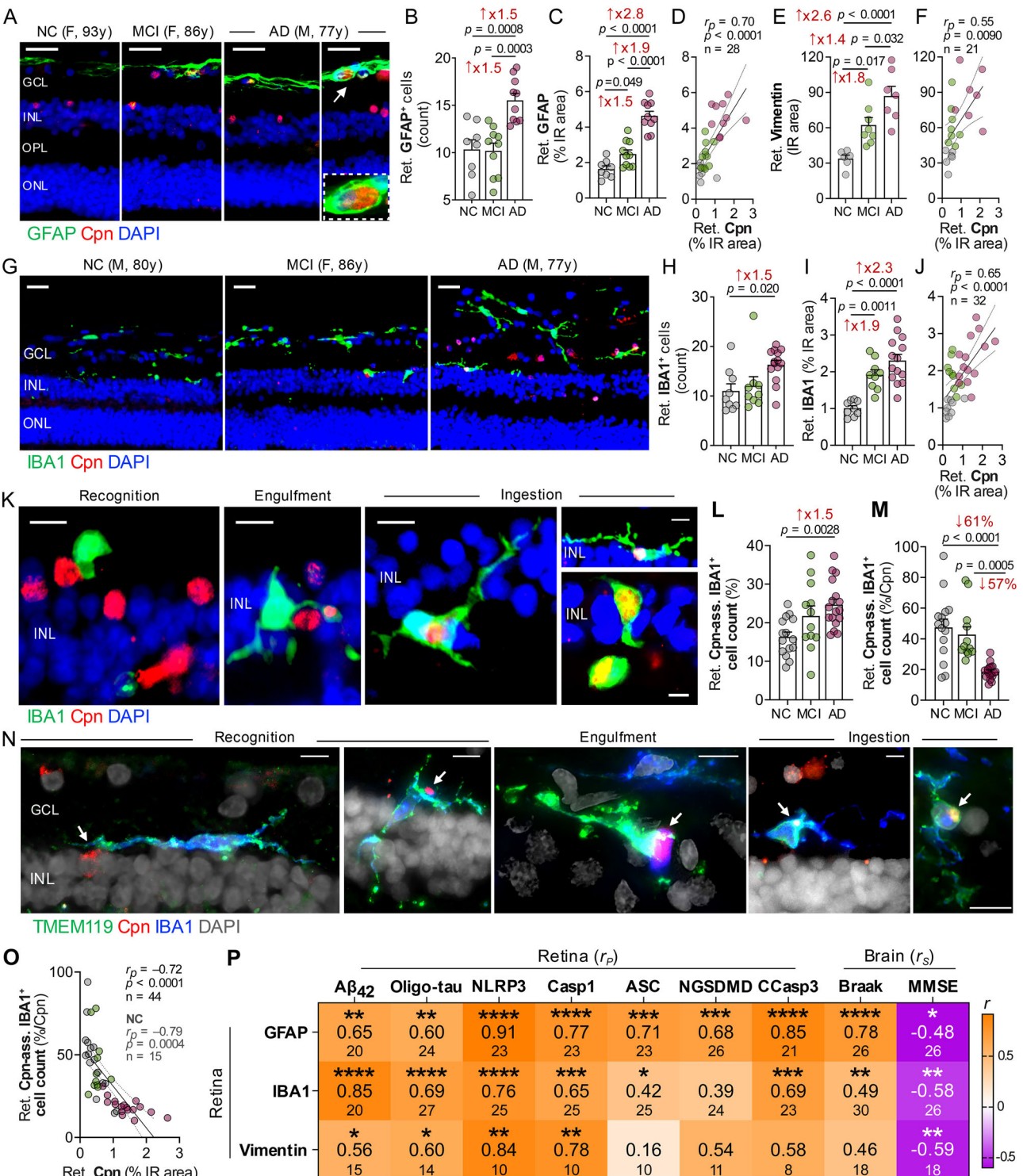

alone (AUC = 0.87) showed its best predictive performance for classifying normal-cognition subjects (AUC = 0.92; Supplementary Fig. 15D). Consistent with the random forest model results, retinal NLRP3, CCasp3, and atrophy exhibited strong AUC values for disease diagnosis, with notable further improvements when combined with $A\beta_{42}$ (0.92, 0.87, and 0.89, respectively; Fig. 6E and Supplementary Fig. 14B–D), including within individual diagnostic groups (Supplementary Fig. 15A–C). Based on the results presented in Fig. 6E, F, we selected the model using *Chlamydia pneumoniae* and retinal $A\beta_{42}$ for evaluation on the test set, consisting of 3/4/7 patients with normal cognition/MCI/AD diagnoses, respectively; the results are reported in

Table 2. The model performed poorly for subjects with MCI but performed reasonably well for identifying normal cognition and AD. We note that the normal cognition and MCI groups were under-represented in the test set.

We further examined the selected model's performance by comparing prediction probabilities with a clinical disease severity (scores 0–3), based on premortem CDR and MMSE cognitive performance. Using the test set, we obtained the prediction probability for each diagnosis and compared these against disease severity (Supplementary Fig. 15E). There are high correlations between the prediction probabilities of both normal cognition ($r = -0.78$) and AD ($r = 0.79$)

**Fig. 5 | *Chlamydia pneumoniae*-associated glial activation and phagocytosis in MCI and AD retina. A** Representative immunofluorescence images of retinal cross-sections from AD and MCI donors versus normal cognition (NC) controls stained for macrogliosis using GFAP (green), Cpn (mAb, red), and nuclei (blue). The high-magnification image show Cpn inclusions within an astrocyte. **B, C** Quantification of retinal GFAP⁺ cell count and percentage immunoreactive (IR) area in 8 NC, 10 MCI, and 10 AD donors. **D** Pearson's correlation ($r_p$) between retinal Cpn and GFAP %IR area. **E** Quantification of retinal vimentin IR area in 6 NC, 8 MCI, and 7 AD donors. **F** Pearson's correlation ($r_p$) between retinal Cpn %IR area and vimentin IR area in the same cohort. **G** Immunofluorescence images of retinas from NC, MCI, and AD donors stained for microglial marker IBA1 (green), Cpn (red), and DAPI (blue). **H, I** Quantification of retinal IBA1⁺ cell counts and %IR area in 9 NC, 9 MCI, and 14 AD donors. **J** Pearson's correlation between retinal Cpn and IBA1 %IR area. **K** Representative images showing three stages of microglial involvement in phagocytosis of Cpn-infected cells: recognition, engulfment, and ingestion.

**L, M** Quantification of Cpn-associated IBA1⁺ cells (%) and retinal Cpn-associated IBA1⁺ cells relative to Cpn load (%/Cpn) in 15 NC, 12 MCI, and 17 AD donors. **N** Immunofluorescence images showing involvement of TMEM119⁺IBA1⁺-microglia in Cpn phagocytosis ($n = 8$ donors, 3 repetitions): TMEM119 (green), Cpn (red), IBA1 (blue), and nuclei (grey). **O** Pearson's correlation analysis between retinal Cpn load and Cpn-associating microglia. **P** Heatmap illustrating Pearson's correlations between retinal gliosis and retinal Aβ₄₂, oligo-tau, inflammasome, and cell death markers. Spearman's correlations ($r_s$) of retinal gliosis with Braak stage and MMSE score. Stars denote significance (unadjusted $p$ values), middle-row numbers show $r_p$ or $r_s$, and lower-row values indicate sample sizes. Scale bars, 25 μm (**A, G**), 10 μm (**K, N**). Data are shown as individual values with group means ± SEM. Fold or % changes are shown in red. *$p < 0.05$, **$p < 0.01$, ***$p < 0.001$, ****$p < 0.0001$, by one-way ANOVA with Tukey's post hoc test (**B, C, E, H, I, L, M**). Source data are provided as a Source Data file.

against disease severity. No correlation was observed between MCI prediction and disease severity, as expected. Looking at the prediction probabilities for each diagnosis, we see that the standard deviation across normal cognition, MCI, and AD is 0.41, 0.22, and 0.34, respectively. Highly certain predictions would result in higher standard deviations; predictions for MCI patients are less certain, indicating the possibility that MCI could be detectable with additional training samples.

## Discussion

This study identifies *Chlamydia pneumoniae* inclusions in the human retina and positions pathogen-driven inflammatory dysregulation as a potential amplifier of AD pathogenesis in both retina and brain. Using multiple complementary approaches, including a bacterial-specific monoclonal antibody, in-situ hybridization, and *argR* qPCR, we confirm the presence of *Chlamydia pneumoniae* in the retina and show that its burden is markedly increased in AD. Retinal *Chlamydia pneumoniae* load tightly tracks with matched cortical load and is enriched in APOE ε4 carriers and patients with higher Braak stage and worse cognitive status, thereby linking a retinal bacterial signature to widespread AD neuropathology and dementia severity. Retinal *Chlamydia pneumoniae* burden strongly associates with retinal amyloidogenic Aβ₄₂ and Aβ₄₀ species, while moderately associating with selective tau isoforms. MS-based proteomics in independent retinal and cortical cohorts reveal a dysregulated *Chlamydia* interactome enriched for gram-negative bacterial infection, NLRP3 inflammasome components, pyroptosis executors, and cell-death pathways. Our functional data show that *Chlamydia pneumoniae* infection is sufficient to induce NLRP3 activation, IL1β maturation, neurotoxicity, and Aβ₄₂ accumulation in human neurons and to exacerbate neuroinflammation, Aβ plaque burden, and visuocognitive impairment in AD⁺ mice. In human MCI and AD retinas, we further delineate a *Chlamydia pneumoniae*–linked NLRP3 inflammasome axis that converges on IL1β processing, apoptosis, and pyroptosis, and retinal and brain atrophy. This inflammasome–degeneration axis is embedded within a gliotic milieu in which astrocytes and Müller glia surround *Chlamydia pneumoniae* inclusions, while microglia, despite increased recruitment, exhibit impaired phagocytic engagement with bacteria-infected cells. Finally, machine-learning models integrating retinal *Chlamydia pneumoniae* with Aβ₄₂, NLRP3, apoptosis, gliosis, and atrophy demonstrate that multimodal retinal signatures can robustly predict AD diagnosis, neuropathological stage, and cognitive status, highlighting infection- and inflammasome-linked retinal biomarkers as promising, image-accessible indicators of AD-related disease trajectories. Overall, these findings support a model in which *Chlamydia pneumoniae* acts as a disease amplifier in vulnerable retina–brain circuits and highlight pathogen- and NLRP3-targeted strategies, including timely antibiotic and inflammasome-modulating interventions, as testable avenues to modify AD progression.

In the current study, we reveal the presence and elevated burden of retinal and brain *Chlamydia pneumoniae* inclusions in AD dementia compared with normal-cognition controls, and a strong concordance between these two CNS tissues. These findings extend prior reports of increased *Chlamydia pneumoniae* in the CSF and brains of AD patients[10–15,64] and are consistent with epidemiological studies linking *Chlamydia pneumoniae* infection to chronic inflammation and dementia risk[7,10,12–14], showing a five-fold increased risk of AD in the presence of this pathogen[14]. *Chlamydia pneumoniae* infection is highly prevalent and often lifelong in humans, with antibody incidence in peripheral blood reaching 50% by age 20 and 80% by age 60–70 years[65], and the organism is capable of forming persistent aberrant bodies under immune or antibiotic pressure[66]. In this study, we confirmed the presence of *Chlamydia pneumoniae* in the human retina, particularly in both somatic and perinuclear compartments of cells within the GCL and INL, using complementary approaches including monoclonal and polyclonal immunolabeling, Giemsa staining, bacterial DNA–specific in situ hybridization, and *argR*-targeted qPCR. *Chlamydia pneumoniae* is distinct from other *Chlamydia* species, particularly *Chlamydia trachomatis*, which is the leading cause of infectious blindness and the most common sexually transmitted urogenital infection[67]. Both species can cause chronic, low-grade infections, suggesting a broader link between *Chlamydia* and chronic inflammation, including in CNS disorders. *Chlamydia pneumoniae* is thought to enter the CNS via nasal and intravascular routes through monocytes, with supporting evidence including detection of its DNA in the olfactory bulb of AD patients and accelerated Aβ plaque development in *Chlamydia pneumoniae*-inoculated wild type mice[18,19]. Given the retina's direct anatomical connection to the brain and the parallel development of AD pathology in both tissues[31–41,44], the presence of *Chlamydia pneumoniae* infection in the AD retina is therefore not unexpected. These findings reinforce the parallel susceptibility of the AD brain and retina to *Chlamydia pneumoniae* infection and underscore its potential role in exacerbating AD pathology and cognitive decline.

Retinal *Chlamydia pneumoniae* strongly correlates with retinal amyloidogenic Aβ alloforms (Aβ₄₂, Aβ₄₀) and modestly, but selectively, with tau species, including PHF-tau and tau oligomers, but not AT8⁺ p-tau, MC1⁺ tangles or multiple isoforms of p-tau/total tau ratios. These relationships are compatible with the hypothesis that Aβ may function as an antimicrobial peptide[68], with *Chlamydia pneumoniae*–driven retinal infection promoting Aβ deposition as a maladaptive host-defense response, as reflected in infected neuroblastoma cultures and AD⁺ mouse brains in this study. Retinal and brain *Chlamydia pneumoniae* burdens further track with brain NFTs, NTs, Aβ plaques, CAA, ABC score, Braak stage, gliosis, atrophy, and poorer MMSE, CDR, and MOCA scores, implicating this pathogen in AD progression rather than isolated ocular infection. The higher *Chlamydia pneumoniae* burden in APOE ε4 carriers is consistent with the

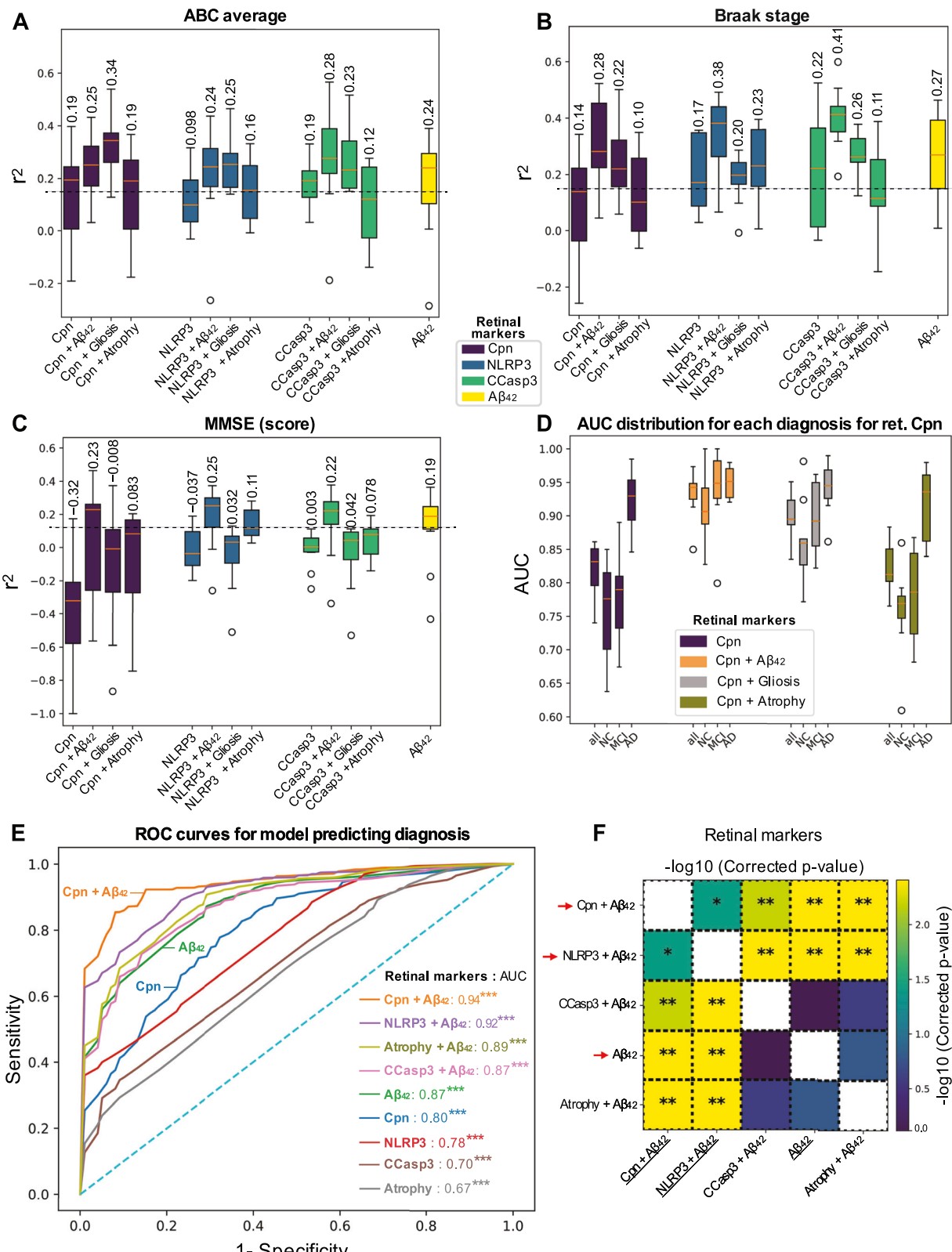

requirement of intracellular lipids for *Chlamydia pneumoniae* growth[69] and with APOE ε4-driven lipid dysregulation[70], suggesting a gene–infection interaction that may modulate CNS susceptibility. Although *Chlamydia pneumoniae* is detectable in a subset of aged cognitively normal controls and MCI patients, and group-level load is not significantly elevated in MCI, its near-universal presence in AD dementia, together with early vascular dysfunction in both brain and retina[30,36,41] and potential nasal entry routes[18], argues that *Chlamydia pneumoniae* is more likely a disease amplifier within vulnerable circuits than a universal initiating trigger of AD.

Our mechanistic data support a model in which *Chlamydia pneumoniae*, like other gram-negative pathogens, activates NLRP3 inflammasomes[71] and thereby couples infection to neuroinflammation and neurodegeneration. In human neuroblastoma cells, *Chlamydia*

**Fig. 6 | Multivariable predictions of brain AD pathology and cognitive dysfunction conferred by retinal *Chlamydia pneumoniae*, NLRP3, CCasp3, and Aβ₄₂ markers.** Random forest regressor using 80 estimators was trained on the data to predict several brain pathologies. Box plots show the spread of results in quartiles with the same number of samples; the mean is displayed above each box. including **A** ABC average ($n = 24/25$ for train/validation), **B** Braak stage ($n = 24/25$ for train/validation), and **C** mini-mental state examination (MMSE) score ($n = 20/20$ for train/validation). The distributions show the spread of models trained on different folds of the 5-repeated 2-fold cross-validation. Only models performing with a variance coefficient of determination $r^2 > 0.15$ (gray dotted line) were retained. **D** Box plots representing the AUC measure for retinal Cpn for each diagnostic groups NC, MCI, and AD. For each model, AUC was measured ($n = 28/28$ for train/validation) using features either individually or combined with retinal Aβ₄₂, retinal gliosis (IBA1, GFAP, and Vimentin), or retinal atrophy. **E** The ROC curves for different retinal biomarkers, including *Chlamydia pneumoniae* (Cpn), Aβ₄₂, NLRP3, CCasp3, and retinal atrophy, either individual or combined with retinal Aβ₄₂. Each

model was obtained by averaging the curves across diagnosis separately in each cross-validation fold. In the ROC curves plot, AUC is listed for each curve and unadjusted. *P*-values for whether the results were different from baseline dummy models using permutation tests with $k = 10,000$ iterations (estimated ***$p < 0.001$). **F** The models were compared by using a Wilcoxon signed-rank test, and *p* values were adjusted for multiple comparisons using Benjamini-Hochberg correction. The heat map shows that among the top 5 performing models, we have 3 that are different from one another. The model trained on retinal Cpn + retinal Aβ₄₂ performed best and was significantly different from the second-best model (retinal NLRP3 + retinal Aβ₄₂) with $p < 0.05$. The other three models were different from the top 2, but not from one another. Red arrows highlight retinal markers, individually or in combination, which were significantly different among the performance models to predict disease diagnosis. Statistics: *$p < 0.05$ and **$p < 0.01$, adjusted for multiple comparisons with Benjamini-Hochberg procedure. Source data are provided as a Source Data file.

## Table 2 | Model performance on the machine learning test set

| | Precision | Recall | F1-Score | Support |
|---|---|---|---|---|
| AD | 0.83 | 0.71 | 0.77 | 7 |
| MCI | 0.50 | 0.33 | 0.40 | 3 |
| Normal cognition | 0.67 | 1.00 | 0.80 | 4 |
| Accuracy | | | 0.71 | 14 |
| Macro average | 0.67 | 0.68 | 0.66 | 14 |
| Weighted average | 0.71 | 0.71 | 0.70 | 14 |

The final test set consisting of 14 samples with a distribution of diagnoses matching the training set was used to evaluate the random forest model using *Chlamydia pneumoniae* and retinal Aβ₄₂ to predict AD diagnosis.

*pneumoniae* infection is sufficient to increase NLRP3, IL1β, NGSDMD, and Aβ₄₂, and to induce LDH release, recapitulating key cellular features of AD pathology. In AD⁺ mice, intranasal *Chlamydia pneumoniae* inoculation establishes brain infection and is accompanied by upregulation of *Nlrp3*, *Il1β*, and *Il6* transcripts, amplified IBA1⁺ microgliosis and GFAP⁺ astrogliosis, and increased cortical and hippocampal Aβ plaques. Moreover, infected AD⁺ mice show worsened visuocognitive performance in X-maze and Barnes maze tasks, consistent with infection-driven aggravation of neuroinflammation, amyloidosis, and cognitive decline. These results demonstrate that long-term *Chlamydia pneumoniae* infection exacerbates brain pathology and behavioral dysfunction in a mouse model of AD. In human MCI and AD retinas, we identify a *Chlamydia pneumoniae*-Aβ₄₂-linked NLRP3 inflammasome axis: NLRP3, ASC, and caspase-1 correlate strongly with retinal *Chlamydia pneumoniae* load and Aβ₄₂, and associate with CCasp3 apoptosis, NGSDMD pyroptosis, retinal and brain atrophy, higher Braak stage, and worse MMSE.

Notably, similar to early increases in Aβ and tauopathy, NLRP3 is already elevated in MCI, whereas caspase-1, ASC, NGSDMD, and cleaved IL1β are most prominent in AD dementia stages, aligning with a two-step model of inflammasome activation[72] in which early misfolded Aβ/tau may prime NLRP3 and *Chlamydia pneumoniae*-induced cell damage further acts as a secondary signal to fully engage caspase-1–dependent cytokine processing and pyroptosis. Our findings of increased mature IL1β in the AD retina further support the active status of the retinal NLRP3 inflammasome in AD. In this framework, priming (signal 1) is triggered by inflammatory stimuli, such as those recognized by TLRs or cytokines like tumor necrosis factor (TNF)-α, leading to NF-κB activation and subsequent upregulation of NLRP3 and pro-IL1β, whereas activation (signal 2) is driven by diverse pathogen-associated molecular patterns (PAMPs) and damage-associated molecular patterns (DAMPs) that promote NLRP3 inflammasome assembly, caspase-1 activation, and cleavage of pro-IL1β and pro-IL18 to their

mature forms[72] (reviewed in ref. [73]). Within this two-signal paradigm, *Chlamydia pneumoniae* can plausibly function as the secondary activating cue that fully engages the NLRP3 inflammasome, thereby further potentiating inflammation and neurodegeneration in the AD retina, a hypothesis that warrants direct mechanistic testing.

Proteomic reanalysis further substantiates this infection–inflammasome–degeneration axis. We identify enrichment of proteins previously implicated in immune responses to intracellular Gram-negative infection and *Chlamydia*-related interactions. These include components of TLR–MYD88 signaling, NLRP3 inflammasome activation, pyroptosis (GSDMA, GSDMD, GSDME), and apoptosis (Casp3, Fas-Associated Death Domain, FADD) in AD retinas and brains. Approximately 13% of the 607 cortical and 710 retinal proteins in the reported *Chlamydia* interactome[48–52] are dysregulated in each tissue. A shared subset [Reticulon-4 (RTN4), endoplasmic reticulum stress (TECR), dolichyl-diphosphooligosaccharide-protein glycosyltransferase subunit STT3B, transmembrane emp24 domain-containing protein 4 (TMED4), AP2M1, heat shock protein β−1 (HSPB1), tropomyosin α−3 chain (TPM3), BAG3, LRRFIP1, ATPase H⁺ transporting V1 subunit G1(ATP6V1G1)] is altered in both AD retina and brain. Many of these proteins regulate apoptosis and mitochondrial function (RTN4, HSPB1, BAG3), inflammatory and synaptic plasticity pathways, TECR, and Aβ/tau handling (LRRFIP1, HSPB1, BAG3)[55,56,74–77], and several (e.g., LRRFIP1) interface directly with TLR4–MYD88 signaling[55]. Their coordinated dysregulation in AD retina and cortex supports a conserved host-response profile to *Chlamydia pneumoniae* that intersects with core AD pathways, although mechanistic dissection will require future studies.

We also uncover a complex glial response to *Chlamydia pneumoniae* in the AD retina. Retinal GFAP⁺ and vimentin⁺ macrogliosis and IBA1⁺ microgliosis findings suggest that glial activation and process hypertrophy emerge early along the AD continuum (in MCI due to AD), followed by overt glial proliferation in established AD dementia. Retinal GFAP⁺ astrocytosis and IBA1⁺ microgliosis tightly correlate with retinal and brain *Chlamydia pneumoniae* burden, retinal NLRP3 inflammasome components, CCasp3-defined apoptosis, Aβ₄₂, and oligomeric tau, and are further associated with higher Braak stage and lower MMSE scores. Retinal IBA1⁺TMEM119⁺ resident microglia frequently surround or internalize *Chlamydia pneumoniae* inclusions, yet the proportion of CAM relative to bacterial burden is reduced by ~61% in AD compared with controls and inversely correlated to bacterial load, implying impaired microglial recognition and/or phagocytosis of infected cells. A similar dysfunctional pattern was evident in three female clinically normal individuals with disproportionately high retinal *Chlamydia pneumoniae* burden but low CAM-to-bacterial load ratios, despite exhibiting retinal and cerebral AD-related pathology comparable to that seen in normal-cognition donors. Together with the strong inverse relationship between CAM-to-bacterial percentages

and bacterial burden, these findings suggest that failure of microglial containment of *Chlamydia pneumoniae*–infected cells can emerge even before overt cognitive decline. We propose that in these individuals, microglia may remain relatively competent in managing Aβ clearance yet be selectively impaired in recognizing or clearing *Chlamydia pneumoniae*, allowing pathogen burden to escalate. In this framework, convergence of high *Chlamydia pneumoniae* load, elevated Aβ, and sustained inflammasome activation may represent a critical transition point linking chronic infection to progressive neurodegeneration and the eventual emergence of clinical symptoms.

Finally, machine-learning models incorporating retinal *Chlamydia pneumoniae*, NLRP3, and CCasp3, alone and in combination with Aβ$_{42}$, gliosis, and atrophy, demonstrate that integrating infection- and inflammasome-related signatures with canonical AD markers improves prediction of AD diagnosis, disease stage, brain gliosis, and cognitive dysfunction. Among these features, retinal *Chlamydia pneumoniae* in combination with retinal Aβ$_{42}$ provides the strongest predictive power, achieving AUCs of up to 0.94. Further, retinal *Chlamydia pneumoniae* is highly informative for discriminating AD status and severity, and although performance for MCI classification is limited, likely reflecting small sample size, the reduced prediction certainty in MCI suggests that with larger datasets and additional features, infection- and inflammasome-linked retinal markers could help resolve intermediate disease states. Importantly, models were not retrained after test-set evaluation to avoid data leakage, and the present results do not exclude the possibility that MCI will become reliably identifiable as additional data and features are incorporated.

Despite the breadth of our data, several limitations warrant caution. First, the relatively small number of brain samples and the absence of an MCI group in the proteomic MS cohorts constrain generalizability and preclude full staging of the *Chlamydia* interactome; larger, prospectively collected series will be essential to validate and refine these associations. Second, the comprehensive histological and biochemical evidence of infection and immune activation remains correlative, and targeted functional studies are needed to dissect how specific *Chlamydia* interactome proteins and inflammasome components mechanistically couple infection to retinal and brain neurodegeneration and dysfunction. Finally, the human data are cross-sectional. Thus, although our neuronal and AD⁺ mouse experiments strongly support a mechanistic link between *Chlamydia pneumoniae*, NLRP3 activation, AD-like pathology, and cognitive decline, they do not establish causality in patients, a question best addressed by longitudinal and interventional clinical studies. Notably, a large Taiwanese nationwide cohort found that patients with *Chlamydia pneumoniae*-type pneumonia who received appropriate macrolide (e.g., azithromycin) or fluoroquinolone therapy had a lower subsequent risk of AD dementia versus inadequately treated cases[17], further supporting a causal contribution of *Chlamydia pneumoniae* to AD risk.

In conclusion, this study identifies and spatially characterizes *Chlamydia pneumoniae* in the retinas of MCI and AD patients and links bacterial burden to retinal NLRP3 inflammasome activation, glial impairment, and neurodegeneration, as well as to corresponding brain pathology and clinical status. Together with convergent evidence from human neuronal cultures and AD⁺ mouse models, our findings position *Chlamydia pneumoniae* as an amplifier of AD pathology and dysfunction and highlight pathogen- and NLRP3-targeted strategies, including timely antibiotic and inflammasome-modulating interventions, as testable routes to modify disease progression and accelerate development of accessible retinal imaging biomarkers of infection-linked AD burden.

## Methods

### Human eye and brain samples

Postmortem human eye globes and brain tissues were obtained from the Alzheimer's Disease Research Center (ADRC) Neuropathology Core at the Department of Pathology at the University of Southern California (USC, Los Angeles, CA; Institutional Review Board (IRB) protocol HS-042071). In addition, eye globes were obtained from the National Disease Research Interchange (NDRI, Philadelphia, PA; under Cedars-Sinai IRB protocol Pro00019393) and the Rush Alzheimer's Disease Center (RADC) at Rush University (Chicago, IL; ORA# 18011111). For a subset of patients and controls, brain specimens were also obtained from the ADRC Neuropathology Core at the University of California, Irvine (UCI IRB protocol HS#2014–1526). USC-ADRC, NDRI, and UCI ADRC maintain human tissue collection protocols that are approved by their respective oversight committees and subject to the National Institutes of Health and institutional guidelines. All histological procedures were conducted at Cedars-Sinai Medical Center under IRB protocols (Pro00053412, Pro00019393, and Pro00055802). For histological examinations, 69 retinas were collected from deceased donors with confirmed AD ($n = 34$) or MCI (mild cognitive impairment) due to AD ($n = 14$), and from age- and sex-matched deceased donors with normal cognition ($n = 21$). In a subset of patients, paired brain tissues were also analyzed ($n = 16$). For Giemsa staining we analyzed a subset of 4 AD, 2 MCI, and 2 normal-cognition cases ($n = 8$). For in situ hybridization, we analyzed a subset of 5 AD, 3 MCI, and 3 normal-cognition controls ($n = 11$). For analysis of retinal proteins using MS, ELISA, and Western blot (WB), fresh-frozen retinas were collected from another cohort of deceased donors ($n = 17$) with clinically and neuropathologically confirmed MCI/AD patients ($n = 10$) and matched controls with normal cognition ($n = 7$). For brain protein MS analyses, fresh-frozen human brain tissue was obtained from an additional donor cohort ($n = 18$) consisting of clinically and neuropathologically confirmed AD patients ($n = 10$) and matched normal-cognition controls ($n = 8$). For *Chlamydia pneumoniae* qPCR confirmation, fresh-frozen retinal samples were analyzed from 2 AD, 1 MCI, and 2 normal-cognition controls ($n = 5$). Comprehensive cohort details are presented in Table 1 and Supplementary Tables 1–5. The human cohort used in this study exhibited no significant differences in age, sex, or post-mortem interval (PMI) hours. Patient confidentiality was maintained by de-identifying all tissue samples, ensuring that donors could not be traced.

### Clinical and neuropathological assessments

The detailed clinical and neuropathological assessment procedures are described in our recent publication[33,40]. In summary, clinical and neuropathological reports detailing patients' neurological examinations and neuropsychological and cognitive assessments were generously provided by the ADRC system using the Unified Data Set[78]. NDRI provided patient information including sex, ethnicity, age at death, cause of death, medical background indicating AD, the presence or absence of dementia, and accompanying medical conditions. Most cognitive assessments were conducted annually, typically within one year prior to death. In this study, we utilized cognitive scores assessed closest to the patient's death. Three global indicators of cognitive status were used for clinical assessment: the CDR scores (0 = normal; 0.5 = very mild impairment; 1 = mild dementia; 2 = moderate dementia; or 3 = severe dementia)[79], MOCA scores (≥26 = cognitively normal or <26 = cognitively impaired)[80], and the MMSE scores (normal cognition = 24–30; MCI = 20–23; moderate dementia = 10–19; or severe dementia ≤9)[81]. The assessment of cerebral Aβ burden comprised the analysis of diffuse and neuritic plaques (including both immature and mature forms), along with amyloid angiopathy, NFTs, NTs, granulovacuolar degeneration, Lewy bodies, Hirano bodies, Pick bodies, balloon cells, neuronal loss, microvascular changes, and gliosis. These evaluations were conducted across different brain regions, notably in the hippocampus, entorhinal cortex, superior frontal gyrus in the frontal lobe, the superior temporal gyrus in the temporal lobe, the superior parietal lobule in the parietal lobe, the primary visual cortex, and the visual

association area in the occipital lobe. All brain samples were uniformly collected by a neuropathologist.

Formalin-fixed, paraffin-embedded brain sections were used to determine the severity of amyloid plaques and NFTs using anti Aβ mAb clone 4G8, anti phospho-tau mAb clone AT8, Thioflavin-S (ThioS), and Gallyas silver staining. Two neuropathologists independently rated the burden of Aβ, NFTs, and NTs on a scale of 0, 1, 3, and 5 [(0 = none, 1 = sparse (0–5), 3 = moderate (6–20), 5 = abundant/frequent (21–30 or above), n.a. = not applicable)], and final scores were averaged. The final diagnosis included AD neuropathologic change. The Aβ plaque scoring system was adapted from Thal et al. (A0 = no Aβ or amyloid plaques, A1 = Thal phase 1 or 2, A2 = Thal phase 3, and A3 = Thal phase 4 or 5)[82]. NFT staging was adjusted from Braak for silver-based histochemistry or p-tau immunohistochemistry (B0 = No NFTs, B1 = Braak stage I or II, B2 = Braak stage III or IV, B3 = Braak stage V or VI)[83]. The neuritic plaque score was adapted from CERAD (C0 = no neuritic plaques, C1 = CERAD sparse, C2 = CERAD moderate, C3 = CERAD frequent)[84]. Additional evaluations included neuronal loss, gliosis, granulovacuolar degeneration, Hirano bodies, Lewy bodies, Pick bodies, and balloon cells using hematoxylin and eosin staining, with scores of 0 for absent and 1 for present. Amyloid angiopathy was classified into 4 grades: Grade I indicates amyloid around normal/atrophic smooth muscle cells of vessels; Grade II shows media replaced by amyloid without blood leakage; Grade III involves extensive amyloid deposition with vessel wall fragmentation and perivascular leakage; Grade IV includes extensive amyloid deposition with fibrinoid necrosis, microaneurysms, mural thrombi, lumen inflammation, and perivascular neuritis.

### Collection and processing of eyes and brain cortical tissues

Donor eyes were collected and preserved within an average of 10 h postmortem. These eyes were either preserved in Optisol-GS media (Bausch & Lomb, 50006-OPT), snap frozen upon delivery and stored at −80 °C, or punctured once at the limbus and fixed in 10% neutral buffered formalin or 4% paraformaldehyde (PFA) and stored at 4 °C. Brain tissues (hippocampus, Brodmann Area 9 of the prefrontal cortex, and Brodmann Area 17 of the primary visual cortex) were collected from the same donors, snap frozen, and stored at −80 °C. For MS, fresh-frozen human brain tissues (hippocampus, medial temporal gyrus, and cerebellum) were obtained from an additional donor cohort. The same tissue collection and processing procedures were applied consistently, regardless of the source institution for the donor eyes and brains.

### Preparation of retinal and brain cross-sections

Fresh eyes preserved in Optisol-GS and fixed eyes were dissected on ice by removing the anterior segments to form eyecups. The vitreous humor was thoroughly removed manually. Retinas were then carefully dissected, detached from the choroid, and prepared as flatmounts following established procedures[32,33,40]. Geometric regions of the four retinal quadrants were defined for both left and right eyes by identifying the macula, optic disc, and blood vessels. Flatmount-strips, measuring 2–3 mm in width from fixed retinas and 5 mm in width from fresh retinas, spanning from the ora serrata to the optic disc (OD), were dissected along the margins of these quadrants to create four strips: superior-temporal (ST), inferior-temporal (TI), inferior-nasal (IN), and superior-nasal (NS). Flatmount ST strips (~2–3 mm thick) from fixed retinas were initially embedded in paraffin, then rotated 90° horizontally and re-embedded in paraffin blocks. Retinal strips approximately 2–2.5 cm in length, encompassing central (C), mid (M), and far (F) retinal subregions, were sectioned into 8–10-μm thick slices and mounted on microscope slides treated with 3-aminopropyltriethoxysilane (APES, Sigma A3648). Flatmount strips from fresh-frozen retinas were stored at −80 °C for MS protein analysis. Fresh-frozen paired brain tissues from Area 9 (located in the

dorsolateral prefrontal cortex) were fixed in 4% PFA for 16 h at 4 °C, following paraffin embedding. They were sectioned (10-μm thick) and mounted on microscope slides treated with APES. This sample preparation technique allowed for extensive and consistent access to retinal quadrants, layers, and pathological subregions.

### IHC analysis

Paraffin-embedded cross-section slides were first deparaffinized with 100% xylene twice (10 min each), rehydrated with decreasing concentrations of ethanol (100% to 70%), and washed with distilled water followed by PBS. Following deparaffinization, retinal and brain cross-sections were treated with antigen retrieval solution (pH 6.1; S1699, DAKO) at 99 °C for 1 h, washed with PBS, and then treated with 70% formic acid (ACROS) for 10 min at room temperature (RT).

For peroxidase-based labeling, we used a Vectastain Elite Avidin–Biotin–Complex with Horseradish Peroxidase kit [Vector Laboratories Inc., Peroxidase Mouse IgG (PK-6102) and Peroxidase Rabbit IgG (PK-6101)] according to the manufacturer's instructions. In summary, following incubation with 3% $H_2O_2$ for 20 min, tissues were washed with PBS and incubated with blocking serum containing 0.25% Triton X-100 (Sigma, T8787) for 45 min at RT. Primary antibodies against *Chlamydia pneumoniae* (Supplementary Table 6) were diluted in PBS containing blocking serum and incubated overnight at 4 °C. The following day, tissues were rinsed three times with PBS, incubated for 30 min at 37 °C with secondary antibody, rinsed three times with PBS, and incubated with Avidin–Biotin–Complex reagent for 30 min at RT. After washing with PBS, *Chlamydia pneumoniae* inclusions in brain or retinal tissue sections were visualized with 3,3′-diaminobenzidine substrate (DAKO K3468). Hematoxylin counterstaining was performed, followed by mounting with Paramount aqueous mounting medium (DAKO, S3025). Routine controls were processed using identical protocols while omitting the primary Ab to assess nonspecific labeling.

For fluorescence-based immunostaining, sections were first treated with a blocking solution (DAKO X0909) containing 0.25% Triton X-100 (Sigma, T8787), followed by overnight incubation with primary antibodies (Supplementary Table 6) at 4 °C. The next day, following washing with PBS, sections were incubated with fluorophore conjugated secondary antibodies (Supplementary Table 6) for 1 h at RT. Tissue sections were mounted using Fluoromount-G™ with DAPI (Thermo Fisher, # 00-4959-52). Control sections processed without primary antibodies were used to assess nonspecific labeling. To minimize background autofluorescence, brain sections were treated with 1x True Black (Biotium, #23007), diluted in 70% ethanol (v/v) for 40 s at RT before the application of primary antibodies.

### Microscopy and quantitative immunohistochemistry

Fluorescence and brightfield images were acquired using a Carl Zeiss Axio Imager Z1 fluorescence microscope (with motorized Z-drive) equipped with ApoTome, AxioCam MRm, and AxioCam HRc cameras (at a resolution of 1388 × 1040 pixels, 6.45 μm × 6.45 μm pixel size, and dynamic range of >1:2200, delivering low-noise images due to a Peltier-cooled sensor) with ZEN 2.6 blue edition software (Carl Zeiss MicroImaging, Inc.). Multi-channel image acquisition was used to generate images containing multiple channels. Images were repeatedly captured at the same focal planes with identical exposure times for each marker and human donor. Images were captured at 20×, 40× (at respective resolutions of 0.5 and 0.25 μm), 63×, and 100× objectives for different applications. For selected representative imaging, Z-stack images were repeatedly captured at the same tissue thickness. To cover the ST retinal strip, we consistently acquired 3 images from the central (C), 4 images from the mid-peripheral (M), and 3 images from the far-peripheral subregions, for analytical purposes[33]. Thickness measurements (μm) were manually performed using Axiovision Rel. 4.8 software. Retinal thickness assessments were taken from the ILM

through the OLM. Images were exported to NIH ImageJ2/Fiji (v2.14.0) to analyze parameters of interest. For each biomarker (e.g., *Chlamydia pneumoniae*, NLRP3, caspase-1, ASC, NGSDMD, vimentin), single-channel 20× images (covering ~$150 \times 10^3 \, \mu m^2$ area/per image) were converted to grayscale and standardized to baseline intensity by using a histogram-based threshold in ImageJ2/Fiji. This baseline-derived threshold was then applied uniformly to the corresponding single channel for all subjects across diagnostic groups, and ImageJ2/Fiji particle analysis was used to quantify, for each biomarker, total IR area and percentage IR area. For cell counts, the same fields of view used for IR area quantification in the 20× images were analyzed, and cells co-expressing DAPI with either *Chlamydia pneumoniae*, GFAP, or IBA1 were manually enumerated using the grid-overlay mode in Adobe Photoshop or ZEN software tools. Throughout the analysis process, researchers were blinded to each patient's diagnosis.

## Quantification of *Chlamydia pneumoniae*-associated microglia

Three distinct stages of retinal IBA1$^+$ microglial involvement in the phagocytosis of *Chlamydia pneumoniae/Chlamydia pneumoniae*-infected cells were analyzed from all acquired images (≥10 images/retinal section per donor), as described above. The recognition stage was defined as the count of IBA1$^+$ microglia in direct contact with *Chlamydia pneumoniae/Chlamydia pneumoniae*-infected cells (<50% of cell circumference). The engulfment stage was determined as the count of IBA1$^+$ microglia whose processes surrounded ≥50% of the *Chlamydia pneumoniae/Chlamydia pneumoniae*-infected cells. The ingestion stage refers to the count of IBA1$^+$ microglia in which *Chlamydia pneumoniae/Chlamydia pneumoniae*-infected cells were fully internalized, as evidenced by co-localization of *Chlamydia pneumoniae* (red) and IBA1 (green), appearing yellow in merged images. Microglial cells participating in any of these three stages were classified as CAM. The relative contribution of microglia to *Chlamydia pneumoniae/Chlamydia pneumoniae*-infected cell recognition and uptake was quantified by calculating the proportion of CAM cells relative to the total IBA1$^+$ microglial population.

## Determination of A$\beta_{1-42}$ levels by sandwich ELISA in human retina

Fresh-frozen human retinal tissues from the temporal hemisphere were homogenized (1 mg tissue/10 μl buffer) in cold homogenizing buffer (100 mM tetraethylammonium bromide [Sigma, 241059], 1% sodium deoxycholate [SDC; Sigma, D6750], and 1x Protease Inhibitor cocktail set I [Calbiochem, 539,131]). Retinal homogenates were sonicated, and the amount of retinal A$\beta_{1-42}$ was determined using an anti-human A$\beta_{1-42}$ end-specific sandwich ELISA kit (Thermo Fisher, KHB3441) and normalized to total protein concentration (Thermo Fisher Scientific)[33]. In this study, we reanalyzed these data, using retinal A$\beta_{1-42}$ values to perform correlations with newly identified differentially expressed proteins.

## NanoString GeoMx Spatial Profiling of total tau and p-tau reanalysis

We performed a secondary analysis of NanoString GeoMx® Digital Spatial Profiling of total tau and 5 different p-tau (S214; T231; S199; S396; S404) species in formalin-fixed paraffin embedded human retinal cross-sections from 8 AD, 6 MCI, and 8 normal-cognition control cases (*n* = 22)[40]. In this study, we used the values of total tau and p-tau to perform the correlation study with retinal *Chlamydia pneumoniae* burden, with p-tau/total tau ratio as a secondary analysis.

## Proteome mass spectrometry (MS) reanalysis

In this study, we performed a secondary analysis on retinal and brain mass spectrometry data[33], to identify DEPs associated with gram-negative bacterial infection and *Chlamydia* interactomes. In brief, this analysis included the following: (1) preparation of retinal and brain homogenates; (2) tandem mass tag labeling; (3) nanoflow liquid chromatography electrospray ionization tandem mass spectrometry; (4) database searching, peptide quantification, and statistical analysis. DEPs are defined as |FC| ≥1.2 and unadjusted (*p*) < 0.05 by two-sided *t*-test; FDR-adjusted (adj. *p*) values are also shown in Supplementary Tables 10 and 11.

**Functional Network and Computational Analysis.** GO analysis of DEPs was performed in Metascape (https://metascape.org/; cutoffs: overlap ≥3, *p*-value < 0.01, enrichment ≥1.5) and included the GO Biological Processes, Reactome, Kyoto Encyclopedia of Genes and Genomes (KEGG), and WikiPathways databases. Enrichment analysis results are reported with z-scores, unadjusted *p*-values, and Benjamini-Hochberg adjusted *p*-values to control the FDR. Networks of pathways related to infection, neuroinflammation, immune response, and cell death were created in Metascape and subsequently loaded and modified in Cytoscape 3.10.2 (https://cytoscape.org/). Protein interaction networks were generated in String v12.0 and modified in Cytoscape. Volcano plots representing expression changes [log$_2$(FC)] and significance level [-log$_{10}$(*p*)] in AD versus normal cognition were created using Prism 10.3.1 (GraphPad) and included human proteins linked to *Chlamydia* inclusion membrane. The list of human proteins interacting with *Chlamydia* inclusions (termed as *Chlamydia* interactome) was determined from four original studies and a meta-analysis study[48-52]. Heatmaps corresponding to the protein expression level of select proteins in the retina of the 6 normal cognition individuals and the 6 AD patients, standardized by unit variance scaling, were generated in ClustVis (https://biit.cs.ut.ee/clustvis/). Chord diagrams representing associations of DEPs with select functional pathways were created in Circos online (https://mk.bcgsc.ca/tableviewer/). The original mass spectrometry proteomics data have been deposited in the ProteomeXchange Consortium via the PRIDE partner repository under the dataset identifier PXD040225.

## Giemsa staining

FFPE human retinal and *Chlamydia pneumoniae*–infected or non-infected mouse lung tissue sections were deparaffinized and subjected to antigen retrieval as described above. Slides were then immersed in a freshly prepared Giemsa working solution consisting of a 1:20 dilution of Giemsa stock solution (Catalog No. 48900-100ML-F, Sigma-Aldrich) in Sorensen's phosphate buffer [0.067 M each of monobasic potassium phosphate (KH$_2$PO$_4$) and dibasic sodium phosphate anhydrous (Na$_2$HPO$_4$), pH 6.8] for 15 min at RT. Following staining, the slides were gently rinsed with Sorensen's phosphate buffer, dipped in distilled water to remove excess dye, air-dried, and mounted using DAKO Far-amount aqueous mounting medium (Catalog #S3025). Stained sections were examined immediately under a light microscope to visualize bacterial localization.

## FISH assay

FFPE retinal tissues were deparaffinized and subjected to antigen retrieval as described above. Sections were then digested with proteinase K (10 μg/mL) for 7 min at 37 °C, followed by blocking [0.3 g bovine serum albumin, 10 μl Tween-20, 2 ml 20x Saline-Sodium Citrate buffer (SSC buffer containing: 175.3 g sodium chloride, 88.2 g Sodium citrate, in 1 liter of dH$_2$0, pH − 7)] buffer to make 10 mL with dH$_2$0) for 2 hours at RT. For hybridization, tissues were denatured at 80 °C for 10 min in hybridization buffer (for 20 mL: 10 mL of formamide, 4 mL of 50% dextran sulfate, 2 mL of 20x SSC, and 4 mL of nuclease-free water) containing a *Chlamydia pneumoniae*-specific nucleotide probe (Cpn1046, 5′-AGCCAATCACTCCAACGTGCATACTCGACC-3′, 0.2 μM) labeled at the 5′ end with 6-FAM (Carboxyfluorescein, company, catalog). The probe targets the Cpn1046 gene encoding a putative aromatic amino acid hydroxylase, which is unique to the *Chlamydia pneumoniae* genome and absent in other chlamydial species. After

rapid cooling on ice, tissues were hybridized overnight with same probe mix at 42 °C in a humidified chamber, washed stringently with 1x SSC buffer (15 min for 3 times), quenched with True Black as described above, mounted with DAPI-containing Fluoromount, and imaged as described above.

## Western blot analysis on human retina

Fresh-frozen postmortem temporal hemiretinal tissues from human donors were homogenized in radioimmunoprecipitation assay (RIPA) buffer (0.5 M Tris-HCl, pH 7.4, 1.5 M NaCl, 2.5% deoxycholic acid, 10% NP-40, 10 mM EDTA, Millipore; 20–188) supplemented with protease and phosphatase inhibitors. Protein concentration was determined using the bicinchoninic acid protein assay kit (Pierce™). Lysates were cleared by brief centrifugation for 10 min at $8000 \times g$, normalized, and boiled at 95 °C after addition of 6X SDS loading dye. Equal amounts of protein (30 μg per sample) were separated on 4–20% precast poly-acrylamide gels (Bio-Rad, catalog #4561094) and transferred to poly-vinylidene difluoride membranes. Membranes were blocked with 2.5% bovine serum albumin in 1x TBS-T (Tris-buffered saline with 0.1% Tween-20) for 1 h at RT, followed by overnight incubation at 4 °C with the anti-IL1β primary antibody (Supplementary Table 6). After washing with 1x TBS-T, membranes were incubated with fluorescently labeled secondary antibodies, and bands were detected using the LI-COR Odyssey imaging system. Band intensities were quantified using Image Studio software (LI-COR), and relative protein expression levels were calculated by normalizing target protein signals to GAPDH.

## Genomic DNA synthesis and qPCR analysis

Genomic DNA (gDNA) was extracted from fresh frozen human retinal tissues (superior quadrant). Retinal tissues were lysed using a Bead Ruptor Elite Bead Mill Homogenizer (Omni International) with ZR Bashing Bead Lysis Tubes containing beads [0.1 & 0.5 mm (S6012-50, ZYMO)] for 3 cycles at 1.6 m/s, with a 5-minute rest between cycles, and gDNA was extracted using the ZymoBIOMICS DNA Miniprep Kit according to the manufacturer's protocol (D4300, ZYMO). Positive control samples were obtained from gDNA extracted from *Chlamydia pneumoniae* elementary bodies (CM-1), which was prepared by homogenization of 49 mg of uninfected mouse brain tissue, ex vivo injected with $2 \times 10^6$ IFU of *Chlamydia pneumoniae* titer. Nuclease free water was used as a negative control (blank). For qPCR, one specific primer/probe set was chosen for *Chlamydia pneumoniae*. For the Chlamydial *argR* gene, the primer and probe sequences were: Cpn-*argR* F: CGTGGTGCTCGTTATTCTTTACC, Cpn-*argR* R: TGGCGAATA-GAGAGCACCAA, and TaqMan probe Cpn-*argR*: FAM-CTTCAACAGA-GAAGACCACGACCCGTCA-ZEN/Iowa Black FQ. The working concentrations for primers and probe were 500 nM and 50 nM, respectively. Thermal cycling parameters were: 3 min at 95 °C for activation, 15 s at 95 °C for denaturing, and 1 min at 59 °C for annealing/extension, for a total of 45 cycles.

## Mice

Double-transgenic B6.Cg-Tg(APP_SWE/PS1_ΔE9)85Dbo/Mmjax (AD⁺) mice and age-matched non-transgenic WT C57BL/6J littermates were obtained from the Mutant Mouse Resource and Research Center (MMRRC) at The Jackson Laboratory (RRID:MMRRC_034832-JAX). All mice were housed under identical environmental conditions at the Cedars-Sinai Medical Center vivarium under standardized conditions: a 14 h light/10 h dark cycle, ambient temperature maintained at 74 °F (23 °C) ± 2 °F, and relative humidity at 30–70%, with ad libitum access to food and water and a maximum of five animals per cage. Prior to infection, mice were maintained in ventilated cages on Allentown IVC racks. Following infection or PBS treatment, mice were housed in ventilated cages on metro racks within negative-pressure cubicles. Both male and female mice were used for all experiments, with experimental groups balanced for age and genotype. All procedures were approved by the Institutional Animal Care and Use Committee of Cedars-Sinai Medical Center (Protocols: IACUC008314 and IACUC008475) and conducted in accordance with the NIH Guide for the Care and Use of Laboratory Animals and the ARRIVE guidelines.

## *Chlamydia pneumoniae* infection in AD⁺ mice

For *Chlamydia pneumoniae* infection, we used 2-months-old (acute infection model) and 8-months-old (long-term infection model of 6-months post-infection) AD⁺ mice. Mice were intranasally infected with $1 \times 10^6$ *Chlamydia pneumoniae* titer [+Cpn; strain CM-1 (VR-1360, ATCC)] along with uninfected (PBS-treated; -Cpn) controls. For the acute *Chlamydia pneumoniae* infection experiments, we studied 8 PBS-administered AD⁺ mice and 14 *Chlamydia pneumoniae*-infected AD⁺ mice (total of $n = 22$). For the long-term *Chlamydia pneumoniae* infection experiments, we studied 6 PBS-administered WT mice, 8 PBS-administered AD⁺ mice, and 9 *Chlamydia pneumoniae*-infected AD⁺ mice (total of $n = 23$). In the acute infection model, mice were sacrificed 7 days post-infection, and their brains were either fresh-frozen or fixed in 4% paraformaldehyde. Lung tissues were also harvested for histological analyses. In the long-term model, mice underwent cognitive assessments 6-months post-infection encompassing open field, Barnes maze, and visual stimuli X-maze tests, prior to sacrifice, and brain extraction for histological analyses.

## Behavioral tests in mice

**Open field test.** Spontaneous locomotor activity was assessed for 30 min using the Photobeam Activity System (www.sandiegoinstruments.com). Ambulatory and rearing activities were recorded as horizontal and vertical photobeam breaks, respectively.

**Barnes maze test.** (hippocampus-based spatial learning and memory test). Mice were first trained to locate an escape box in a 20-hole circular platform during three daily 4-min trials for 4 days (acquisition phase). Following a 2-day break, long-term memory retention of each mouse was evaluated on day 7. Spatial memory extinction and learning of a new escape location were assessed on days 8–9 (reversal phase). For each trial, we recorded the number of errors at each hole location relative to the target (−9 to +9, middle, escape, and opposite). Error counts were averaged across trials within each day for each mouse.

**Visual-stimuli X-maze test.** (visual-cognitive test). To assess visuo-cognitive behavior, mice were tested using our custom-made X-shaped maze in color and contrast modes[57,85]. In the color mode, the arms of the maze were illuminated with different color LED lights: red mono-chromatic light (wavelength, $\lambda = 628$ nm), green monochromatic light ($\lambda = 517$ nm), blue monochromatic light ($\lambda = 452$ nm), and white light ($\lambda 1 = 441$ nm and $\lambda 2 = 533$ nm). In the contrast mode, different grey-scale objects (black, white, grey, and clear objects) were placed in the middle of each arm of the maze, contrasting against the black walls and the white floors of the maze. For each mode, the mice were individually placed in the center of the X-maze and allowed to freely explore for 5 min. The sequences of arm entries were manually documented according to video recordings. From the sequences of arm entries, we determined several parameters, which assess multiple behavioral domains: total number of entries (locomotion/general activity), percentage of each arm entries and percentage of bidirectional transitions between arms (color vision/contrast sensitivity), and percentage of spontaneous alternations (cognition, vision). An alternation was defined as a sequential visit to all four arms without returning to a previously visited arm.

## Mouse brain collection, processing, and immunohistochemistry

After completion of behavioral testing, all mice were deeply anaes-thetized (50 mg/kg ketamine/xylazine) and transcardially perfused with ice-cold saline solution containing 0.5 mM EDTA. Mouse brains

were collected and either snap-frozen for biochemical analyses or fixed in 2.5% PFA overnight, followed by cryoprotection in 30% sucrose for histological analyses. Mouse brains were then coronally sectioned at 30 μm thickness using a cryostat (Leica CM3050 S; Leica Biosystems, Nussloch, Germany) and processed for immunohistochemical analyses. In brief, mouse brain sections were blocked with DAKO blocking solution, incubated with primary antibodies against IBA1, GFAP, and Aβ (clone 6E10; Supplementary Table 6) overnight at 4 °C, followed by corresponding secondary antibodies and mounting with using DAPI-containing medium, as described above. Images were acquired using a Carl Zeiss Axio Imager Z1 fluorescence microscope at 10× magnification to capture larger brain regions. For each mouse, 4-6 cortical and hippocampal 10× images were collected from each of 3 brain sections, covering approximately $9 \times 10^6 \, \mu m^3$ brain area. Image processing and quantification of IBA1-, GFAP-, and 6E10+Aβ-immunoreactive areas were performed for each single channel as grayscale using NIH ImageJ2/Fiji (v2.14.0), as described above.

### Real-time PCR analysis of mRNA levels in mouse brains

Total RNA was extracted from *Chlamydia pneumoniae*-infected or uninfected AD+ mouse brain tissues using TRIzol Reagent (Invitrogen; 15596018). One microgram of total RNA was reverse transcribed (RT) to synthesize complementary DNA (cDNA) by reverse transcription using the RevertAid RT kit (ThermoScientific; #K1691) containing 5X reverse transcription buffer, 10 mM dNTP mix, RiboLock RNase Inhibitor (20 U/μl), random hexamer primers, and reverse transcriptase. For reverse transcription, samples were mixed, centrifuged gently, and incubated for 5 min at 25 °C, followed by 60 min at 42 °C using a thermal cycler (Applied Biosystem). cDNAs were amplified using specific primers (Supplementary Table 7) and Power-Up-SYBR green (Applied Biosystems; A25742) on a CFX96 Real-Time System (BioRad). Expression levels of *Nlrp3*, *Il6*, and *Il1β* mRNAs were quantified using the relative threshold $^{\Delta\Delta}Ct$ method: $^{\Delta\Delta}Ct = (\text{primer efficiency})^{\wedge}(-^{\Delta\Delta}Ct)$, where $^{\Delta\Delta}Ct$ means ΔCt (target gene) −ΔCt (reference gene, GAPDH).

### *Chlamydia pneumoniae* IFU determination in mouse brain

HEp-2 cells were inoculated with mouse brain specimens infected with *Chlamydia pneumoniae* or with PBS to quantify bacterial burden, measured as *Chlamydia pneumoniae* IFU[86]. Specifically, HEp-2 cells were incubated with diluted brain suspensions in the presence of RPMI 1640 medium containing 1 μg/ml cycloheximide. Centrifugation was performed for 1 h at 800 × g, after which cells were placed in an incubator (37 °C, 0.5% CO2). After 72 hours of culture, cells were washed with PBS, fixed with methanol, and stained with FITC-conjugated anti-*Chlamydia* genus-specific mAb (Pathfinder *Chlamydia* Culture Confirmation System; Bio-Rad, Hercules, CA), according to the manufacturer's instructions. *Chlamydia pneumoniae* inclusions in cells were counted under a fluorescence microscope.

### *Chlamydia pneumoniae* infection in SH-SY5Y human neuroblastoma cell

SH-SY5Y human neuroblastoma cells (CRL-2266™, ATCC) were used to examine the effect of *Chlamydia pneumoniae* infection on the production of Aβ, inflammatory mediators, and neurotoxicity. Cells were grown and incubated at 37 °C in a humidified atmosphere of 5% CO2 and 95% air and maintained in Dulbecco's-modified Eagle's medium (Gibco, #11-320-033) supplemented with 10% fetal bovine serum, 1% penicillin-streptomycin (100 U/mL each, Gibco, #15140122), and 1% sodium pyruvate. When the cells reached approximately 80% confluence, they were infected with *Chlamydia pneumoniae* (CM-1 strain; American Type Culture Collection, Manassas, VA) at an MOI of 5 for 68 h, after which they were harvested for further analyses.

**Immunocytochemistry.** SH-SY5Y cells grown on gelatin-coated glass coverslips ($1 \times 10^5$ cells per well in 24-well plates, 6 wells per condition)

were post-fixed with 4% PFA and blocked with DAKO protein-block. *Chlamydia pneumoniae*-infected, or PBS-treated cells, were then incubated with primary antibodies against *Chlamydia pneumoniae*, Aβ42, NLRP3, and IL1β, overnight at 4 °C, followed by incubation with the respective secondary antibodies (Supplementary Table 6) and mounting. Images were acquired using a Carl Zeiss Axio Imager Z1 fluorescence microscope at 20× (for quantification) and 40× (for representative images) magnification. For each *Chlamydia pneumoniae*–infected or PBS-treated condition, 8–10 images at 20× magnification were collected for analysis (n = 41–74 cells per experimental group in duplicates). Image processing and quantification of H31L21+Aβ42-, NLRP3-, and IL1β-immunoreactive areas were performed for each single channel as grayscale using NIH ImageJ2/Fiji (v2.14.0), as described above.

**LDH cytotoxicity assay.** Culture media from *Chlamydia pneumoniae*-infected or uninfected SH-SY5Y cells were collected ($1 \times 10^4$ cells per well in 96-well plates; n = 6 wells per group), and LDH release amount was quantified using a cytotoxicity detection kit (Cayman Chemical, Ann Arbor, MI, #601170) as per manufacturer's protocol.

**Western blot analyses.** SH-SY5Y cells were lysed in RIPA buffer supplemented with protease and phosphatase inhibitors ($1.5 \times 10^6$ cells per well in 6-well plates; n = 3–6 wells per group). Protein concentration was determined using a bicinchoninic acid (BCA) assay (Pierce™). Equal amounts of protein (30 μg) were resolved on 4–20% precast polyacrylamide gels (Bio-Rad #4561094) and transferred to polyvinylidene fluoride membranes. Membranes were blocked with 2.5% bovine serum albumin and incubated overnight at 4 °C with primary antibodies (against NLRP3, IL1β, NGSDMD, 12F4+Aβ42; Supplementary Table 6), followed by incubation with appropriate Horseradish Peroxidase kit or fluorescently labeled secondary antibodies. Band intensities were quantified using Image Studio software (LI-COR), and relative protein expression levels were calculated by normalizing target protein signals to GAPDH or β-actin.

### Machine learning prediction

**Prediction of brain measures and diagnosis.** Two sets of machine learning models were trained to predict brain-based measures using information from the retina: one set of regressors for predicting continuous measures, and one set of classifiers for predicting disease diagnosis. Due to the sample size and the degree of missingness across the variables of interest, random forests[87] with 80 estimators were used for both the regression and classification tasks. The data were split into two portions stratified by diagnosis: 80% for model training and 20% for testing. The 20% test set was left untouched until after the final model was selected and was used only to evaluate model performance. The 80% training set consisted of 56 subjects (17/12/27 normal cognition/MCI/AD), and the 20% test set consisted of 14 subjects (4/3/7 normal cognition /MCI/AD). All models were evaluated using 5-repeated 2-fold cross-validation. The 2-fold cross-validation consisted of randomly splitting the training dataset into two parts stratified by diagnosis, training the model on one part and evaluating it on the other, then swapping the two parts and repeating the process. This is repeated 5 times to obtain a distribution of predicted performance. Data processing and model training were performed using a combination of Scikit-learn[88], Numpy[89], Pandas[90], Scipy[91], and custom Python 3.11 code. Code for performing the machine learning analysis in this study is available on GitHub[92].

**Prediction of brain-based measures.** Random forest models were used to predict brain Aβ plaques, brain NFTs, brain gliosis, brain atrophy, Braak stage, and ABC severity scores, as well as MMSE and MOCA cognitive scores. Different subsets of retinal features were used to train each model: *Chlamydia pneumoniae*, NLRP3, and CCasp3

individually, or each combined with $A\beta_{42}$, gliosis (IBA1, vimentin, GFAP), or the atrophy index. Additionally, $A\beta_{42}$ alone was used as a baseline. For each prediction target, the feature combination produced 13 models to be evaluated. Since the sample size was relatively small and k-fold cross-validation could result in high variance, model performance was evaluated using 5-repeated 2-fold cross-validation, yielding a distribution of coefficients of determination ($r^2$).

**Prediction of disease diagnosis.** Similar to the prediction of brain biomarkers, random forest models were used to predict disease diagnosis from information gathered from the retina. Each subject was identified by one of three disease statuses: MCI (due to AD), AD (dementia), or normal cognition. To obtain ROC curve for this non-binary task, we averaged the ROC curves across disease status. For each of the reported models, we included the area under the ROC curve (AUC) as summary of model performance. Each model was compared against the null distribution by obtaining the AUC distribution generated by dummy classifiers and comparing the model's AUC value to the distribution. To evaluate models against one another, AUC distributions from the 5-repeated 2-fold cross-validation were compared using a Wilcoxon signed-rank test, with $p$ values adjusted for multiple comparisons using the Benjamini-Hochberg procedure.

**Model evaluation against clinical disease severity.** To relate model performance of retinal *Chlamydia pneumoniae* to clinical status, we defined an ordinal disease severity score based on premortem MMSE and CDR and adapted from previous work[81,93,94] (Score 0: no dementia/cognitively normal, CDR = 0, MMSE > 27; Score 1: MCI or mild AD dementia, CDR = 0.5–1, MMSE 20–27; Score 2: moderate AD dementia, CDR = 2, MMSE 10–19; Score 3: severe AD dementia, CDR = 3, MMSE < 10). For each individual in the held-out test set, after a training set, we obtained class prediction probabilities (normal cognition, MCI, AD) from the selected model and quantified their association with disease severity. Model certainty was further characterized by the standard deviation of class probabilities per subject, and prediction probabilities were visualized as a function of disease severity, with subjects colored by true diagnosis.

### Statistics and reproducibility
GraphPad Prism 10.3.1 was used for statistical analyses. Normality of data sets was assessed using the Shapiro–Wilk test to determine suitable statistical comparisons. Comparisons of three or more groups were performed using one- or two-way analysis of variance (ANOVA) followed by Tukey's, Šídák's, or Fisher's least significant difference (LSD) multiple comparison post-hoc tests. Two-group comparisons were analyzed using a two-sided unpaired *t*-test or Mann–Whitney U test. The statistical association between two or more variables was determined by Pearson's ($r_p$ for parametric) or Spearman's ($r_s$ for non-parametric) correlation coefficient test. Pair-wise Pearson's ($r_p$) or Spearman's ($r_s$) coefficients with two-sided unadjusted $p$ values were used for multivariate correlation analyses, as specified in the respective figure legends. Correlation strength was defined by the coefficient ($r$) value as follows: very strong 0.80–1.00, strong 0.60–0.79, moderate 0.40–0.59, and weak 0.20–0.39. Required sample sizes for comparisons of two groups (differential means) were calculated using the nQUERY *t*-test model, assuming a two-sided α level of 0.05, 80% power, and unequal variances, with the means and common standard deviations for the different parameters. Results are expressed as mean ± standard error of the mean (SEM), with $p < 0.05$ considered significant. Fold changes (FC) or percent decreases were calculated and presented. All collected data were included in the analyses. Experiments were randomized, and investigators were blinded during allocation, experimentation, and outcome assessment. Histological analyses

in human and mouse tissues were performed at least three times. Cell culture and Western blot experiments were replicated twice, and qPCR experiments were performed at least three times to ensure reproducibility.

### Reporting summary
Further information on research design is available in the Nature Portfolio Reporting Summary linked to this article.

## Data availability
Most data generated or analyzed for this study are included in this manuscript and supplementary material. Data generated for multi-variable analyses are available on Zenodo and GitHub. All the processed proteomics data generated in this study have been included in the manuscript and the online Supplementary Information file. The mass spectrometry raw files and search results have been deposited to the ProteomeXchange Consortium via the PRIDE partner repository with the dataset identifier PXD040225. Additional data are available from the corresponding author upon reasonable request. Source data are provided with this paper.

## Code availability
The full code and model weights of the deep learning are publicly accessible at GitHub: https://github.com/xomicsdatascience/Retinal_Alzheimer_Prediction.

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

## Acknowledgements

This work has been supported by the NIH/NIA grants: R01AG056478, R01AG055865, and AG056478-04S1 (M.K.H.), R01AG075998 (M.K.H. and T.R.C.), and Alzheimer's Association grant AARG-NTF-21-846586 (T.R.C.). MKH is also supported by The Goldrich and Snyder Foundations. ER has been supported by The Ray Charles Foundation. We thank Elijiah Maxfield for assisting with manuscript editing. We acknowledge the University of Southern California Alzheimer's Disease Research Center (USC-ADRC) and Rush University Alzheimer's Disease Center (RADC) Neuropathology Cores for providing eye and corresponding brain tissues and the neuropathological reports. We thank Drs. Carol A. Miller and Gregory Klein for providing a portion of the human tissues and neuropathological reports. Finally, we thank Drs. Giovanni Meli and Antonino Cattaneo for providing the anti–Aβ oligomer antibody scFvA13, and Drs. Rakez Kayed, Maj-Linda B. Selenica, Daniel C. Lee, and the late Peter Davies for sharing the T22, CitR209, MC-1, and PHF-1 tau antibodies used in our previously published work.

## Author contributions

Study conception and design: B.P.G., Y.K., T.R.C., and M.K.H.; data acquisition: B.P.G., Y.K., J.P.V., A.H., L.S., D.T.F., N.S., A.R., S.S., E.R., J.G.M., T.R.C., and M.K.H.; clinical and behavioral assessment, brain and retinal tissue isolation and processing, as well as histological, neuropathological, and biochemical analyses: Y.K., B.P.G., L.S., J.P.V., D.T.F., N.S., A.R., S.S., D.M., A.V.L., J.A.S., L.S.S., D.H., and M.K.H.; mass spectrometry and data analysis: J.P.V., M.M., B.P.G., Y.K., and M.K.H.; machine learning analysis: A.H., J.G.M., B.P.G., and M.K.H.; statistical analysis: B.P.G., Y.K., D.T.F., J.P.V., A.H., L.S., S.S., J.G.M., and M.K.H.; interpretation of data and discussion: B.P.G., Y.K., K.L.B., M.M., M.A., T.R.C., and M.K.H.; manuscript writing: M.K.H., B.P.G., and Y.K.; manuscript editing and revision: B.P.G., Y.K., J.P.V., A.H., D.T.F., A.V.L., J.A.S., L.S.S., S.L.G.,

V.K.G., M.M., M.A., T.R.C., J.G.M., and M.K.H.; study supervision: M.K.H. All authors read and approved the final manuscript.

## Competing interests

Y.K., K.L.B., and M.K.H. are co-founders of NeuroVision Imaging, Inc. NeuroVision Imaging had no role in study design, data collection, analysis, interpretation, or manuscript preparation, and this work is not related to any product or service of the company. All other authors declare no competing interests.

## Additional information

[1]Department of Neurosurgery, Maxine Dunitz Neurosurgical Institute, Cedars-Sinai Medical Center, Los Angeles, CA, USA. [2]Department of Computational Biomedicine, Cedars-Sinai Medical Center, Los Angeles, CA, USA. [3]Division of Infectious Diseases and Immunology, Department of Pediatrics, Guerin Children's at Cedars-Sinai Medical Center, Los Angeles, CA, USA. [4]Department of Biomedical Sciences, Infectious and Immunologic Diseases Research Center, Cedars-Sinai Medical Center, Los Angeles, CA, USA. [5]Department of Biomedical Sciences, Division of Applied Cell Biology and Physiology, Cedars-Sinai Medical Center, Los Angeles, CA, USA. [6]Board of Governors Regenerative Medicine Institute Eye Program, Cedars-Sinai Medical Center, Los Angeles, CA, USA. [7]David Geffen School of Medicine, University of California Los Angeles, Los Angeles, CA, USA. [8]Alzheimer's Disease Center, Rush University Medical Center, Chicago, IL, USA. [9]Psychiatry and the Behavioral Sciences and Department of Neurology, Keck School of Medicine, University of Southern California, Los Angeles, CA, USA. [10]Leonard Davis School of Gerontology, University of Southern California, Los Angeles, CA, USA. [11]Department of Pathology and Laboratory Medicine, Keck School of Medicine, Children's Hospital Los Angeles, University of Southern California, Los Angeles, CA, USA. [12]Macquarie Medical School, Faculty of Medicine, Health and Human Sciences, Macquarie University, Sydney, NSW, Australia. [13]ProHeme Diagnostics Pty Ltd, Sydney, NSW, Australia. [14]Smidt Heart Institute, Cedars-Sinai Medical Center, Los Angeles, CA, USA. [15]Department of Neurology, Cedars-Sinai Medical Center, Los Angeles, CA, USA. [16]These authors contributed equally: Bhakta Prasad Gaire, Yosef Koronyo. ✉e-mail: timothy.crother@csmc.edu; maya.koronyo@csmc.edu

