## [Transparent Peer Review file · Nature Communications]

Identification of *Chlamydia pneumoniae* and NLRP3 inflammasome activation in Alzheimer's disease retina

Corresponding Author: Professor Maya Koronyo-Hamaoui

Version 0:

Reviewer comments:

Reviewer #1

(Remarks to the Author)

Dr. Gaire and the authors present a well-constructed and supported examination of *Chlamydia pneumoniae* infection in the eye and its association with AD pathology. They expand their analysis into inflammasome pathways for more mechanistic detail and culminate their work with the utilization of machine learning to leverage their findings into potential applications in AD detection via the eye.

This work appropriately cites the previous studies and presents novel findings that are significant to the field in their application to non-invasive diagnosis of AD.

The authors use a variety of methodologies to examine their hypothesis, and additionally provide well-thought-out limitations and shortfalls of their study in their discussion.

Some minor points should be addressed to improve the manuscript:

- 1) It is unclear from the methods how the quantitative immunofluorescent analysis was normalized between images. The images were converted to grayscale and then used thresholding in ImageJ2 to determine the total and percent immunoreactive area. Does the IR total area encompass a single channel, the DAPI channel for cell bodies, or all analyzed channels collapsed into a single image? This should be clarified as it directly impacts the interpretation and statistical analysis.
- 2) It is unclear in Fig 2K if sufficient replicates of the experiment were performed.
- 3) Fig 1 displays differences in Cp infection expression via IF, with some Cp present intracellular with strong signal throughout the cell, and others with loose/punctate staining, particularly in NC samples. This differentiation in Cp infection is lost in using % IR area where individual infection counts are lost. A cell count-based analysis of Cp infection may be a valuable addition in differentiating NC from MCI.
- 4) Similarly to point #3, in Figure 4 the analysis once again uses IR area rather than cell count to examine astrocytic and microglial activation. Given the knowledge that these cells have significantly different shapes and therefore IR areas (as supported in Fig 4I), it is pertinent to assess by cell count rather than IR area.
- 5) While not necessary for the publication, I feel there was a missed opportunity for analysis of the author's existing data. As pointed out by the authors, with 100% of the AD samples expressing Cp signal, there was no availability for comparison to extract the role of Cp on AD pathogenesis or the reverse of AD on Cp infection. While the AD vs NC provides the easiest and most stark contrast, the application of prediction models to observe the presence of AD is clinically too late for intervention. A much more valuable tool would be the detection of MCI. It would be more powerful if the authors were able to show model efficacy after training on NC vs AD for the detection of AD in MCI sample sets.
- 6) Continuing from point #5, there is a clear subset of NC samples that express high levels of Cp comparable to AD. A thorough analysis of this unique NC subset vs the AD equivalent could aid significantly in elucidating the impacts of Cp on AD.

Reviewer #2

(Remarks to the Author)

Reviewer Comments

This is an excellent manuscript addressing a critical and understudied aspect of Alzheimer's disease (AD) pathology—namely, the potential contribution of *Chlamydia pneumoniae* infection and NLRP3 inflammasome activation to both brain and retinal inflammation. The study is compelling and has significant implications for understanding neuroinflammation, amyloid pathology, and neurodegeneration in AD. The integration of immunohistochemistry, proteomics, and machine learning to investigate retinal biomarkers represents a powerful multi-modal approach.

That said, given the novelty and significance of the findings, particularly the detection of *Chlamydia pneumoniae* in the retina, I have one important point that should be addressed prior to publication:

1. Validation of *Chlamydia pneumoniae* Detection

While the data suggesting the presence of *Chlamydia pneumoniae* in AD and mild cognitive impairment (MCI) retinas is intriguing, it is critical that the authors validate these findings using multiple independent techniques to rule out cross-reactivity or non-specific staining. This is especially important given the possibility that the antibodies used (MBS534621 and MA5-18183) could potentially cross-react with other antigens.

To strengthen the claim, the authors should confirm the presence of *Chlamydia pneumoniae* in the retinal samples using at least two of the following methods:

- Polymerase Chain Reaction (PCR) targeting *C. pneumoniae*-specific genes
- In situ hybridization (ISH) for *C. pneumoniae*-specific RNA
- Transmission electron microscopy (TEM) to identify chlamydial developmental forms within retinal cells
- Giemsa staining, while less sensitive, can be used to visualize cytoplasmic inclusion bodies

This is particularly crucial given conflicting reports in the literature. For example, Balin et al. (1998, *Med Microbiol Immunol*) reported detection of *C. pneumoniae* in AD brains using PCR and EM, while Taylor et al. (2002, *Neurology*) were unable to replicate these findings. Given these prior inconsistencies, additional validation with quantification of the results is essential to ensure reproducibility and confidence in the results.

Minor

2. Clarification of Abbreviations and Terminology

Please avoid shorthand notations such as “Cp” for *Chlamydia pneumoniae* and “NC” for normal cognition. These abbreviations are not universally recognized and can be confusing—e.g., “CP” is often used to refer to the choroid plexus. For clarity, *Chlamydia pneumoniae* and normal cognition should be written out in full throughout the manuscript.

3. Clarification on Mass Spectrometry-Based Proteomics

The proteomic data presented in Figure 2 closely resembles data published in Koronyo et al. (*Acta Neuropathol*, 2023) and appears similar to datasets used in Shi et al. (*Alzheimers Dement*, 2023). The Methods section states that the proteomics was conducted by the University of Queensland and implies a new analysis. However, to avoid any confusion or concerns of data duplication, the authors should explicitly state whether the proteomics data in the current manuscript is entirely new, or a reanalysis of previously published datasets.

If reanalyzed, the Methods and Results sections should clearly reference the original datasets. Transparency here is essential, particularly as the dataset identifier PXD040225 could not be located in the PRIDE database at the time of review.

Conclusion

This is an impactful and innovative study with broad implications for our understanding of neuroinflammation and infection in Alzheimer's disease. If the authors can confirm the presence of *Chlamydia pneumoniae* in retinal tissue using at least two additional independent techniques, I would be strongly supportive of publication.

Reviewer #3

(Remarks to the Author)

1. Please clarify the number of subjects in each group in the Materials and Methods section and the results section.

2. Are there measurements of pTau to total tau or pTau at Threonine 217 levels in the retinal samples to see if they correlate with Cp expression?

3. The Inner plexiform layer is mentioned on Line 449 as one of the regions checked for Cp expression, but is not mentioned on Lines 421-422.

4. Please clarify the use of the polyclonal Antibody that is not specific to only Cp.

5. The manuscript describes associations between DEPs and Aβ1-42 or tau levels (lines 526–527), yet no statistical measures (e.g., correlation coefficients or p-values) are reported to support these claims. Additionally, the assertion that these DEPs are “specifically *Chlamydia*-related proteins” (line 529) overstates their specificity, as many are broadly involved in general bacterial or immune responses. To strengthen this section, we recommend (1) reporting the statistical strength of the associations or referencing the relevant figure/table, and (2) rephrasing to reflect that these proteins have been previously implicated in *Chlamydia*-related interactions, rather than implying exclusivity.

6. As IBA is also a marker for macrophages, has another marker specific for microglia been studied, such as Cx3cr1, to confirm the result that Cp phagocytosis by microglia is impaired in AD patients?

7. In the discussion, there is no mention of cleaved Caspase 3, even though it is mentioned in the abstract and the results.

Although the study provides compelling correlations between *Chlamydia pneumoniae* and various AD-related pathologies in the retina and brain, a few fundamental limitations temper the strength of its conclusions. First, the study does not strongly establish a causal relationship between Cp infection and the progression of AD. Although increased Cp inclusions are observed in MCI and AD tissues, these remain associative and lack evidence demonstrating that Cp initiates or accelerates neurodegeneration. Second, the specificity of the Cp detection methods, particularly the use of a polyclonal antibody that may cross-react with other *Chlamydia* species. Although monoclonal antibodies were later used for specificity, the conclusion of Cp, especially within complex tissues like retina and brain, could still be confounded by non-specific staining. Third, the analyses largely rely on correlative data without functional validation. Many conclusions hinge on statistical associations rather than experimentally dissecting the mechanistic pathways by which Cp might induce inflammasome activation or contribute to tau and amyloid pathology.

In conclusion, this is an important study that is likely to generate a lot of discussion on the role of infection in AD. The authors may consider addressing the concerns raised regarding the paper.

Reviewer #4

(Remarks to the Author)

Version 1:

Reviewer comments:

Reviewer #1

(Remarks to the Author)

The authors took significant time in addressing all of the reviewers' comments. Clarity, experimental depth, reanalysis, and new supporting findings successfully combine to convincingly answer the hypothesis. The additional experiments provide the necessary data to support the hypothesis and expand the construction at home. and further refine existing runs with weak n.

I support this revised manuscript moving forward toward publishing.

(Remarks on code availability)

Reviewer #2

(Remarks to the Author)

The authors have addressed my previous comments, and I am now supportive of publication of the manuscript.

(Remarks on code availability)

Reviewer #3

(Remarks to the Author)

As my co-reviewer and I stated before this was a compelling work. The authors have addressed most of the reviewers' comments. Although they did not get positive results from one of the microglia markers that was suggested but their rationale for using TMEM119 was justified. Most importantly this work will be another important stepping stone in cementing the role of infection in AD progression. This work when published will likely generate a lot of interesting discussions.

(Remarks on code availability)

Reviewer #4

(Remarks to the Author)

(Remarks on code availability)

*We thank the editors and reviewers for their thoughtful, constructive assessments and the opportunity to revise our manuscript. We carefully addressed every comment raised by the reviewers. Briefly, we implemented extensive new validation of Chlamydia pneumoniae in the human retina using Giemsa staining, Chlamydia pneumoniae-specific in situ hybridization, argR-targeted qPCR, and immunolabeling with a species-specific monoclonal antibody (revised **Figure 1G–I** and **Suppl. Figures 3–4**). We increased the number of retinal donors for Western blotting, added Chlamydia pneumoniae-positive cell counts, and new astrocyte/microglia cell-count endpoints, validated Chlamydia pneumoniae-associated IBA1⁺ cells as TMEM119⁺ resident microglia, clarified proteomic data analysis and accessibility, and added correlations with p-tau/total tau ratios. Notably, we now provide direct functional evidence that Chlamydia pneumoniae acts as a driver rather than a bystander by showing that infection of human neuroblastoma cells induces NLRP3 inflammasome activation, neurotoxicity, and A β ₄₂ accumulation, and that intranasal infection of APP_{SWE}/PS1 Δ E9 mice amplifies brain Nlrp3/Il1 β /Il6 expression, gliosis, A β plaque burden, and visuocognitive deficits (new **Figure 3** and **Suppl. Figure 9**). Finally, we included machine-learning evaluation against clinical disease severity and MCI (revised **Figure 6D** and new **Suppl. Figure 15E**), systematically reported group sizes, and standardized nomenclature and limitations throughout. Together, these additions and revisions markedly strengthen the robustness, mechanistic insight, and translational relevance of our study while remaining fully responsive to the reviewers' concerns.*

Reviewer #1.

Dr. Gaire and the authors present a well-constructed and supported examination of Chlamydia pneumoniae infection in the eye and its association with AD pathology. They expand their analysis into inflammasome pathways for more mechanistic detail and culminate their work with the utilization of machine learning to leverage their findings into potential applications in AD detection via the eye. This work appropriately cites the previous studies and presents novel findings that are significant to the field in their application to non-invasive diagnosis of AD. The authors use a variety of methodologies to examine their hypothesis, and additionally provide well-thought-out limitations and shortfalls of their study in their discussion.

We thank the reviewer for their thoughtful and positive evaluation of our work.

Some minor points should be addressed to improve the manuscript:

1) It is unclear from the methods how the quantitative immunofluorescent analysis was normalized between images. The images were converted to grayscale and then used thresholding in ImageJ2 to determine the total and percent immunoreactive area. Does the IR total area encompass a single channel, the DAPI channel for cell bodies, or all analyzed channels collapsed into a single image? This should be clarified as it directly impacts the interpretation and statistical analysis.

Response 1: To address this comment, in the revised manuscript we further elaborated regarding the methods of how the quantitative immunofluorescent analysis was normalized between images, see revised **Materials and Methods (Page 9-10, Line 273-282)**, as follows:

“...For each biomarker (e.g., *Chlamydia pneumoniae*, NLRP3, caspase-1, ASC, NGSDMD, vimentin), single-channel 20× images (covering $\sim 150 \times 10^3 \mu\text{m}^2$ area/per image) were converted to grayscale and standardized to baseline intensity by using a histogram-based threshold in ImageJ2/Fiji. This baseline-derived threshold was then applied uniformly to the corresponding single channel for all subjects across diagnostic groups, and ImageJ2/Fiji particle analysis was used to quantify, for each biomarker, the total immunoreactive (IR) area and the percentage IR area. For cell counts, the same fields of view used for IR area quantification in the 20× images were analyzed, and cells co-expressing DAPI with either *Chlamydia pneumoniae*, GFAP, or IBA1 were manually enumerated using the grid-overlay mode in Adobe Photoshop or ZEN software tools.”

2) It is unclear in Fig 2K if sufficient replicates of the experiment were performed.

Response 2. We infer that this comment refers to Fig. 3K panel in the original submission (as there was no Fig. 2K), which presented pro- and cleaved IL-1 β . To directly address this concern, we carried out an additional Western blot experiment to increase the number of human donors. Given the extremely limited availability of fresh retinal tissue from clinically and neuropathologically confirmed MCI/AD cases and normal-cognition controls, we collected and analyzed the maximum number of additional samples obtainable, adding 4 new retinal tissues. The revised dataset now includes results from 5 AD donors (red dots), 1 MCI due-to-AD donor (green dot), and 4 normal controls (gray dots), as shown in the updated **Figure 4K** (see below). The corresponding full, uncropped gels are provided in the Source Data file. These extended data are included in the revised **Results (Page 29, Lines 877-879)** and **Figure 4 legend** (donors' information is detailed in **Suppl. Table 2 and 4**).

3) Fig 1 displays differences in Cp infection expression via IF, with some Cp present intracellular with strong signal throughout the cell, and others with loose/punctate staining, particularly in NC samples. This differentiation in Cp infection is lost in using % IR area where individual infection counts are lost. A cell count-based analysis of Cp infection may be a valuable addition in differentiating NC from MCI.

Response 3. We agree with the reviewer and have now included a new retinal *Chlamydia pneumoniae*-positive cell-count analysis in the revised **Figure 1E**. All images and human donors were analyzed for *Chlamydia pneumoniae*-positive cell counts. The resulting data (revised **Fig. 1E**, see below) closely parallel our original quantification of *Chlamydia pneumoniae*-positive percentage IR area (now **Fig. 1F**), in that they likewise do not distinguish MCI from normal-cognition controls. On this basis, and because percentage IR area provides a more continuous and robust measure of retinal pathogen burden, we retained %IR area as the primary quantitative metric for all correlation and ML-based prediction analyses. The new *Chlamydia pneumoniae*-positive cell-count analysis is now described in the **Materials and Methods (Page 10, Lines 279-282)** and incorporated into the revised **Figure 1 legend and Results (Page 21, Lines 617-620)**.

4) Similarly to point #3, in Figure 4 the analysis once again uses IR area rather than cell count to examine astrocytic and microglial activation. Given the knowledge that these cells have significantly different shapes and therefore IR areas (as supported in Fig 4I), it is pertinent to assess by cell count rather than IR area.

Response 4. As suggested by the reviewer, we performed additional glial cell-count analyses for retinal DAPI⁺IBA1⁺ microglia and retinal DAPI⁺GFAP⁺ astrocytes and have included these results into the revised figures (**Fig. 5B** for retinal astrocyte count and **Fig. 5H** for retinal microglia cell count), see below.

These analyses revealed increased glial cell numbers only at the AD dementia stage, but not in MCI. By contrast, the IR area-based measures, which also capture changes in glial morphology and process complexity, revealed earlier retinal glial alterations that

significantly discriminated MCI from normal-cognition controls. These new data were incorporated in the revised manuscript text, as follows:

Results (Page 30-31, Line 911-919):

“Quantitative IHC analyses of retinal GFAP⁺ astrocytes and IBA1⁺ microglia revealed significantly increased glial cell counts in the AD retina (1.5-fold, $p < 0.05$ – 0.001) but not in the MCI retina (**Fig. 5B,H**). Interestingly, analysis of the IR area of GFAP⁺ and vimentin⁺ macroglia, as well as IBA1⁺ microglia, thereby accounting for cell morphology and process hypertrophy, showed significant expansion of all gliosis markers in MCI retinas (1.5–1.9-folds, $p < 0.05$ – 0.01), with further marked expansion observed in AD retinas (2.3–2.8-folds, $p < 0.0001$) relative to normal-cognition controls (**Fig. 5C,E,I**; extended data in **Suppl. Fig. 11C–E**). These findings suggest that glial activation and process hypertrophy emerge early along the retinal AD continuum, followed by overt glial proliferation in established AD dementia.”

Discussion (Page 38, Line 1138-1141):

“We also uncover a complex glial response to *Chlamydia pneumoniae* in the AD retina. Retinal GFAP⁺ and vimentin⁺ macrogliosis and IBA1⁺ microgliosis findings suggest that glial activation and process hypertrophy emerge early along the AD continuum (in MCI due to AD), followed by overt glial proliferation in established AD dementia.”

Regarding the *Chlamydia pneumoniae*-associated microglia (original Fig. 4I), this was already a cell-count analysis, and the revised graph labeling is now clarified to better reflect this type of analysis (see revised **Fig. 5L** and **5M**, below). Figure 5 legends and related Results were adjusted accordingly.

5) While not necessary for the publication, I feel there was a missed opportunity for analysis of the author's existing data. As pointed out by the authors, with 100% of the AD samples expressing Cp signal, there was no availability for comparison to extract the role of Cp on AD pathogenesis or the reverse of AD on Cp infection. While the AD vs NC provides the easiest and most stark contrast, the application of prediction models to observe the presence of AD is clinically too late for intervention. A much more valuable tool would be the detection of MCI. It would be more powerful if the authors were able to show model efficacy after training on NC vs AD for the detection of AD in MCI sample sets.

Response 5: We agree that the detection of MCI would be more clinically useful; given the paucity of subjects in the MCI stage (confirmed postmortem as AD), working around the MCI diagnosis is particularly difficult. Having already gone through model selection and evaluating the test set, modifying the train/test split and re-training the model would risk significant data leakage. Instead, we have examined how the retinal *Chlamydia pneumoniae* model's prediction probabilities vary along diagnosis. We have also added some granularity for prediction of disease severity, based on premortem cognitive status (CDR and MMSE) and previous studies, as described below in revised Materials and Methods. We find that prediction probabilities for predicting AD increases with severity; more severe cases are rated higher. However, the model made less-certain predictions across all labels for the MCI cases, as indicated by the lower std. across labels compared to normal cognition (NC) and AD. Given the low number of MCI patients in the training set, it is a promising direction as it suggests that there could be a continuum between NC/MCI/AD that would become apparent with additional patients in the dataset (revised **Suppl. Fig. 15E**, see below).

We have also included the AUC distribution predictions by retinal *Chlamydia pneumoniae* for each diagnostic group, showing improved prediction for AD but less for MCI and NC groups (revised **Figure 6D**, see below):

We further highlighted **Table 2** referring to the MCI predictions. We have now revised the manuscript text as follows:

Materials and Methods (Page 19, Lines 568-577):

“Model evaluation against clinical disease severity. To relate model performance of retinal Chlamydia pneumoniae to clinical status, we defined an ordinal disease severity score based on premortem MMSE and CDR and adapted from previous work^{87,109,110} (Score 0: no dementia/cognitively normal, CDR = 0, MMSE > 27; Score 1: MCI or mild AD dementia, CDR = 0.5–1, MMSE 20–27; Score 2: moderate AD dementia, CDR = 2, MMSE 10–19; Score 3: severe AD dementia, CDR = 3, MMSE < 10). For each individual in the held-out test set, after a training set, we obtained class prediction probabilities (normal cognition, MCI, AD) from the selected model and quantified their association with disease severity. Model certainty was further characterized by the standard deviation of class probabilities per subject, and prediction probabilities were visualized as a function of disease severity with subjects colored by true diagnosis.”

Results (Page 34, Lines 1011-1020):

*“We further examined the selected model’s performance by assessing its predictions compared to a measure of clinical disease severity (scores 0–3), based on premortem CDR and MMSE cognitive performance. Using the test set, we obtain the prediction probability for each diagnosis and compare these against disease severity (**Suppl. Fig. 15E**). There are high correlations between the prediction probabilities of both normal cognition ($r=-0.78$) and AD ($r=0.79$) against disease severity. There was no correlation between the MCI prediction and disease severity, though that is expected. Looking at the prediction probabilities for each diagnosis, we see that the standard deviation across normal cognition, MCI, and AD are 0.41, 0.22, and 0.34 respectively. Highly certain predictions would result in higher standard deviations; predictions for MCI patients are less certain, indicating the possibility that MCI could be detectable with additional training samples.”*

Discussion (Page 38-39, Lines 1165-1172):

“Further, retinal Chlamydia pneumoniae is highly informative for discriminating AD status and severity, and although performance for MCI classification is limited, likely reflecting small sample size, the reduced prediction certainty in MCI suggests that with larger datasets and additional features, infection- and inflammasome-linked retinal markers could help resolve intermediate disease states. Importantly, models were not retrained after test-set evaluation to avoid data leakage, and the present results do not exclude the possibility that MCI will become reliably identifiable as more data and features are incorporated.”

6) Continuing from point #5, there is a clear subset of NC samples that express high levels of Cp comparable to AD. A thorough analysis of this unique NC subset vs the AD equivalent could aid significantly in elucidating the impacts of Cp on AD.

Response 6: To address this point, in the revised manuscript, we re-examined the subset of 3 normal-cognition donors with high *Chlamydia pneumoniae* burden (all females) and compared them with other normal cognition individuals. We found that the retinal and brain AD related pathologies in these *Chlamydia pneumoniae*-high normal cognition cases were largely comparable to other normal cognition subjects. However, high-*Chlamydia pneumoniae* normal-cognition individuals exhibited more impaired capacity to recognize/engulf *Chlamydia pneumoniae*-positive cells, compared to the other normal-cognition samples. Intriguingly, retinal *Chlamydia pneumoniae*-associated IBA1⁺ microglia (CAM) strongly and inversely correlated with retinal *Chlamydia pneumoniae* burden (new **Fig. 5O**; $r_p = -0.72$, $p < 0.0001$, for all diagnostic groups; and new **Suppl. Fig. 12E**; $r_p = -0.79$, $p = 0.0004$, for normal-cognition donors only; black dots show the 3 females with high retinal *Chlamydia pneumoniae* burden).

Figure5O

Suppl. Figure 12E

In the revised manuscript text, we addressed this point in the Results and Discussion sections as follows:

Results (Page 31-32, Lines 948-954):

“Within the normal-cognition group, three female individuals exhibited a disproportionately low percentage of CAM relative to retinal bacterial burden, comparable to the pattern observed in AD (Fig. 5M and Suppl. Fig. 12E). Notably, these individuals also showed the highest retinal Chlamydia pneumoniae loads (Fig. 1F), with comparable levels of retinal and brain AD-related pathology, suggesting a selective impairment of microglial engagement with bacteria-infected cells even in some clinically normal individuals.”

Discussion (Page 38, Lines 1148-1159):

“A similar dysfunctional pattern was evident in three female clinically normal individuals with disproportionately high retinal Chlamydia pneumoniae burden but low CAM-to-bacterial load ratios, despite exhibiting retinal and cerebral AD-related pathology comparable to that seen in normal-cognition donors. Together with the strong inverse relationship between CAM-to-bacterial percentages and bacterial burden, these findings suggest that a failure of microglial containment of Chlamydia pneumoniae–infected cells can emerge even before overt cognitive decline. We propose that in these individuals, microglia may remain relatively competent in managing A β clearance yet be selectively impaired in recognizing or clearing Chlamydia pneumoniae, allowing pathogen burden to escalate. In this framework, convergence of high Chlamydia pneumoniae load, elevated A β , and sustained inflammasome activation may represent a critical transition point that links chronic infection to progressive neurodegeneration and the eventual emergence of clinical symptoms.”

Reviewer #2.

This is an excellent manuscript addressing a critical and understudied aspect of Alzheimer's disease (AD) pathology—namely, the potential contribution of Chlamydia pneumoniae infection and NLRP3 inflammasome activation to both brain and retinal inflammation. The study is compelling and has significant implications for understanding neuroinflammation, amyloid pathology, and neurodegeneration in AD. The integration of immunohistochemistry, proteomics, and machine learning to investigate retinal biomarkers represents a powerful multi-modal approach.

We thank the reviewer for the positive feedback and constructive comments.

That said, given the novelty and significance of the findings, particularly the detection of Chlamydia pneumoniae in the retina, I have one important point that should be addressed prior to publication:

1. Validation of Chlamydia pneumoniae Detection. While the data suggesting the presence of Chlamydia pneumoniae in AD and mild cognitive impairment (MCI) retinas is intriguing, it is critical that the authors validate these findings using multiple independent techniques to rule out cross-reactivity or non-specific staining. This is especially important given the possibility that the antibodies used (MBS534621 and MA5-18183) could potentially cross-

react with other antigens. To strengthen the claim, the authors should confirm the presence of *Chlamydia pneumoniae* in the retinal samples using at least two of the following methods:

- Polymerase Chain Reaction (PCR) targeting *C. pneumoniae*-specific genes
- In situ hybridization (ISH) for *C. pneumoniae*-specific RNA
- Transmission electron microscopy (TEM) to identify chlamydial developmental forms within retinal cells
- Giemsa staining, while less sensitive, can be used to visualize cytoplasmic inclusion bodies

This is particularly crucial given conflicting reports in the literature. For example, Balin et al. (1998, *Med Microbiol Immunol*) reported detection of *C. pneumoniae* in AD brains using PCR and EM, while Taylor et al. (2002, *Neurology*) were unable to replicate these findings. Given these prior inconsistencies, additional validation with quantification of the results is essential to ensure reproducibility and confidence in the results.

Response 1: *We thank the reviewer for this important comment and fully agree that independent validation of Chlamydia pneumoniae detection is critical, particularly in light of prior conflicting reports in the literature. In response, and as now detailed in the revised Materials and Methods, Results, and Discussion, we have implemented multiple, orthogonal validation strategies that go beyond single-antibody IHC and directly address concerns regarding cross-reactivity and non-specific staining. First, as previously shown, to minimize antibody-related artifacts, we used two independent anti-Chlamydia reagents: a broadly reactive polyclonal antibody and a Chlamydia pneumoniae-specific monoclonal antibody that does not cross-react with other Chlamydia species. Both antibodies, applied with fluorescence- and peroxidase-based IHC in retina and matched A9 cortex, revealed morphologically consistent intracellular inclusions.*

Second, in direct alignment with the reviewer's suggestions, we performed three additional, independent validation assays: Giemsa staining, Chlamydia pneumoniae-specific fluorescence in situ hybridization (FISH), and quantitative PCR. Giemsa staining, particularly useful for visualizing bacteria, and although not species-specific, revealed dark blue intracytoplasmic inclusion bodies that indicate bacterial presence, in retinal cross-sections (especially from AD donors), with morphology and distribution closely resembling inclusions as detected by Chlamydia pneumoniae mAb in the AD retina (Fig. 1G, see below). In addition, Giemsa staining in Chlamydia pneumoniae-infected mouse lungs (positive control) demonstrated the same patterns of dark blue inclusions of Chlamydia pneumoniae as identified in the retina (Suppl. Fig. 3A,B).

FISH: using a fluorescently labeled Chlamydia pneumoniae-specific DNA probe further verified the presence of Chlamydia pneumoniae genomic material within retinal cells (Fig. 1H, see below; Suppl. Fig. 4); importantly, no fluorescent signal was observed in “no probe” controls for either AD or normal-cognition tissues, confirming specificity and low background. Both Giemsa and FISH analyses showed a higher burden of inclusions in AD retinas relative to normal-cognition controls, in agreement with the Chlamydia pneumoniae-specific monoclonal antibody-based quantification.

Third, qPCR analysis targeting the Chlamydia pneumoniae-specific argR gene detected Chlamydia pneumoniae DNA in retinal tissues from 2/2 AD cases, 0/1 MCI case, and 1/2 normal-cognition controls (Fig. 1I, see below). Although the sample size is necessarily small due to the scarcity of fresh, well-characterized retinal tissue and the partial consumption of tissue for histology and biochemical analyses, these qPCR data independently confirm that Chlamydia pneumoniae genomic material is present in human retinas, including AD cases.

qPCR (human retina)

Taken together, these convergent lines of evidence—(i) consistent detection of inclusion-like structures with both a broadly reactive pAb and a *Chlamydia pneumoniae*-specific mAb in retina and matched cortex; (ii) morphologically compatible inclusion bodies by Giemsa staining; (iii) species-specific genomic signal by FISH with appropriate negative controls; and (iv) detection of the *Chlamydia pneumoniae*-specific *argR* gene by qPCR, provide robust, multi-modal support for the presence of *Chlamydia pneumoniae* in the human retina and its preferential elevation in AD.

We have clarified these validation steps in the revised **Materials and Methods (Pages 12-13)**, revised **Figure 1G-I** and **Suppl. Figs. 3-4**, Results, and Discussion text, as follows:

Results (Page 21-22, Lines 637-652):

“The existence of *Chlamydia pneumoniae* in the human retina was further validated using three complementary histological and molecular approaches: Giemsa staining, fluorescence in situ hybridization (FISH), and genomic DNA by real-time quantitative polymerase chain reaction (q)PCR (**Fig. 1G-I**; extended data in **Suppl. Fig. 3-4**). Although Giemsa staining is not specific to a particular bacterial species, the inclusion bodies in retinal cross-sections appear as dark blue structures, consistent with those observed in *Chlamydia pneumoniae*-infected mouse lung tissues (**Fig. 1G, Suppl. Fig. 3A,B**; $n = 8$ donors) and *Chlamydia pneumoniae* inclusions identified by immunostaining with the mAb in the AD retina and brain (**Fig. 1C,D**). A FISH analysis using a fluorescently labeled *Chlamydia pneumoniae*-specific DNA probe further verified the presence of this bacterial genomic material within retinal tissues (**Fig. 1H** and **Suppl. Fig. 4**; $n = 11$ donors), which was absent in the no probe retinal AD and normal cognition tissues. Both Giemsa and FISH analyses demonstrated a higher burden of inclusions in AD retinas compared with those from individuals with normal cognition. Notably, qPCR analysis detected the *Chlamydia pneumoniae*-specific arginine repressor (*argR*) gene in retinal tissues from 2 of 2 AD cases, 0 of 1 MCI case, and 1 of 2 normal cognition controls, confirming the presence of *Chlamydia pneumoniae* in the human retina (**Fig. 1I**; $n = 5$ donors).”

Discussion (Page 34, Lines 1025-1027): “Using multiple complementary approaches, including bacterial-specific monoclonal antibody, in-situ hybridization, and *argR* qPCR, we

confirm the presence of Chlamydia pneumoniae in the retina and show that its burden is markedly increased in AD.”

Discussion (Page 35, Lines 1060-1064): *“In this study, we confirmed the presence of Chlamydia pneumoniae in the human retina, particularly in both somatic and perinuclear compartments of cells within the GCL and INL, using complementary approaches including monoclonal and polyclonal immunolabeling, Giemsa staining, bacterial DNA-specific in situ hybridization, and argR-targeted qPCR.”*

Minor

2. Clarification of Abbreviations and Terminology

Please avoid shorthand notations such as “Cp” for Chlamydia pneumoniae and “NC” for normal cognition. These abbreviations are not universally recognized and can be confusing—e.g., “CP” is often used to refer to the choroid plexus. For clarity, Chlamydia pneumoniae and normal cognition should be written out in full throughout the manuscript.

Response 2: *As recommended, we have adopted standard scientific nomenclature for Chlamydia pneumoniae throughout the revised manuscript text and no longer use an abbreviation in the main text. To address space limitations in figures and tables, we now consistently use the abbreviation “Cpn” (rather than “Cp”) for Chlamydia pneumoniae, in line with prior literature (PMID: 2554007; PMID: 20445987; PMID: 31626972). Similarly, in the main text we uniformly use the term “normal cognition” rather than “NC,” while retaining the abbreviation “NC” only in figures and tables, due to space limitation and consistent with previous reports, including our own [Alzheimer’s & Dementia: The Journal of Alzheimer’s Association 2025 (PMID: 39560003); Acta Neuropathologica 2023 (PMID: 36773106); Progress in Retinal and Eye Research 2024 (PMID: 38759947); Nature Communications 2024 (PMID: 39690174)].*

3. Clarification on Mass Spectrometry-Based Proteomics

The proteomic data presented in Figure 2 closely resembles data published in Koronyo et al. (Acta Neuropathol, 2023) and appears similar to datasets used in Shi et al. (Alzheimer’s Dement, 2023). The Methods section states that the proteomics was conducted by the University of Queensland and implies a new analysis. However, to avoid any confusion or concerns of data duplication, the authors should explicitly state whether the proteomics data in the current manuscript is entirely new, or a reanalysis of previously published datasets.

If reanalyzed, the Methods and Results sections should clearly reference the original datasets. Transparency here is essential, particularly as the dataset identifier PXD040225 could not be located in the PRIDE database at the time of review.

Response 3: *We appreciate the opportunity to clarify this matter. The mass spectrometry data were reanalyzed from our original dataset (Acta Neuropathologica 2023, PMID 36773106) to identify Chlamydia pneumoniae-related proteomic changes in the retina and*

brain. This has been clearly stated and cited in the Materials and Methods and Results sections of the revised manuscript, as below.

Materials and Methods (Page 11, Lines 314-316): “...In this study, we performed a secondary analysis on previously published retinal and brain mass spectrometry data⁵⁴, to identify DEPs associated with gram-negative bacterial infection and Chlamydia interactomes.”

Materials and Methods (Page 12, Lines 336-338): “The original mass spectrometry proteomics data have been deposited to the ProteomeXchange Consortium via the PRIDE⁹⁶ partner repository with the dataset identifier PXD040225.”

Results (Page 20, Lines 587-589): “To investigate protein expression profiles related to Chlamydia pneumoniae infection, we reanalyzed previously described mass spectrometry (MS)-based proteomic datasets⁵⁴ from human retinal and cortical tissues.”

In addition, we have confirmed that the dataset identifier PXD040225 is now publicly accessible.

Data Availability Statement (Page 43, Lines 1271-1273): “The mass spectrometry raw files and search results have been deposited to the ProteomeXchange Consortium via the PRIDE partner repository with the dataset identifier PXD040225.”

Conclusion

This is an impactful and innovative study with broad implications for our understanding of neuroinflammation and infection in Alzheimer's disease. If the authors can confirm the presence of Chlamydia pneumoniae in retinal tissue using at least two additional independent techniques, I would be strongly supportive of publication.

We appreciate the reviewer's positive assessment. In direct response to this key recommendation, we implemented three additional, orthogonal validation assays (Giemsa staining, C. pneumoniae-specific FISH, and qPCR), all of which independently confirmed the presence of Chlamydia pneumoniae in human retinal tissue.

Reviewer #3.

1. Please clarify the number of subjects in each group in the Materials and Methods section and the results section.

Response 1: We have specified the number of subjects for each group and analysis within the revised **Material and Methods** (Human eye and brain samples: **Pages 5-6, Lines: 143-157, Page 11, Line 310**; Mice: **Page 14, Lines: 409-413**; Cells: **Pages 17, Lines: 491-501**)

and throughout the **Results** section. In addition, **Figure 1A** analyses workflow, figure legends, and tables further detail sample sizes.

2. Are there measurements of pTau to total tau or pTau at Threonine 217 levels in the retinal samples to see if they correlate with Cp expression?

Response 2: As suggested by the reviewer, we reanalyzed five phospho-tau isoform-to-total tau ratios (pS396, pS404, pS214, pS199, pS231/total tau) in the same human retinal cohort, using our previously reported NanoString GeoMx DSP dataset (PMID: 38980423). Retinal pT217-tau measurements were not available. In the revised manuscript, we assessed correlations between retinal *Chlamydia pneumoniae* burden and each pTau/total tau ratio; none were statistically significant. These new analyses are now included in **Supplementary Figure 5C–G**.

And are mentioned in the revised manuscript text, as follows:

Materials and Methods (Page 11, Lines 307-313):

“NanoString GeoMx® Digital Spatial Profiling of total tau and p-tau reanalysis: We performed a secondary analysis of the NanoString GeoMx® Digital Spatial Profiling of total tau and 5 different p-tau (S214; T231; S199; S396; S404) species in formalin-fixed paraffin embedded human retinal cross-sections from 8 AD, 6 MCI, and 8 normal-cognition control cases ($n = 22$), as detailed in our previous study⁷¹. In this study, we used the values of total tau and p-tau to perform the correlation study with retinal *Chlamydia pneumoniae* burden with p-tau/total tau ratio as a secondary analysis.”

Results (Page 22, Lines 668-670):

“..., nor with p-tau/total tau ratios at phosphorylation sites S404, S396, S199, S231, or S214 quantified by NanoString GeoMx digital spatial profiling in a subset of this cohort⁷¹ (Suppl. Fig. 5C-G; $n = 22$ donors).”

Discussion (Page 36, Lines 1078):

“..., but not AT8⁺ p-tau, MC1⁺ tangles or multiple isoforms of p-tau/total tau ratios.”

3. The Inner plexiform layer is mentioned on Line 449 as one of the regions checked for Cp expression, but is not mentioned on Lines 421-422.

Response 3: Thank you for pointing out this typo. “IPL” was mistakenly included in the original line 449, as *Chlamydia pneumoniae* is predominantly observed in the nuclear layers (GCL and INL). In the revised manuscript, we have corrected this typo as follows:

Results (Page 21, Lines 610-611):

“The retinas of MCI and AD patients... exhibited predominant *Chlamydia pneumoniae*-positive signals in retinal ganglion cell layer (GCL) and inner nuclear layer (INL).”

4. Please clarify the use of the polyclonal Antibody that is not specific to only Cp.

Response 4: We thank the reviewer for this point and have clarified the role of the polyclonal antibody in the revised manuscript. Our approach was intentionally stepwise. First, to broadly assess the presence of *Chlamydia* species in the AD retina, we used a polyclonal antibody (pAb) that had been verified against three species (*C. pneumoniae*, *C. trachomatis*, *C. psittaci*). This pAb, applied with both fluorescence- and peroxidase-based IHC, revealed characteristic *Chlamydia*-positive inclusion structures in AD and MCI retinas. Then, we confirmed the specific presence of *Chlamydia pneumoniae* using a *Chlamydia pneumoniae*-selective monoclonal antibody (mAb) that does not cross-react with other *Chlamydia* species, in both retina and matched A9 cortex. The inclusions detected with the mAb recapitulated the subcellular patterns observed with the pAb (cytosolic puncta, perinuclear and nuclear-associated aggregates) and matched previously reported inclusion morphologies in AD brain. Importantly, all quantitative data attributed to *Chlamydia pneumoniae* in the manuscript, including positive cell counts, immunoreactive area measurements, and correlation analyses, are based exclusively on the *Chlamydia pneumoniae*-specific mAb. The polyclonal antibody is used only to document concordant staining and to illustrate the initial broad screen for *Chlamydia* species, and this distinction is explicitly stated in the revised manuscript, as below:

Results (Page 20, Lines 597-609):

“We applied a stepwise strategy to establish the existence and distribution of *Chlamydia pneumoniae* in the human retina. Initially, we broadly screened for *Chlamydia* species by performing IHC with an anti-*Chlamydia* polyclonal antibody (pAb) verified against *Chlamydia pneumoniae*, *Chlamydia trachomatis*, and *Chlamydia psittaci*, and thus capable of cross-reacting with these species. Using this antibody, *Chlamydia*-positive inclusion bodies were readily visualized by both fluorescence-based (**Fig. 1B**) and peroxidase-based (**Suppl. Fig. 1A,B**) immunolabeling. We then confirmed the specific presence of *Chlamydia pneumoniae* inclusions in retinal cross-sections using an anti-*Chlamydia pneumoniae* monoclonal

antibody (mAb) that does not cross-react with other *Chlamydia* species, again with fluorescence- and peroxidase-based detection (Fig. 1C,D; Suppl. Fig. 1C,D). All quantitative analyses of *Chlamydia pneumoniae* burden reported in this study, including positive cell counts, immunoreactive (IR) area measurements, and correlation analyses, are based exclusively on this *Chlamydia pneumoniae*-specific mAb.”

5. The manuscript describes associations between DEPs and A β ₁₋₄₂ or tau levels (lines 526–527), yet no statistical measures (e.g., correlation coefficients or p-values) are reported to support these claims. Additionally, the assertion that these DEPs are “specifically *Chlamydia*-related proteins” (line 529) overstates their specificity, as many are broadly involved in general bacterial or immune responses. To strengthen this section, we recommend (1) reporting the statistical strength of the associations or referencing the relevant figure/table, and (2) rephrasing to reflect that these proteins have been previously implicated in *Chlamydia*-related interactions, rather than implying exclusivity.

Response 5: To address this point, we made the corresponding revisions in the manuscript: (1) we added the referred correlation coefficients and p-values to the Results text; and (2) as suggested, we revised our manuscript text while avoiding the word “specifically” and rephrasing to avoid overstatement, as follows:

Results (Page 25, Lines 739-753):

“Notably, retinal A β ₁₋₄₂ levels strong-to-very strongly correlated with proteins associated with cell degeneration, including CASP3 ($r_p = 0.77$, $p = 0.0099$), BAG3 ($r_p = 0.76$, $p = 0.012$), and GSDMD ($r_p = 0.89$, $p = 0.0006$), as well as inflammatory regulators such as DCD ($r_p = 0.78$, $p = 0.0084$) and LRRFIP1 ($r_p = 0.81$, $p = 0.0046$) (Suppl. Fig. 8A–E). In contrast, cytoprotective and anti-inflammatory proteins, including TMT1A ($r_p = -0.77$, $p = 0.012$) and AP2A2 ($r_p = -0.88$, $p = 0.0008$), were inversely correlated with retinal A β ₁₋₄₂. Similar to retinal amyloidosis, *Chlamydia* interactome proteins also exhibited significant associations with retinal (0N4R) tau isoform and brain (1N3R, 2N4R) tau isoforms (Suppl. Fig. 8F–J and Suppl. Fig. 7G,H). In particular, retinal 0N4R tau strong-to-very strongly correlated with GSDMD ($r_p = 0.65$, $p = 0.022$), BAG3 ($r_p = 0.85$, $p = 0.0005$), RAD23B ($r_p = 0.90$, $p < 0.0001$), LRRFIP1 ($r_p = 0.81$, $p = 0.0015$), CAST ($r_p = 0.91$, $p < 0.0001$), and TMT1A ($r_p = -0.92$, $p < 0.0001$; Suppl. Fig. 8F–J). Overall, these findings indicate enrichment of proteins implicated in intracellular Gram-negative bacterial infection, including *Chlamydia*-associated proteins, together with signatures of inflammasome activation and degeneration in AD brain and retina.”

6. As IBA is also a marker for macrophages, has another marker specific for microglia been studied, such as Cx3cr1, to confirm the result that Cp phagocytosis by microglia is impaired in AD patients?

Response 6: We thank the reviewer for raising this important point. To address it, we systematically tested additional microglial markers on FFPE human retinal sections co-

stained with IBA1. CX3CR1 (Cat. #ab8020 and Cat. #149048; two antibodies), as suggested by the reviewer and known to label both microglia and macrophages, did not yield reliable microglial staining in the human retina under our conditions; no clear IBA1/CX3CR1 co-labeling was observed. P2RY12 (Cat. #HPA014518) produced only faint labeling of IBA1-positive cells and additionally stained Müller glia, based on their morphology and radial localization, thereby limiting its specificity and applicability in human FFPE retinal tissue. By contrast, TMEM119 (transmembrane protein 119; Cat. #PA5-62505) robustly and specifically co-labeled the vast majority of IBA1-positive cells in the human retina, and all observed IBA1-positive microglia associated with *Chlamydia pneumoniae* co-expressed TMEM119. In the revised manuscript, we now include representative images of retinal IBA1⁺TMEM119⁺ microglia involve in recognition, engulfment, and ingestion of *Chlamydia pneumoniae* (Fig. 5N, below; expanded views and separate channels in Suppl. Fig. 12D).

In the human brain, TMEM119 is widely accepted as a marker of resident microglia rather than blood-derived macrophages or perivascular macrophages (PMID: 26250788), and single-cell and multiplex immunofluorescence studies further demonstrate that TMEM119, together with IBA1, defines microglial clusters in both control and AD brains (e.g., PMID: 35247551). Complementary work in human retina shows that TMEM119 is enriched in the microglial cluster by single-cell RNA-seq (PMID: 31653841) and is expressed by iPSC-derived microglia engrafted into neuroretina (PMID: 39514271). In murine retina, TMEM119 has been repeatedly used as a microglia-specific marker in models of retinal injury (e.g., PMID: 31853425; experimental autoimmune uveoretinitis, PMID: 39930455), and even when TMEM119 expression is partially downregulated in activated subretinal microglia, co-expression with IBA1 and APOE confirms their microglial origin (PMID: 36088481).

Taken together, our co-localization data and these prior studies support the conclusion that the IBA1/TMEM119 double-positive cells engaging *Chlamydia pneumoniae* inclusions in the human retina represent resident microglia rather than infiltrating macrophages/perivascular macrophages. We have clarified this point and the rationale for marker selection in the Results and Discussion sections of the revised manuscript, as specified below.

Results (Page 31, Lines 939-945):

“To further verify that *Chlamydia pneumoniae*-associated IBA1⁺ cells were resident microglia rather than perivascular or infiltrating macrophages, we co-labeled retinal

sections with transmembrane protein 119 (TMEM119), a marker enriched in resident microglia in human brain and retina (**Fig. 5N**, $n = 12$; extended images in **Suppl. Fig. 12D**)¹²¹⁻¹²⁴. Strikingly, all observed IBA1⁺ cells engaged in recognition, engulfment, or ingestion of *Chlamydia pneumoniae* inclusions co-expressed TMEM119, indicating a microglial identity. We refer to these cells as *Chlamydia pneumoniae*-associated microglia (CAM).”

Discussion (Page 38, Lines 1144-1148):

“Retinal IBA1⁺TMEM119⁺ resident microglia frequently surround or internalize *Chlamydia pneumoniae* inclusions, yet the proportion of *Chlamydia pneumoniae*-associated microglia (CAM) relative to bacterial burden is reduced by ~61% in AD compared with controls and inversely correlated to bacterial load, implying impaired microglial recognition and/or phagocytosis of infected cells.”

7. In the discussion, there is no mention of cleaved Caspase 3, even though it is mentioned in the abstract and the results.

Response 7: In the revised manuscript cleaved caspase-3 (CCasp3) is mentioned in several Discussion texts, as specified below:

Discussion (Page 37, Line 1107): “... and associate with CCasp3 apoptosis, NGSDMD pyroptosis, retinal and brain atrophy, higher Braak stage, and worse MMSE.”

Discussion (Page 38, Line 1143): “...inflammasome components, CCasp3-defined apoptosis, A β ₄₂ and oligomeric tau, and are further...”

Discussion (Pages 38, Line 1161): “Finally, machine-learning models incorporating retinal *Chlamydia pneumoniae*, NLRP3, and CCasp3, alone and in combination with A β ₄₂, gliosis, and atrophy, demonstrate that integrating...”

- Although the study provides compelling correlations between *Chlamydia pneumoniae* and various AD-related pathologies in the retina and brain, a few fundamental limitations temper the strength of its conclusions. First, the study does not strongly establish a causal relationship between Cp infection and the progression of AD. Although increased Cp inclusions are observed in MCI and AD tissues, these remain associative and lack evidence demonstrating that Cp initiates or accelerates neurodegeneration. Second, the specificity of the Cp detection methods, particularly the use of a polyclonal antibody that may cross-react with other *Chlamydia* species. Although monoclonal antibodies were later used for specificity, the conclusion of Cp, especially within complex tissues like retina and brain, could still be confounded by non-specific staining. Third, the analyses largely rely on correlative data without functional validation. Many conclusions hinge on statistical associations rather than experimentally dissecting the mechanistic pathways by which Cp might induce inflammasome activation or contribute to tau and amyloid pathology.

Response: We appreciate the reviewer’s feedback. In the revised manuscript we addressed the raised points as follows:

1) **To address the first point** of a causal role for *Chlamydia pneumoniae* (initiating or accelerating) in AD, we have now conducted two *in vivo* studies in *Chlamydia pneumoniae*-infected AD⁺ mice and complementary *in vitro* experiments in human neuroblastoma cells. The new experiments are described in revised **Materials and Methods (Pages 14-17, Lines 396-514)**, **Results section 3 (Pages 25-28, Lines 755-827)**, **Discussion (Page 34, 1035-1037, Pages 36-37, Lines 1093-1104)**, **Figure 3**, and **Supplementary Figure 9** (see below).

- **In vitro Results (Pages 25-26, Lines 757-776):** “To determine whether *Chlamydia pneumoniae* acts as a driver rather than a bystander in AD, we next tested whether infection of neuronal cells and AD transgenic mice is sufficient to trigger inflammasome activation and exacerbate AD-related pathology. Infection of SH-SY5Y human neuroblastoma cells with *Chlamydia pneumoniae* (multiplicity of infection [MOI] 5) for 68 hours markedly induced A β ₄₂, NLRP3, and IL1 β levels, and triggered cell membrane damage as assessed by lactate dehydrogenase (LDH) release (**Fig. 3A–F**). Immunocytochemistry confirmed robust *Chlamydia pneumoniae* infection of SH-SY5Y neurons and revealed that infected cells compared with uninfected controls exhibited 2.5-fold higher NLRP3, 3.2-fold higher IL1 β , and 3.5-fold higher H31L21⁺ A β ₄₂ IR areas (all $p < 0.0001$; **Fig. 3B,C**; extended data in **Suppl. Fig. 9A**; $n = 6$ wells per condition, $n = 41$ –74 cells per group). Moreover, pronounced 3.9-fold increase in LDH leakage was detected in infected neuronal cells versus uninfected controls (**Fig. 3D**, $p < 0.001$, $n = 6$ wells per group), as assessed by LDH release into the culture medium, suggesting that *Chlamydia pneumoniae* infection induces neurotoxicity. We further substantiated these findings by Western blot analysis, which demonstrated increases in NLRP3, cleaved IL1 β (1.5 fold, $p < 0.05$), 12F4⁺ A β ₄₂ (6.7 fold, $p < 0.05$), and NGSDMD in infected neurons compared with controls (**Fig. 3E,F**; $n = 3$ –6 wells per group). Together, these findings demonstrate that *Chlamydia pneumoniae* infection is sufficient to drive NLRP3 inflammasome activation, pyroptotic cell death, and A β ₄₂ accumulation in neuronal cells—cellular features of AD pathology. Future work will be required to delineate the molecular pathways by which *Chlamydia pneumoniae* amplifies A β production, sustains inflammasome signaling, and ultimately promotes neurodegeneration.”

New **Figure 3A–F**: *C. pneumoniae*-infection in human neuroblastoma SH-SY5Y cells:

- In vivo in APP_{SWE}/PS1_{ΔE9} (AD⁺) mice Results (Pages 26-28, Lines 777-827): “These *in vitro* observations prompted us to examine the impact of acute and long-term *Chlamydia pneumoniae* infection on *in vivo* Alzheimer-like pathology and cognition in APP_{SWE}/PS1_{ΔE9} (AD⁺) mouse models (**Fig. 3G–S**). In the acute *Chlamydia pneumoniae* infection paradigm, we examined 8 PBS-treated AD⁺ controls and 14 infected AD⁺ mice ($n = 22$). In the long-term paradigm, we studied 6 PBS-treated WT, 8 PBS-treated AD⁺, and 9 infected AD⁺ mice ($n = 23$). Intranasal inoculation with *Chlamydia pneumoniae* (1×10^6 IFU) resulted in a marked increase in bacterial inclusions in the AD⁺ mouse brain (**Fig. 3G,H**), as confirmed by higher *Chlamydia pneumoniae* IFUs in HEp2 cells treated with brain lysates from infected versus uninfected mice and by increased *Chlamydia pneumoniae* genomic DNA copy numbers (**Fig. 3H**). Seven-days (acute) *Chlamydia pneumoniae* postinfection caused a 3.2-fold increase in IBA1⁺ microgliosis and a 2.5-fold increase in GFAP⁺ astrogliosis in the hippocampi and cortices of infected versus uninfected AD⁺ mice (**Fig. 3I, J**; $p < 0.05$ – 0.01 ; extended data in **Suppl. Fig. 9B–D**), indicating amplified neuroinflammation due to infection. Furthermore, infected AD⁺ mice exhibited elevated mRNA expression of *Il6* ($p < 0.01$), *Il1β* ($p < 0.05$), and *Nlrp3* ($p < 0.01$) (**Fig. 3K**), further supporting activation of NLRP3-inflammasome signaling in response to *Chlamydia pneumoniae* infection. These findings demonstrate that acute intranasal infection is sufficient for *Chlamydia pneumoniae* to reach the brain and establish infection, subsequently triggering inflammasome activation and neuroinflammation.

The long-term behavioral and pathological consequences of *Chlamydia pneumoniae* infection in AD⁺ mice were assessed 6 months after a single intranasal inoculation (1×10^6 IFU *Chlamydia pneumoniae* or PBS) administered at 8 months of age (**Fig. 3L–S**; extended data in **Suppl. Fig. 9E–J**). Multi-domain behavioral testing was conducted over 12 days in 14-month-old mice, with PBS-treated WT animals serving as healthy behavioral controls. In the open field and X-maze tests, *Chlamydia pneumoniae* infection did not affect locomotor function of AD⁺ mice, as indicated by rearing and ambulatory activity or total arm entries (**Fig. 3L and Suppl. 9E,H**). However, alternations in both color and contrast-stimuli modes of the X-maze, which assess visuo-cognitive function and decreased in AD⁺ mice¹⁰⁰, were further reduced in *Chlamydia pneumoniae*-infected AD⁺ mice (**Fig. 3M, N**). In the color-mode X-maze, while infection did not affect specific arm entries, it further decreased bidirectional blue (B)↔white (W) transitions in infected AD⁺ mice, which indicates color vision dysfunction (**Fig. 3M and Suppl. Fig. 9F,G**). In the contrast-mode X-maze, entries in the arm with the white object were increased in the PBS-control AD⁺ mice and further increased in the *Chlamydia pneumoniae*-infected AD⁺ mice (**Fig. 3N and Suppl. Fig. 3I**). The infected AD⁺ mice also exhibited increased black (B)↔white (W) and decrease of black (B)↔clear (C) bidirectional transitions, indicating a worsening of contrast sensitivity vision due to infection (**Suppl. Fig. 3J**). In the Barnes maze, PBS-control AD⁺ mice (vs. WT) made significantly more errors prior to finding the escape box, during the 4-day acquisition phase, the long-term memory retention phase, and the 2-day reversal phase (**Fig. 3O**). Importantly, *Chlamydia pneumoniae* infection in AD⁺ mice further increased the number of errors made on reversal day 9 (**Fig. 3O,P**), which measures spatial learning and cognitive flexibility. Search coverage analysis showed that *Chlamydia pneumoniae*-infected AD⁺ mice made more errors locally in the area that is both on the side of the old and new escape box locations (**Fig. 3P**, right

panel). These results indicate that long-term *Chlamydia pneumoniae* infection exacerbates visuo-cognitive dysfunctions in AD⁺ mice, without affecting locomotor function. We subsequently examined AD-related pathology in the cortex and hippocampus of long-term infected versus uninfected AD⁺ mice (Fig. 3Q-S). Our analysis revealed significant increases in 6E10⁺ A β plaques (1.6-fold, $p < 0.001$), IBA1⁺ microglia (1.3-fold, $p < 0.01$), and GFAP⁺ astrocytes (1.3-fold, $p < 0.05$) in the cortex of *Chlamydia pneumoniae* infected AD⁺ mice compared with PBS-administered AD⁺ mice (Fig. 3R). Similar increases were also observed for the hippocampus (Fig. 3S). These findings demonstrate that long-term *Chlamydia pneumoniae* infection in AD⁺ mice aggravates neuroglial activation and A β pathology, supporting the hypothesis that chronic infection exacerbates AD-like neuropathology.”

New **Figure 3G-S**: Acute and chronic *C. pneumoniae*-infection APP_{SWE}/PS1 Δ E9 (AD⁺) mice:

Discussion (Page 34, Lines 1035-1037): “Our functional data show that *Chlamydia pneumoniae* infection is sufficient to induce NLRP3 activation, IL1 β maturation, neurotoxicity, and A β ₄₂ accumulation in human neurons and to exacerbate neuroinflammation, A β plaque burden, and visuocognitive impairment in AD⁺ mice.”

Discussion (Page 36-37, Lines 1093-1104): “Our mechanistic data support a model in which *Chlamydia pneumoniae*, like other gram-negative pathogens, activates NLRP3 inflammasomes¹⁴⁹ and thereby couples infection to neuroinflammation and neurodegeneration. In human neuroblastoma cells, *Chlamydia pneumoniae* infection is sufficient to increase NLRP3, IL1 β , NGSDMD, and A β ₄₂, and induce LDH release, recapitulating key cellular features of AD pathology. In AD⁺ mice, intranasal *Chlamydia pneumoniae* inoculation establishes brain infection and is accompanied by upregulation of *Nlrp3*, *Il1 β* , and *Il6* transcripts, amplified IBA1⁺ microgliosis and GFAP⁺ astrogliosis, and increased cortical and hippocampal A β plaques. Moreover, infected AD⁺ mice show worsened visuocognitive performance in X-maze and Barnes maze tasks, consistent with infection-driven aggravation of neuroinflammation, amyloidosis, and cognitive decline. To our knowledge, this is the first demonstration that *Chlamydia pneumoniae* infection exacerbates brain pathology and behavioral dysfunction in a mouse model of AD.”

In summary, while we agree that human postmortem data alone are inherently associative, our functional experiments in human neurons and AD⁺ mice directly address the question of causality and move the work beyond correlation. In SH-SY5Y neurons, *Chlamydia pneumoniae* infection is sufficient to induce NLRP3 inflammasome activation, IL1 β maturation, GSDMD-mediated pyroptotic injury, and marked A β ₄₂ accumulation, recapitulating core cellular features of AD pathology in the absence of any other AD-related insult. In AD⁺ mice, a single intranasal inoculation of *Chlamydia pneumoniae* establishes persistent brain infection and, on acute and/or long-term timescales, exacerbates hippocampal and cortical microgliosis, astrogliosis, *Nlrp3/Il1 β /Il6* upregulation, A β plaque burden, and visuocognitive deficits, without affecting locomotor function. Together, these data demonstrate that *Chlamydia pneumoniae* is not merely a bystander but can drive and aggravate AD-like neuropathology and cognitive impairment in susceptible brains. We therefore conclude that, while *Chlamydia pneumoniae* is unlikely to be the sole initiating event in all AD, our mechanistic *in vitro* and *in vivo* findings provide strong evidence that infection can act as a causal amplifier of inflammasome activation, amyloidosis, and neurodegeneration, directly addressing the reviewer’s concern about causality.

2) **To address the second point** related to the specificity of the *Chlamydia pneumoniae* detection methods, please see our extensive response 1 to **Reviewer #2** (above) for the additional experiments added to validate the presence of *Chlamydia pneumoniae* in the human retina. In summary, we performed three additional, independent validation assays that now provide convergent lines of evidence: (i) consistent detection of inclusion-like structures with both a broadly reactive pAb and a *Chlamydia pneumoniae*-specific mAb in retina and matched cortex; (ii) morphologically compatible inclusion bodies by Giemsa staining; (iii) species-specific genomic signal by FISH with appropriate negative controls;

and (iv) detection of the *Chlamydia pneumoniae*-specific *argR* gene by qPCR, provide robust, multi-modal support for the presence of *Chlamydia pneumoniae* in the human retina and its preferential elevation in AD. As specified above, these data are now included in the revised **Materials and Methods (Pages 12-13)**, **Figure 1G-I** and **Supplementary Figures 3-4**, **Results (Page 21-22, Lines 637-652)**, and **Discussion (Page 34, Lines 1025-1027; Page 35, Lines 1060-1064)**.

3) **To address the third concern** that the original study was predominantly correlative and would benefit from additional functional experiments and clarification of how *Chlamydia pneumoniae* drives inflammasome activation and AD pathology, we have now incorporated functional-cognitive data in AD⁺ mice. In addition to the extensive and novel findings in human MCI/AD retinas and brains, we now demonstrate that *Chlamydia pneumoniae* infection directly triggers NLRP3 inflammasome activation, neurotoxicity, and A β ₄₂ production in neuronal cells, and in vivo exacerbates brain gliosis, NLRP3 inflammasome activation, A β plaque burden, and cognitive impairment in AD⁺ mice (as detailed above). A full dissection of the molecular pathways by which *Chlamydia pneumoniae* drives inflammasome activation and AD-related neurodegeneration remains beyond the scope of the current study; these outstanding mechanistic questions and proposed future directions are outlined in the revised Discussion/Limitations section of the revised manuscript (see below).

Discussion (Page 37, Lines 1121-1123): “...*Chlamydia pneumoniae* can plausibly function as the secondary activating cue that fully engages the NLRP3 inflammasome, thereby further potentiating inflammation and neurodegeneration in the AD retina, a hypothesis that warrants direct mechanistic testing.”

Discussion (Page 38, Lines 1136-1137): “...*Chlamydia pneumoniae* that intersects with core AD pathways, although mechanistic dissection will require future studies.”

Discussion, Limitations (Page 39, Lines 1177-1184): “...comprehensive histological and biochemical evidence of infection and immune activation remain correlative, and targeted functional studies are needed to dissect how specific *Chlamydia*-interactome proteins and inflammasome components mechanistically couple infection to retinal and brain neurodegeneration and dysfunction. Finally, the human data are cross-sectional, so although our neuronal and AD⁺ mouse experiments strongly support a mechanistic link between *Chlamydia pneumoniae*, NLRP3 activation, AD-like pathology, and cognitive decline, they do not establish causality in patients, questions best addressed by longitudinal and interventional clinical studies.”

In conclusion, this is an important study that is likely to generate a lot of discussion on the role of infection in AD. The authors may consider addressing the concerns raised regarding the paper.

We thank the reviewers for their concluding and positive comment.

Reviewer #4.

Thank you. We have addressed all the comments as detailed above.

REVIEWERS' COMMENTS

Reviewer #1 (Remarks to the Author):

The authors took significant time in addressing all of the reviewers' comments. Clarity, experimental depth, reanalysis, and new supporting findings successfully combine to convincingly answer the hypothesis. The additional experiments provide the necessary data to support the hypothesis and expand the construction at home. and further refine existing runs with weak n.

I support this revised manuscript moving forward toward publishing.

We thank Reviewer #1 for the supportive comments and endorsement of the revised manuscript.

Reviewer #2 (Remarks to the Author):

The authors have addressed my previous comments, and I am now supportive of publication of the manuscript.

We thank Reviewer #2 for the positive feedback and support for publication of the revised manuscript.

Reviewer #3 (Remarks to the Author):

As my co-reviewer and I stated before this was a compelling work. The authors have addressed most of the reviewers' comments. Although they did not get positive results from one of the microglia markers that was suggested but their rationale for using TMEM119 was justified. Most importantly this work will be another important stepping stone in cementing the role of infection in AD progression. This work when published will likely generate a lot of interesting discussions.

We thank Reviewer #3 for the thoughtful and favorable response; we are pleased that the revisions and additional experiments adequately addressed the reviewer's concerns and strengthened the manuscript.

Reviewer #4 (Remarks to the Author):

We thank Reviewer #4 for the time and contribution as part of the co-review process.